# NAP-seq reveals multiple classes of structured noncoding RNAs with regulatory functions

Shurong Liu[1,5], Junhong Huang[1,2,5], Jie Zhou[1], Siyan Chen[1,2], Wujian Zheng[1], Chang Liu[1], Qiao Lin[1], Ping Zhang[1], Di Wu[1,2], Simeng He[2], Jiayi Ye[1], Shun Liu[3], Keren Zhou [4], Bin Li [1] ✉, Lianghu Qu [1] ✉ & Jianhua Yang [1,2] ✉

Up to 80% of the human genome produces "dark matter" RNAs, most of which are noncapped RNAs (napRNAs) that frequently act as noncoding RNAs (ncRNAs) to modulate gene expression. Here, by developing a method, NAP-seq, to globally profile the full-length sequences of napRNAs with various terminal modifications at single-nucleotide resolution, we reveal diverse classes of structured ncRNAs. We discover stably expressed linear intron RNAs (sliRNAs), a class of snoRNA-intron RNAs (snotrons), a class of RNAs embedded in miRNA spacers (misRNAs) and thousands of previously uncharacterized structured napRNAs in humans and mice. These napRNAs undergo dynamic changes in response to various stimuli and differentiation stages. Importantly, we show that a structured napRNA regulates myoblast differentiation and a napRNA DINAP interacts with dyskerin pseudouridine synthase 1 (DKC1) to promote cell proliferation by maintaining DKC1 protein stability. Our approach establishes a paradigm for discovering various classes of ncRNAs with regulatory functions.

More than 80% of the human genome can be transcribed into RNA[1] by three RNA polymerases: RNA polymerase I (Pol I)[2–4], RNA polymerase II (Pol II)[5,6] and RNA polymerase III (Pol III)[4,7,8]. These transcribed RNAs are further processed into capped RNAs (capRNAs)[9,10] and noncapped RNAs (napRNAs). Almost all capRNAs are messenger RNAs (mRNAs), which are mainly translated into proteins, but napRNAs often serve as noncoding RNAs (ncRNAs) to regulate gene expression[3,4,9,11]. However, in contrast to the advanced annotation of capRNAs, which compose less than 5% of the human transcriptome[1], the genome-wide prevalence, mechanism and function of napRNAs are very poorly understood.

The identification of short napRNAs (<50 nt) by small RNA-seq (sRNA-seq) technology has led to the discovery of several classes of functional small RNAs, such as microRNAs (miRNAs), PIWI-interacting

RNAs (piRNAs), endogenous siRNAs, and 21U RNAs[12–14]. However, almost all studies involving sRNA-seq have focused on small napRNAs (<50 nt) rather than all napRNAs of various lengths and diverse terminal modifications, especially long napRNAs (≥100 nt)[15]. In general, given the limitation of sequencing length, traditional RNA-seq often involves the generation of cDNA via reverse transcription (RT) of fragmented RNAs with random hexamer primers and cannot determine the full-length sequence[16–18], which is essential for classifying napRNAs and ascertaining their precise secondary structure and functional potential. Even TGIRT-seq using thermostable group II intron reverse transcriptase[19] could not detected long RNA molecules which is also limited by the sequencing length. Thus, although napRNAs compose the majority of the human transcriptome, the well-annotated napRNAs are just the tip of the iceberg of the complete

[1]MOE Key Laboratory of Gene Function and Regulation, State Key Laboratory of Biocontrol, School of Life Sciences, Sun Yat-sen University, Guangzhou 510275 Guangdong, China. [2]The Fifth Affiliated Hospital, Sun Yat-sen University, Zhuhai 519082 Guangdong, China. [3]Department of Chemistry, The University of Chicago, Chicago, IL 60637, USA. [4]Department of Systems Biology, Beckman Research Institute of City of Hope, Monrovia, CA 91016, USA. [5]These authors contributed equally: Shurong Liu, Junhong Huang. ✉e-mail: libin73@mail.sysu.edu.cn; lssqlh@mail.sysu.edu.cn; yangjh7@mail.sysu.edu.cn

population of napRNAs, and an important dimension of transcriptome complexity has remained largely unexplored.

To discover previously undescribed classes of napRNAs that have escaped detection by traditional sRNA-seq and RNA-seq, we develop an approach that we termed napRNA sequencing (NAP-seq), which is performed by full-length sequencing of napRNAs on the Nanopore and Illumina sequencing platforms. Using NAP-seq, we compile human and mouse napRNA transcriptomes and analyze their evolutionary conservation as well as their response to different cellular conditions. Importantly, we detect diverse classes of napRNAs, including stably expressed linear intron napRNAs (sliRNAs), a class of snoRNA-intron (snotron) napRNAs and a class of napRNAs embedded in miRNA spacer regions (misRNA). Simultaneously, thousands of previously uncharacterized structured napRNAs and repetitive-element-derived napRNAs (repRNAs) are discovered in humans and mice. Furthermore, we demonstrate that a structured napRNA regulates myoblast differentiation in mice and a previously undescribed napRNA, DINAP—a composite box C/D-H/ACA RNA—can interact with DKC1 to maintain its protein stability, which further promotes the proliferation of HepG2 cells. Considering these results collectively, we define the transcriptome-wide landscape of full-length napRNAs and provide approaches for classifying napRNAs and elucidating the roles of napRNAs associated with physiological and pathological processes.

## Results

### NAP-seq accurately identifies the full-length sequences of napRNAs with various terminal modifications at single-nucleotide resolution

To detect all full-length napRNAs with diverse terminal modifications across the human transcriptome, we developed NAP-seq by leveraging a combination of three enzymatic treatments (T4PNK, RNase H and SuperScript IV), specific designs of adapters and the nested RT primer, and two different sequencing platforms (Oxford Nanopore and Illumina) (Fig. 1a). The detailed steps were as follows: (1) To identify napRNAs with various terminal modifications, total RNA was initially treated with T4 polynucleotide kinase (T4 PNK), which removed the 3′-phosphate groups from 3′-monophosphates, 3′-diphosphates and 2′,3′-cyclic phosphates (cPs) and catalyzed the replacement of 5′-hydroxyl (OH) groups with 5′-phosphate (P) groups to ensure that all napRNAs harbored a 5′-P and 3′-OH at both termini (Fig. 1a). (2) To discover and enrich long napRNAs missed by traditional sRNA-seq methods, we removed short RNAs (<100 nt) by separating fractions on ZYMO columns. (3) To provide a high ligation efficiency and construct the strand-specific and quantitative full-length library, specific randomized barcode-containing (6N, where N represents A, T or U, G, or C) 5′ adapters and 3′ adapters were synthesized and directly ligated to napRNAs (Supplementary Fig. 1a, see "Methods"). (4) To increase the diversity of napRNA species and discover low-expression napRNAs, we used the RNase H method to remove high-abundance ribosomal RNAs (rRNAs), small nuclear RNAs (snRNAs) and small nucleolar RNAs (snoRNAs)[20] (Fig. 1a and Supplementary Fig. 1b, c). (5) To avoid widespread mispriming artefacts[21–24] and obtain full-length cDNAs, we used a nested RT primer for reverse transcription and a full-length reverse PCR primer for pre-amplification (Supplementary Fig. 1a). (6) To overcome the difficulties imposed by RNA modifications and stable RNA secondary structures during cDNA synthesis, we used the SuperScript IV RT enzyme, with a high level of processivity and thermostability, to generate cDNAs. (7) The full-length cDNAs were then divided into two aliquots, and one aliquot was used to construct third-generation sequencing (TGS) libraries for sequencing on the Oxford Nanopore platform (referred to as NAP-seq-TGS, Fig. 1a). (8) Given the high error rates of nanopore sequencing, the other aliquot of cDNA was fragmented to construct next-generation sequencing (NGS) libraries for Illumina sequencing (referred to as NAP-seq-NGS, Fig. 1a).

We next performed NAP-seq on samples from human cell lines—HepG2, HEK293T and U87, as well as a mouse cell line C2C12, and obtained ~46 million reads and 10 million reads per library, on average, through deep sequencing (Supplementary Data 1) by NAP-seq-NGS and NAP-seq-TGS, respectively. On average, more than 50% of the reads in each library contained at least one specific adapter sequence at the terminus (Fig. 1b), which unambiguously determined the 5′-start and 3′-end sites at single-nucleotide resolution and the strand orientation of each napRNA (Supplementary Fig. 1a). Notably, the NAP-seq reads were highly correlated between each pair of biological replicates (Fig. 1c, d, Pearson correlation coefficients: 0.96 and 0.95), indicating the high robustness of the NAP-seq method. In addition, the NAP-seq-NGS and NAP-seq-TGS reads were highly correlated (Supplementary Fig. 1d) and had a similar distribution of annotated gene types (Supplementary Fig. 1e). Furthermore, by a global analysis of NAP-seq-NGS and NAP-seq-TGS in HepG2 and C2C12 (Supplementary Fig. 1f), we found that majorities of the results obtained from NAP-seq-NGS are supported by NAP-seq-TGS experiments conducted in both humans and mice (Supplementary Fig. 1g). To verify the enriched RNA in NAP-seq are napRNAs, we carried out polyA-selected RNA-seq experiments to investigate the extent of non-capped RNA enrichment. As a result, we found that NAP-seq exhibited a significant enrichment of known non-capped ncRNAs (such as snoRNAs) in comparison to the polyA-selected RNA-seq approach conducted on the same cell lines (Supplementary Fig. 2a). To confirm the removal of capped RNAs in NAP-seq, we also performed CAP-seq to identify 66662 high-confidence 5′-cap sites within HepG2 cell lines (Supplementary Fig. 2b). When comparing these 66662 5′-cap sites to the 5′-start sites of napRNAs identified through NAP-seq, we found that only three sites (0.2% = 3/1220) exhibited overlap (Supplementary Fig. 2b). Upon careful analysis of the terminal characteristics of these three identified RNAs in NAP-seq library, we found that their expression levels were very low and lacked significant 5′ start site signals (Supplementary Fig. 2c). These results indicated that NAP-seq can specifically capture napRNAs. Importantly, the start and end sites of known napRNAs could be precisely identified at single-nucleotide resolution (Fig. 1e–h). For example, the end sites of the reads from both NAP-seq-NGS and NAP-seq-TGS were significantly enriched in the termini of the known snoRNAs ACA48, mgU6-77 and U67 (Fig. 1i). Given that the accuracy of NGS is much higher than that of TGS, the NAP-seq-NGS reads (Fig. 1e, f) coincided with more known ncRNA ends than did the NAP-seq-TGS reads (Fig. 1g, h). To validate the reliability of NAP-seq in different cell lines and explore the dynamic and cell type-specific napRNAs, we further evaluated NAP-seq-NGS in two other human cell lines—embryonic kidney HEK293T cells (Supplementary Fig. 2d) and U87 cells (Supplementary Fig. 2e), which are derived from the brain—and found that the 5′ and 3′ ends of napRNAs identified by NAP-seq were highly reproducible in these cell lines (Supplementary Fig. 2d, e).

Collectively, these results showed that our NAP-seq approach, especially NAP-seq-NGS, could not only capture napRNAs with high specificity and accuracy but also determine their full-length sequences at single-nucleotide resolution.

### NAP-seq discovered structured napRNAs with dynamic expression profiles and histone modifications

To demonstrate the utility of NAP-seq-NGS for identifying previously uncharacterized napRNAs compared to the GENCODE database (human release 30 and mouse release 23)[25], in different cells and under different conditions, we further profiled the transcriptomes of 3 different stress responses (poly (I:C), adriamycin (ADR), and CoCl$_2$) in human HepG2 cells and 4 different

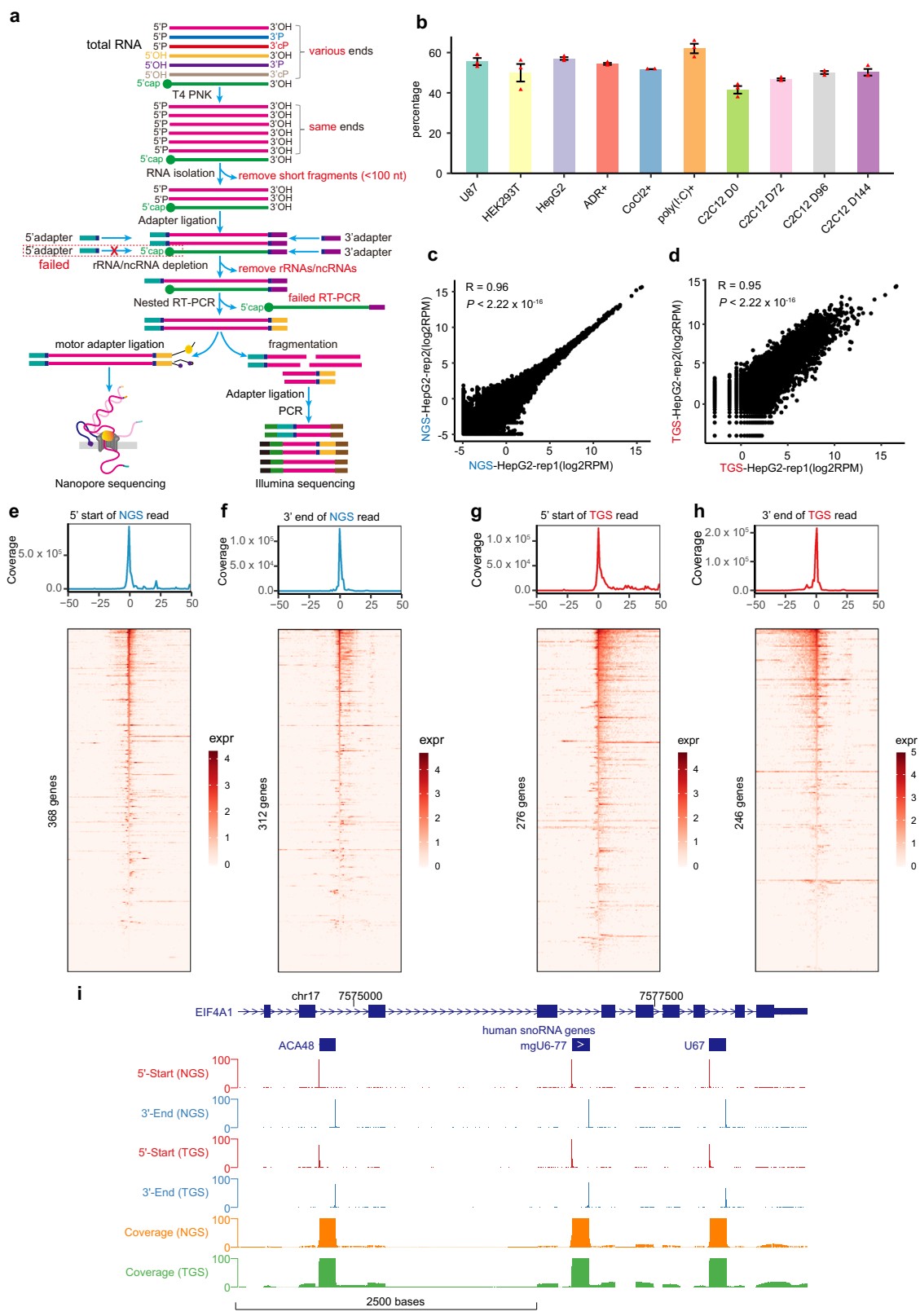

myoblast differentiation stages in the mouse skeletal muscle cell line C2C12, as well as the abovementioned three cell lines. In these stress responses, poly (I:C) mimics double-stranded RNA to induce the activation of immune response[26]; ADR causes DNA damage, triggering cells to activate their DNA damage repair machinery[27,28]; $CoCl_2$ artificially induces hypoxia[29], which has allowed the characterization of the hypoxia response at the

cellular, biochemical and molecular levels[30]. Moreover, we developed a computational tool, napSeeker, to determine the full-length sequences of napRNAs from total NAP-seq data (Supplementary Fig. 3a, see "Methods"). We further filtered the candidate napRNAs identified by napSeeker through requiring that the napRNA must exist in at least two sequencing libraries, have more than 20 sequencing reads and not overlap with known

**Fig. 1 | NAP-seq accurately identifies full-length napRNAs at single-nucleotide resolution. a** Schematic depiction of the NAP-seq method with the construction of TGS libraries and NGS libraries. Total RNA was extracted and pre-size selection of RNA fractions was performed by removing short RNAs. **b** The percentage of reads with at least one specific adapter in the NAP-seq-NGS libraries in humans and mice (3 biological replicates per sample). All data are plotted as the means ± SEMs. The number of reads (RPM, reads per million) for known ncRNAs was highly reproducible between replicates with both the NAP-seq-NGS (**c**) and NAP-seq-TGS (**d**) methods. *p* values were calculated by two-sided Pearson's correlation test. **e**, **f** Coincidence analysis of both terminal sites in the NAP-seq-NGS reads and those in the known ncRNAs in HepG2 cells. The *x*-axis shows the distance from the annotated 5′-start (**e**) or 3′-end (**f**) site in the NAP-seq-NGS read to the corresponding terminal sites in the known ncRNAs, and the *y*-axis shows the number of reads (RPM) within a certain distance from the 5′-start or 3′-end site. The bottom panel shows a heatmap, in which each row represents a gene that matches the terminal sites with the NAP-seq-NGS read exactly, and each column represents the expression values of genes at a specific distance. expr, expression value. **g** Number of coincidences between the 5′-start sites in NAP-seq-TGS reads and the 5′-start sites in known ncRNAs in HepG2 cells. **h** Number of coincidences between the 3′-end sites in NAP-seq-TGS reads and the 3′-end sites in known ncRNAs in HepG2 cells. **i** Genome Browser visualization of NAP-seq 5′-start, 3′-end and coverage signals (RPM, reads per million), which were generated from NAP-seq-NGS and NAP-seq-TGS data, in a 5000-bp region of a human snoRNA cluster (chr17:7,573,750-7,578,750).

annotations, such as rRNAs, tRNAs, snoRNAs and CDSs (coding sequences) (Supplementary Fig. 3a, see "Methods"). The napRNAs identified by NAP-seq-NGS in humans (Supplementary Data 2) were mainly located within introns (40.84%), Alu elements (21.05%) and other regions across the genome (Supplementary Fig. 3b). Interestingly, the genomic distribution of napRNAs in mice was distinct from that in humans, and more than 65% of mouse napRNAs were highly enriched in the repetitive element B2 (Supplementary Fig. 3c). Notably, these napRNAs were longer than known structural ncRNAs (e.g., miRNAs, tRNAs and snoR-NAs) (Supplementary Fig. 3d), suggesting that they were missed by traditional sRNA-seq. Importantly, we found that the napRNAs had lower minimum free energies (MFEs) than random sequences by evaluating their thermodynamic stability using RNAfold (Supplementary Fig. 3e), implying a folded or partially folded structure (detail see "Methods").

To validate the secondary structure of napRNAs in vivo, we developed NAP-SHAPE-MaP, an approach combining SHAPE-MaP[31–35] and our NAP-seq to probe intact RNA structures of napRNAs ("Methods"). NAP-SHAPE-MaP analysis showed that the increase in SHAPE reactivity score was more significant in single-stranded regions than paired regions (Supplementary Fig. 3f), consistent with previous observations[32,33]. We also obtained accurate structure reactivity for non-capped snoRNAs with known secondary structures (Supplementary Fig. 3g). These results indicated the accuracy of NAP-SHAPE-MaP to examine the RNA structure. NapRNAs with a significantly conserved RNA secondary structure identified by RNAalifold software[36] were also verified by our NAP-SHAPE-MaP method (Supplementary Fig. 3h, i). Together, these results show that our NAP-seq method can capture long and structured napRNAs that are missed by previous sRNA-seq and RNA-seq methods.

To investigate whether these expressed napRNAs are conserved across species, we constructed homologous families based on sequence similarity and inferred a stringent minimum evolutionary age of the napRNAs. We discovered that most (14,284) napRNA families were primate-specific, but 3313 families likely originated more than 90 million years (Myr) ago, and 186 families likely originated more than 300 Myr ago (Supplementary Fig. 4a). Notably, the highly conserved sequences predominantly align with intronic and intergenic regions (Supplementary Fig. 4a). For instance, hsa-napRNA-2120 is highly conserved from human to zebrafish (Supplementary Fig. 4b). To easily access these discoveries, we have built the "napRNA structure" webpage (https://rnasysu.com/napSeq/structure.php) for visualization. To gain preliminary insights into the potential functions associated with these highly conserved napRNAs, which originated more than 300 million years ago, we conducted Gene Ontology (GO) enrichment analysis on the host genes of these napRNAs. The result revealed that these host genes primarily participate in cell morphogenesis and skeletal system development (Supplementary Fig. 4c). In addition, by calculating the steady-state levels of the napRNAs and other functional RNA species using total RNA-seq data (GSE88089)[37] and small RNA-seq (ENCSR000CRX), we found that the expression level of napRNAs was higher than that of lncRNAs, but lower than that of snoRNAs (Supplementary Fig. 5a). By RNA half-life analysis, we observed that the half-lives of most napRNAs are similar to that of many mRNAs, and some napRNAs are with a short half-life (Supplementary Fig. 5b–f).

To investigate the dynamic expression profiles of napRNAs, we analyzed their expression levels across different cell types, stress responses and myoblast differentiation stages. We first examined the cell specificity of napRNAs and identified 9318 differentially expressed napRNAs in three different cell lines (Supplementary Fig. 6a and Data 3). One particularly interesting example is the hsa-napRNA-48, which was highly expressed in U87 cells, moderately expressed in HepG2 wild-type cells and treated HepG2 cells, and expressed at low levels in HEK293T cells (Supplementary Fig. 6b). By comparing the napRNA profiles of untreated HepG2 cells to those of HepG2 exposed to the abovementioned stimuli, we detected 7278 dynamically altered napRNAs under the different treatment conditions (Supplementary Fig. 6c and Data 3). Our analysis also revealed that the expression patterns of most napRNAs exhibit weak correlation with their respective host genes (Supplementary Fig. 6d).

Interestingly, we discovered that 6808 differentially expressed napRNAs displayed significant changes in expression for up to 72 h (D72) after C2C12 cells were induced to differentiate and maintained relatively stable expression until 144 h (D144) after induction of differentiation (Supplementary Fig. 6e, f and Data 3). These napRNAs are mainly derived from intron, repetitive elements and intergenic (Supplementary Fig. 6g). To investigate whether changes in these napRNAs during development corresponded to changes in host genes, we quantified the expression of napRNA host genes using RNA-seq and found that the fold change of most napRNA expression during C2C12 differentiation showed weak correlation with their host genes (Supplementary Fig. 6h–j).

It is widely acknowledged that histone modifications play a crucial role in various chromatin-dependent processes, including transcription[38–40]. Regions exhibiting histone modifications are generally associated with high accessibility, facilitating transcription and gene expression. In our study, we conducted an analysis of the chromatin state of napRNAs to gain insights into their expression patterns and provide additional evidences to demonstrate that napRNAs are transcribed. Additionally, histone modifications such as H3K27Ac and H3K4me3 are commonly associated with the transcription activation of nearby genes[41]. Therefore, napRNAs marked with these two histone modifications are potential transcriptional enhancers (e.g., enhancer RNAs) bound by mediators, facilitating enhancer-promoter crosstalk[41]. To explore the chromatin state of these expressed napRNAs, we analyzed the distribution patterns of histone modifications on these napRNAs. We found that at least 7094 napRNAs were located within one region containing various histone modifications in humans (Supplementary Fig. 7a and Data 4). By hierarchical cluster analysis of histone modifications, we found that these napRNAs could be classified into 8 clusters (Supplementary Fig. 7b–i), which were further divided into two groups by the number

of histone species. One group consisted of napRNAs with one kind of histone modification (Clusters H3K27ac, H3K4me1, H3K79me2, H3K36me3, H3K27me3 and H3K9me3 in humans; Supplementary Fig. 7b–g), and the other group consisted of napRNAs with more than one kind of modification (Clusters H4K20me1 and H3K4me3 in humans; Supplementary Fig. 7h, i). Interestingly, most of these clusters in humans were enriched in histone modifications located within gene bodies, except for H3K4me3, which was abundant in the upstream regions of napRNAs (Supplementary Fig. 7i). Similar to the findings in humans, we identified 9289 napRNAs in mice (Supplementary Fig. 7j and Data 4) with enriched histone modifications located within gene bodies, a pattern that seems to be very noticeable (Supplementary Fig. 7k–o).

In summary, by developing the NAP-seq approach, we confidently uncovered thousands of structured napRNAs, some of which have evolutionarily conserved sequences and complex RNA secondary structures. Moreover, numerous napRNAs are differentially expressed in various cells or at various stages of skeletal muscle differentiation and are associated with various histone modifications.

**Repetitive elements were transcribed and processed into multiple groups of ncRNAs**

Given that NAP-seq can accurately identify the full-length sequences of known napRNAs (Fig. 1), we hypothesized that our NAP-seq method could be used to discover members of known noncapped ncRNA families, which are previously uncharacterized based on comparison to GENCODE datasets (human release 30 and mouse release 23)[25]. To test this hypothesis, we first developed computational pipelines to identify RNA Pol III-transcribed napRNAs (referred to as Pol3-napRNAs) from the NAP-seq data (Supplementary Fig. 8a). As expected, we detected 87 and 2215 previously undescribed Pol3-napRNAs with A/B box promoters and a 4U stretch at the 3′-end in humans and mice, respectively (Supplementary Fig. 8b and Supplementary Data 5). Notably, these Pol3-napRNAs were derived mainly from Alu repetitive elements (42.53%) in the human genome (Fig. 2a). Importantly, we revealed that other repetitive elements such as MIR (18.39%) and LINE1 (5.75%) elements, can be processed into Pol3-napRNAs in humans (Fig. 2a). These Pol3-napRNAs were predicted to fold into complex stem–loop structures (Fig. 2b and Supplementary Fig. 8c–e). Surprisingly, we detected more than 2200 Pol3-napRNAs in mice, almost all of which are derived from B2 repetitive elements (98.73%; Supplementary Fig. 8f and Data 5).

We next developed bioinformatic pipelines to identify previously undescribed candidates of snoRNAs, including C/D box and H/ACA box RNAs, from NAP-seq profiles (Supplementary Fig. 8a). We identified 226 and 71 previously undescribed C/D box napRNAs (CD-napRNAs) in humans and mice, respectively (Supplementary Data 5). Intriguingly, more than 20% of the CD-napRNAs were located within various known repetitive elements, such as Alu, LINE1 and B4 elements (Fig. 2c and Supplementary Fig. 9a). Unexpectedly, we found that the CD-napRNAs identified by NAP-seq were significantly longer than the known C/D box RNAs (Supplementary Fig. 9b), further indicating that our NAP-seq method can detect long napRNAs missed by traditional sRNA-seq. For instance, hsa-novel-CD-61, derived from Alu repetitive elements, was more than 600 nt long (Fig. 2d and Supplementary Fig. 9c); hsa-novel-CD-32, derived from the ERVL repetitive element (Supplementary Fig. 9d, e), and hsa-novel-CD-72, derived from the hAT repetitive element, were more than 500 nt long (Supplementary Fig. 9f, g). We further verified that randomly selected C/D box napRNAs indeed interact with the Fibrillarin (FBL) protein by using RIP-qPCR assay (Supplementary Fig. 9h).

In parallel, we detected 889 and 127 previously unknown H/ACA napRNAs (ACA-napRNAs) from NAP-seq profiles in humans and mice, respectively (Supplementary Data 5). Apparently, these ACA-napRNAs were mainly derived from Alu repetitive elements (59%) in the human genome (Fig. 2e) and from B2 elements (29.84%) in the

mouse genome (Supplementary Fig. 10a). Interestingly, some of them were transcribed and processed from other repetitive elements, such as LINE1, MIR and B1 elements (Fig. 2e and Supplementary Fig. 10a). Similar to the findings for CD-napRNAs, the ACA-napRNAs identified by NAP-seq were longer than the known H/ACA box RNAs (Supplementary Fig. 10b). For example, hsa-novel-ACA-232, derived from Alu elements, was 621 nt long and specifically expressed in U87 cells (Fig. 2f and Supplementary Fig. 10c), while hsa-novel-ACA-443 and hsa-novel-ACA-327, derived from MIR elements, were more than 300 nt long (Supplementary Fig. 10d–g). Surprisingly, our NAP-seq method has unveiled a previously undescribed group of Alu-ACA RNAs (Supplementary Fig. 10h) that display a distinct feature. Unlike the known Alu-ACA RNAs that cover ~25% of the entire length of the host Alu element[42], we have discovered that this class of Alu-ACA RNAs spans more than 50% of the length of the Alu element (Fig. 2g). Importantly, we further discovered 157 ACA-napRNAs with a common poly(A) pocket specifically located within the upstream hairpin, that was derived from the Alu A-rich elements (Fig. 2h and Supplementary Data 5). Moreover, RIP-qPCR assay showed that randomly selected H/ACA napRNAs could interacted with DKC1 protein (Supplementary Fig. 10i), which is like other classical H/ACA snoRNAs. Interestingly, only 9 napRNAs are conserved between humans and mice (Supplementary Fig. 10j). This conservation pattern observed aligns with the fact that most of these napRNAs originate from species-specific elements, such as Alu elements in humans and B1/B2/B3 elements in mice.

To explore the potential functions of previously uncharacterized napRNAs derived from repetitive elements, we selected a 767 nt long, structured napRNA (mmu-novel-CD-6) generated from the L2 element for experimental validation (Supplementary Fig. 11a, b). We confirmed its expression in C2C12 cells by using poly(T) RT–PCR (Supplementary Fig. 11c) and used Sanger sequencing to confirm its consensus 3′-end sequence with respect to that identified by our NAP-seq method (Supplementary Fig. 11d). We also detected the subcellular localization of mmu-novel-CD-6 by fluorescence in situ hybridization (FISH) and found that it was localized mainly in the nucleus (Supplementary Fig. 11e). Importantly, we found that mmu-novel-CD-6 showed a trend of downregulation during myoblast differentiation by using both our NAP-seq profiles (Supplementary Fig. 11a) and qRT–PCR analysis (Supplementary Fig. 11f), suggesting that mmu-novel-CD-6 may play a functional role during myoblast differentiation. To test this hypothesis, we overexpressed the wild-type mmu-novel-CD-6 in C2C12 cells, achieving the ectopic expression of which was to be ~300-fold. We observed that its overexpression had no influence on the mRNA level of its host gene (Exosc2) but significantly decreased the protein expression levels of the established myogenic markers MHC and Mef2c. We then introduced a mutation in the D box of mmu-novel-CD-6 (changing the CUGA motif to CUAG) and transfected this mutant into C2C12 cells. As a result, the mutant mmu-novel-CD-6 could not be properly processed, which did not affect the expression of Exosc2, and had no impact on myoblast differentiation (Supplementary Fig. 11g). Moreover, we conducted a knockdown of mmu-novel-CD-6 expression using antisense oligonucleotides (ASOs), which led to the upregulation of myogenic markers MHC and Mef2c (Supplementary Fig. 11h). These results suggest that mmu-novel-CD-6 plays a role in inhibiting myoblast differentiation.

Collectively, our results indicate that repetitive elements contribute greatly to de novo synthesis of a large group of previously undescribed napRNAs, that is repRNAs and could promote the expansion of some napRNA families, such as stably expressed Pol III-transcribed RNAs, C/D box RNAs and H/ACA box RNAs (Supplementary Fig. 5b). In addition, our NAP-seq method not only identified full-length napRNAs but also identified the longest subgroup of napRNAs discovered to date.

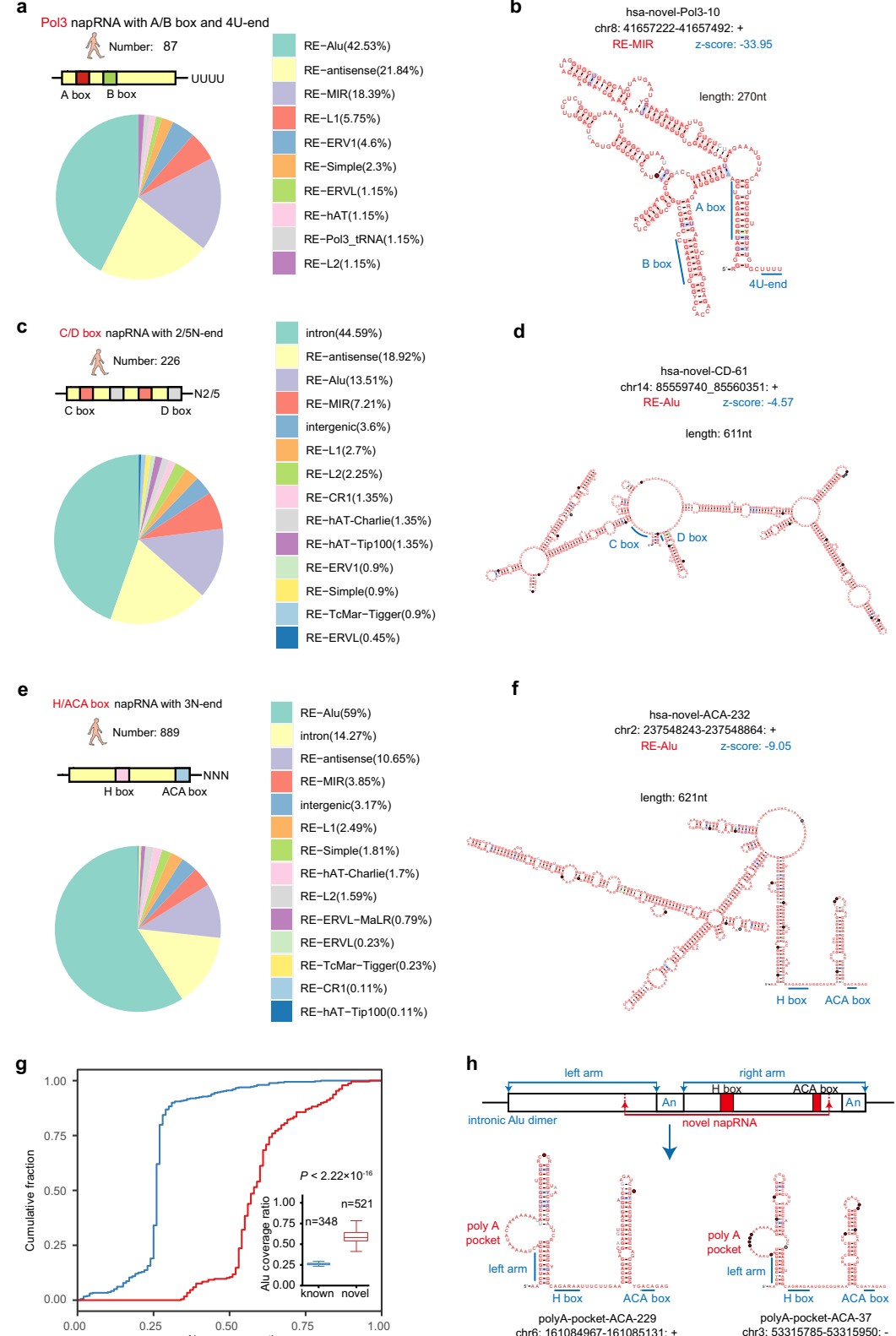

**a** Pol3 napRNA with A/B box and 4U-end

👤 Number: 87

- RE−Alu(42.53%)
- RE−antisense(21.84%)
- RE−MIR(18.39%)
- RE−L1(5.75%)
- RE−ERV1(4.6%)
- RE−Simple(2.3%)
- RE−ERVL(1.15%)
- RE−hAT(1.15%)
- RE−Pol3_tRNA(1.15%)
- RE−L2(1.15%)

**b** hsa-novel-Pol3-10
chr8: 41657222-41657492: +
RE-MIR          z-score: -33.95
length: 270nt

**c** C/D box napRNA with 2/5N-end

👤 Number: 226

- intron(44.59%)
- RE−antisense(18.92%)
- RE−Alu(13.51%)
- RE−MIR(7.21%)
- intergenic(3.6%)
- RE−L1(2.7%)
- RE−L2(2.25%)
- RE−CR1(1.35%)
- RE−hAT-Charlie(1.35%)
- RE−hAT-Tip100(1.35%)
- RE−ERV1(0.9%)
- RE−Simple(0.9%)
- RE−TcMar−Tigger(0.9%)
- RE−ERVL(0.45%)

**d** hsa-novel-CD-61
chr14: 85559740_85560351: +
RE-Alu          z-score: -4.57
length: 611nt

**e** H/ACA box napRNA with 3N-end

👤 Number: 889

- RE−Alu(59%)
- intron(14.27%)
- RE−antisense(10.65%)
- RE−MIR(3.85%)
- intergenic(3.17%)
- RE−L1(2.49%)
- RE−Simple(1.81%)
- RE−hAT-Charlie(1.7%)
- RE−L2(1.59%)
- RE−ERVL−MaLR(0.79%)
- RE−ERVL(0.23%)
- RE−TcMar−Tigger(0.23%)
- RE−CR1(0.11%)
- RE−hAT−Tip100(0.11%)

**f** hsa-novel-ACA-232
chr2: 237548243-237548864: +
RE-Alu          z-score: -9.05
length: 621nt

**g** $P < 2.22 \times 10^{-16}$
n=348    n=521

**h** left arm / right arm / intronic Alu dimer / novel napRNA

polyA-pocket-ACA-229
chr6: 161084967-161085131: +

polyA-pocket-ACA-37
chr3: 53315785-53315950: -

## A class of stable linear intron napRNAs (sliRNAs)

Although introns constitute ~25% of the mammalian genome[43], excised intron lariats are typically destined for rapid debranching and degradation and thus are generally viewed as byproducts of gene expression[44]. To explore the processing of introns, we analyzed our NAP-seq reads located within human introns and discovered that thousands of 5′-start and 3′-end sequencing reads were precisely located within the 5′ splice site (5′-SS) of 6448 introns or the 3′-SS of 17,397 introns in HepG2 cells (Fig. 3a, b), implying that these introns may be stable in linear form rather than rapidly degraded. To verify this hypothesis, we analyzed all NAP-seq-NGS and NAP-seq-TGS sequencing data with napSeeker software to identify the high-confidence napRNAs overlapping with entire intron regions. As a result, we identified 620 previously undescribed napRNAs with both a 5′-SS and 3′-SS

**Fig. 2 | Repetitive elements were transcribed and processed into multiple groups of ncRNAs. a** Distribution of the candidate Pol3-napRNAs in annotated gene types. Only the napRNAs located within intergenic, intronic, and repetitive elements were retained and are shown in the figure. **b** The highly stable RNA secondary structure of Pol3-napRNA. A negative z score indicated that the sequence was more stable than expected by chance. **c** Distribution of candidate CD-napRNAs in annotated gene types. **d** The highly stable RNA secondary structure of CD-napRNA. **e** Distribution of candidate human H/ACA-napRNAs in annotated gene types. **f** The highly stable RNA secondary structure of H/ACA-napRNA. **g** Cumulative curves and box plots showing the Alu coverage ratio of the previously undiscovered Alu H/ACA snoRNAs identified by NAP-seq ($n = 521$ examined over 17 independent experiments) and the known Alu H/ACA snoRNAs ($n = 348$ according to the public Alu ACA reference). $p$ value was calculated by two-sided Mann–Whitney–Wilcoxon test. Each boxplot shows the minima, maxima, center, bounds of box, whiskers, first and third percentile. **h** Schematic diagram of Alu RNA genes and the structure of H/ACA snoRNAs with a poly(A) pocket. The internal and terminal A-rich regions (An), processed sites, H/ACA box region, and poly(A) pocket region are shown.

in humans (Supplementary Fig. 12a and Data 6), suggesting that these intron napRNAs are linear and stable in vivo (Supplementary Fig. 5c); thus, we named them stable linear intron RNAs (sliRNAs). Moreover, we detected 916 sliRNAs in mice (Supplementary Fig. 12b and Data 6). Importantly, the 5′-start and 3′-end sequences of these sliRNAs could be accurately identified by NAP-seq but not by other classical RNA-seq methods (Supplementary Fig. 12c–f).

To preliminarily investigate the potential functions of sliRNAs in cellular biological processes, we first analyzed the conservation of sliRNAs and found that sliRNAs were more evolutionarily conserved than other introns (Supplementary Fig. 13a). Furthermore, we performed Gene Ontology (GO) enrichment analysis of the sliRNA host genes and revealed that these host genes are involved in a variety of biological processes, such as covalent chromatin modification, histone modification, ribonucleoprotein complex biogenesis and mRNA processing (Fig. 3c). We then explored the expression patterns of sliRNAs under various treatments and found that most sliRNAs in humans were downregulated after cells were exposed to stress (Supplementary Fig. 13b); these changes fell into four stress-response patterns (Supplementary Fig. 13c). For example, hsa-sliRNA-142, exhibiting pattern 1, was significantly downregulated during the cellular response to four kinds of stresses, especially $CoCl_2$ treatment (Supplementary Fig. 13d). In addition, we found that the expression of sliRNAs was dynamically changed during mouse myoblast differentiation; sliRNAs generally exhibited high expression at the late stage of myoblast differentiation (Supplementary Fig. 13e). In general, these significantly differentially expressed sliRNAs exhibited one of four patterns during development (Supplementary Fig. 13f). For example, mmu-sliRNA-138, exhibiting pattern 4, was continuously downregulated, implying its highly probable function of blocking myoblast differentiation (Supplementary Fig. 13g).

We next sought to reveal which genomic features determine the differences between sliRNAs and the other introns that undergo rapid degradation. Relative to the other introns, sliRNAs had similar canonical splicing motifs, the same lariat branch point residue (adenosine) (Supplementary Fig. 13h), and a similar distance (-25–45 nucleotides) between the lariat branch point and the 3′-SS (Supplementary Fig. 13i). Moreover, although numerous spliceosome proteins have the potential to bind to the 5′-SS and 3′-SS of sliRNAs, there was no difference of binding specificity between their binding to sliRNAs and their binding to other nearby introns located within the same protein-coding genes (Supplementary Fig. 13j). Notably, we discovered that the length distributions were significantly different between the other introns and sliRNAs, with sliRNAs tending to be shorter both in humans and mice. The length of these sliRNAs ranged mainly from 100 nt to 1000 nt, while the length distribution of the other introns covered a wider range (Fig. 3d). Importantly, we found that the sliRNAs exhibited significantly lower minimum free energies than the other introns (Fig. 3e and Supplementary Fig. 13k). For example, many sliRNAs could fold into various stem–loop structures and had a relatively low z score, implying that they have a stable secondary structure (Fig. 3f–i).

In summary, our NAP-seq method detected a previously undiscovered class of stable sliRNAs in humans and mice. According to the abovementioned comparative analysis, we hypothesized that short sliRNAs may fold more quickly into stable structures (Fig. 3e–i) and be incorporated into a ribonucleoprotein (RNP) complex, which could increase their stability and protect them from exonuclease degradation, while the longer introns rapidly degrade before they fold successfully (Fig. 3j, model).

### A class of snoRNA-intron napRNAs (snotrons)

Some introns excised from primary transcripts can be further processed to generate ncRNAs, such as snoRNAs[45]. Surprisingly, after careful analysis of the distances between the NAP-seq reads and the ends of known snoRNAs, we discovered a previously undescribed class of snoRNA-intron napRNAs (snotrons), in which one end corresponds to the 5′/3′-end position of an intronic snoRNA and the other end corresponds to the 3′/5′-end of an intron (Fig. 4a, b). In total, we identified 35 and 22 snotrons in humans and mice, respectively (Supplementary Data 7). For example, the single-nucleotide resolution maps of the NAP-seq-NGS and NAP-seq-TGS data precisely defined the 5′ and 3′ ends of the snotrons hsa-snotron-1 (Fig. 4c) and mmu-snotron-5 (Fig. 4d). Strikingly, traditional methods, such as poly(A) +RNA-seq, total RNA-seq and poly(A)− RNA-seq, could not identify these snotrons (Fig. 4c, d and Supplementary Fig. 14a), possibly because of the complex structure of snotrons which could be not read-through in reverse transcription (Fig. 4e, f and Supplementary Fig. 14b). Due to the complete snoRNA sequence within snotrons, we hypothesized that snotrons may interact with snoRNP proteins (e.g. FBL) to carry out their functions. In RIP-qPCR of FBL, we observed a significant enrichment of hsa-snotron-1 and hsa-snotron-3 with FBL (Supplementary Fig. 14c). This finding suggests a potential interaction between hsa-snotron-1, hsa-snotron-3, and FBL in HepG2 cells. Moreover, through evolutionary conservation analysis, we observed that the intron sequences of hsa-snotron-1 which contained SNORD2 were highly conserved in mammals (Supplementary Fig. 14d), implying that hsa-snotron-1 may play a crucial biological role as a whole snotron and not only as a snoRNA.

We further explored the expression patterns of snotrons in HepG2 cells and their RNA half-life (Supplementary Fig. 5d). The expression of snotrons are lower than that of their snoRNA counterparts that were detected by PEN-seq[11] (Supplementary Fig. 14e). The expression and the length of hsa-snotron-1 were verified by using irNorthern blotting (Fig. 4g). Interestingly, the majority of snotrons were downregulated in response to cellular stresses (Supplementary Fig. 14f) but generally showed two different expression trends (Supplementary Fig. 14g). For example, hsa-snotron-6 was significantly downregulated in response to four kinds of stresses but exhibited various stress-response levels (poly (I:C)>$CoCl_2$ > ADR, Supplementary Fig. 14h). Similarly, the expression levels of snotrons in mice underwent dynamic changes during C2C12 cell differentiation, decreasing rapidly after the early stage of myoblast differentiation (Supplementary Fig. 14i–k).

We next sought to reveal which genomic features determine the biogenesis of snotrons. We found that the distances between snoRNAs and the 3′ splice sites in the host introns were significantly shorter in snotrons than in the other snoRNAs (Fig. 4h).

Our NAP-seq method also detected previously undiscovered sno-lncRNAs (Supplementary Fig. 15a, b) and precisely redefined the

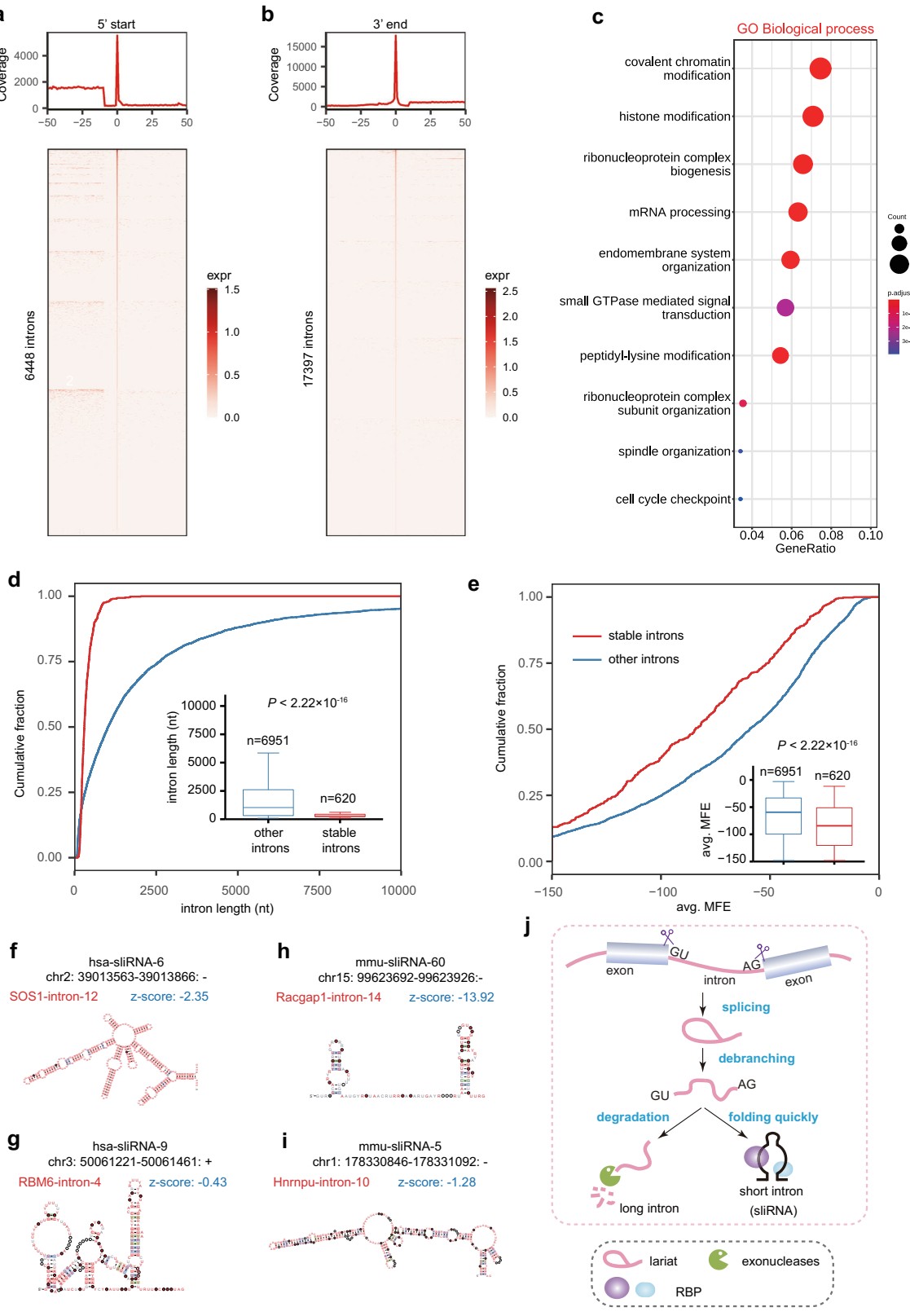

boundaries of an inaccurately annotated snoRNA (Supplementary Fig. 15c). In contrast to previously reported sno-end lncRNAs, the sno-lncRNA SNORD102-SNORA27 consisted of a C/D box snoRNA (U102) and an H/ACA box snoRNA (ACA27) at its termini (Supplementary Fig. 15a, b), which could not be identified by a previous poly(A)-depleted RNA-seq method (Supplementary Fig. 15a). Interestingly, the snoRNA reannotated by NAP-seq (napACA68, mmu-napRNA-9570;

Supplementary Fig. 15c) had a complex secondary structure composed of two C/D box elements at both termini and an internal H/ACA box element (Supplementary Fig. 15d). Importantly, the expression of the reannotated napACA68 increased rapidly from D0 to D72 and remained high thereafter (Supplementary Fig. 15c).

In summary, our NAP-seq method allowed us to discover a previously uncharacterized class of snotron napRNAs in humans and

**Fig. 3 | A class of sliRNAs in humans and mice was discovered by NAP-seq. a** The number of coincident reads between the 5′-start site in NAP-seq-NGS reads and the 5′ splice sites in known introns in HepG2 cells. The x-axis shows the distance from the 5′-start sites of NAP-seq-NGS read to the annotated 5′ splice sites, and the y-axis shows the number of reads within a certain distance. The bottom panel shows a heatmap, in which each row represents an intron that shares the same 5′-start site with the NAP-seq-NGS read, and each column represents the intron expression value at a specific distance. expr, expression value. **b** The number of coincident reads between the 3′-ends in NAP-seq-NGS reads and the 3′ splice sites in known introns in HepG2 cells. **c** Top 10 enriched GO biological processes enriched with the host genes of 620 previously undiscovered sliRNAs identified in humans. p values were calculated by hypergeometric test. **d** Cumulative curves and box plots showing the lengths of sliRNAs identified by NAP-seq (n = 620 examined over 17

independent experiments) and other introns (n = 6951 according to the GENCODE), respectively. **e** Cumulative curves and box plots showing the average minimum free energy (avg. MFE) of sliRNAs and other introns with a length ≤500 nt in humans. p values in (**d**) and (**e**) were calculated by two-sided Mann–Whitney–Wilcoxon test. Each boxplot shows the minima, maxima, center, bounds of box, whiskers, first and third percentile. Examples of sliRNAs with highly stable secondary structures in humans (**f, g**) and mice (**h, i**). A negative z score indicated that a sequence was more stable than expected by chance. **j** The proposed model of sliRNA biogenesis. An intron lariat is spliced from the host gene and debranched to generate a linear intron. Then, the short linear intron may fold quickly and bind to RBPs (RNA binding proteins) to avoid degradation, contributing to its stability as a sliRNA in vivo. In contrast, long introns are degraded quickly by exonuclease digestion.

mice, as well as previously undescribed sno-lncRNAs. Based on our above analysis, we propose a possible model of snotron biogenesis (Fig. 4i, model): snoRNAs located within a short distance from the 3′-SS may facilitate interactions of snoRNPs with spliceosomal proteins that would prevent the degradation of the 3′-ends of introns, while snoRNAs located within a long distance from the 3′-SS prohibit the interaction between snoRNPs and spliceosomal proteins (Fig. 4i, model).

## A class of napRNAs processed from miRNA clusters (misRNAs)
Most pre-miRNAs are cotranscriptionally excised and processed into mature miRNAs by the microprocessor complex containing Drosha and DGCR8 and by the endoribonuclease Dicer[46,47]. Unexpectedly, we found that the previously uncharacterized long napRNAs could intersect more than one pre-miRNA, implying the existence of previously undescribed types of napRNAs that are related to miRNAs (Fig. 5a). By analyzing the boundaries of these napRNAs, we surprisingly found that these linear napRNAs mapped to the sequences of miRNA spacer regions and that each end of these napRNAs coincided with one end of the two mature miRNAs (Fig. 5b and Supplementary Fig. 16a, b). Thus, we named this class of napRNAs "misRNAs" (miRNA spacer RNAs). We identified a total of 26 and 29 misRNAs in humans and mice, respectively (Supplementary Data 8). These misRNAs had various secondary structures, and some had stable expression (Supplementary Fig. 5e) and conserved secondary structures (Fig. 5c, d and Supplementary Fig. 16c, d). Moreover, we verified the exact 3′ end sites of several randomly selected misRNAs expressed in humans and mice (Supplementary Figs. 16a, b and 17a–c) by using poly(T) RT–PCR coupled with Sanger sequencing (Supplementary Fig. 17d, e). Notably, we observed that the expression levels of misRNAs dynamically changed in response to cellular stresses in humans (Fig. 5e and Supplementary Fig. 18a, b) and during myogenesis in mice (Fig. 5f and Supplementary Fig. 18c, d).

Interestingly, some misRNAs even spanned multiple pre-miRNAs embedded in these RNA clusters (Fig. 5g). We speculated that this phenomenon may be explained by the stepwise processing of polycistronic miRNAs. For example, hsa-misRNA-2 was first processed from the miRNA cluster located at chr13:91,350,604-91,350,972, which contains three pre-miRNAs, and subsequently generated two secondary spacers (between miR-17-3p and miR-18a-5p; between miR-18a-3p and miR-19a-5p) that were further cleaved by the Drosha protein (Fig. 5h).

To explore the mechanism of misRNA processing, we integrated the RBP CLIP-seq data and NAP-seq data for further analysis. We identified a number of RBPs, including members of the microprocessor complex, which mediates miRNA processing, that potentially interact with misRNA termini, among which AGO, DGCR8 and Drosha showed the highest percentages (Supplementary Fig. 18e). This is consistent with previous reports indicating that pre-miRNAs are cleaved from pri-miRNAs by a microprocessor complex consisting of Drosha and DGCR8[47]. Once the spacer region between the two pre-

miRNAs was separated, it was folded into a structural misRNA to prevent further degradation (Fig. 5i, model).

## The napRNA DINAP interacts with DKC1 to promote cell proliferation by maintaining protein stability
To illuminate the biological roles of previously undescribed napRNAs, we focused our further investigation on the highly expressed napRNAs that were evolutionarily conserved and interacted with RBPs. Surprisingly, we found that among the top 100 highly expressed napRNAs, one napRNA was the most conserved (Fig. 6a and Supplementary Fig. 19a) and is composed of a box C/D and a box H/ACA snoRNA-like domains (Supplementary Fig. 19b, c), suggesting that DINAP is a composite box C/D-H/ACA RNA like U85 and U87[48] SCARNAs. By analyzing CLIP-seq data, DINAP was interacted with the DKC1 protein (Fig. 6b). We thus named this previously undescribed napRNA (hsa-napRNA-13) as DINAP (DKC1-interacting napRNA). Furthermore, we confirmed the expression and 3′ end sequence of DINAP by carrying out poly(T) RT–PCR coupled with Sanger sequencing in HepG2, HEK293T and U87 cells (Supplementary Fig. 19d, e).

To validate the interaction between DINAP and DKC1, we first determined the subcellular localization of DINAP and DKC1 by separating the nuclear and cytoplasmic fractions (Fig. 6c). We found that both DINAP and the DKC1 protein were localized mainly in the nucleus (Fig. 6c) with GAPDH, FBL and β-Tubulin as subcellular markers (Fig. 6c, right). In addition, we further showed that DINAP was colocalized with DKC1 in the nucleoplasm by using fluorescence in situ hybridization (FISH) (Fig. 6d). We next performed RNA immunoprecipitation targeting DKC1 to evaluate the binding of DINAP to DKC1 in wild-type HepG2 cells (Fig. 6e). As expected, significant enrichment of DINAP was observed in DKC1-bound RNA (Fig. 6e, right).

Given that DINAP interacts with DKC1, we assume that DINAP may regulate the protein stability of DKC1. DINAP was overexpressed in HepG2 cells (Supplementary Fig. 19f), and we then examined the turnover of endogenous DKC1 protein using a cycloheximide (CHX) chase assay[49]. Ectopic expression of DINAP increased the protein level of DKC1 but did not affect its mRNA level (Fig. 6f and Supplementary Fig. 19g). Moreover, we generated DINAP-knockout (KO) cells by deleting the DINAP gene using CRISPR/Cas9 genome editing (Supplementary Fig. 19h). The accuracy of DINAP gene knockout was verified by PCR amplification and Sanger sequencing (Supplementary Fig. 19i, j). The expression of DINAP in KO cells was obviously decreased compared to that in wild-type (WT) HepG2 cells (Supplementary Fig. 19k). In addition, efficient accumulation of the DINAP host gene AC090912.1 in DINAP-KO cells was verified by qPCR (Supplementary Fig. 19l). We then performed western blotting analysis and observed that the protein level of DKC1 was decreased in DINAP-KO cells compared to WT cells (Fig. 6f). To exclude the possibility that the decrease in DKC1 protein expression in DINAP-KO cells was due to undesired off-target activity of CRISPR/Cas9, we rescued WT DINAP expression in KO cells (Supplementary Fig. 19m). As expected, reconstitution of DINAP expression restored the protein level of DKC1

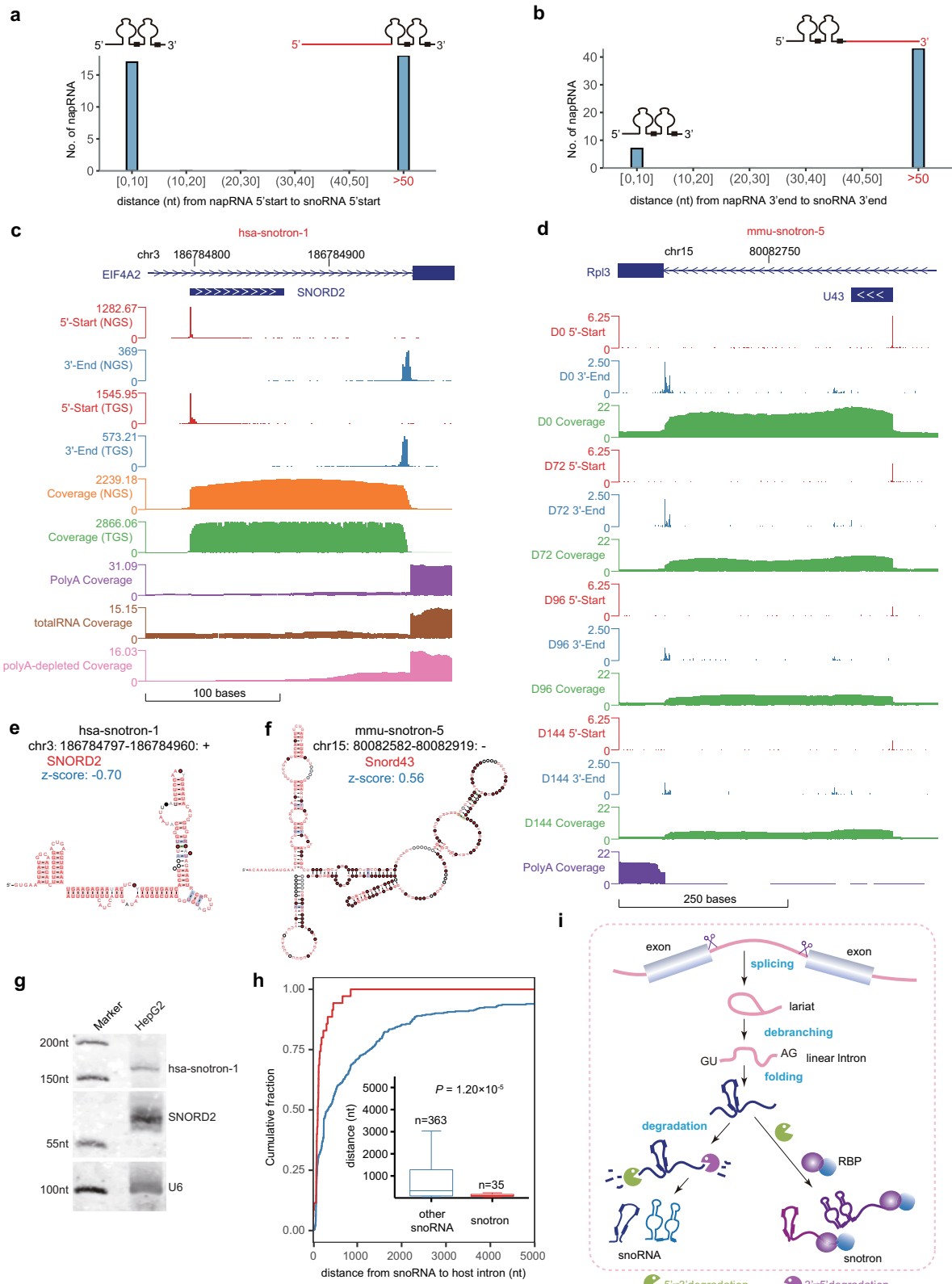

(Fig. 6f). These results revealed that DINAP regulates the stability of the DKC1 protein.

To investigate whether the interaction between DINAP and DKC1 has important roles in tumorigenesis, we measured cell proliferation in DINAP-overexpressing HepG2 cells and control cells by CCK-8 and colony formation assays. We observed that ectopic expression of DINAP significantly increased cell

proliferation (Fig. 6g) and colony formation (Fig. 6h) in HepG2 cells. Moreover, knockout of DINAP in HepG2 cells significantly inhibited their proliferation and reduced their colony formation ability (Supplementary Fig. 20a, b), consistent with the phenotypes resulting from knockdown of DKC1 (Supplementary Fig. 20c–e). The decreases in the proliferation and colony formation abilities of DINAP-KO cells were reversed by forced

**Fig. 4 | A class of snoRNA-intron napRNAs (snotrons).** Statistics of the distances from napRNAs to annotated snoRNAs: start sites (**a**) and end sites (**b**). The *x*-axis shows the distance (nt) from napRNA 5'-start (or 3'-end) sites to snoRNA 5'-start (or 3'-end) sites, and the *y*-axis shows the number of napRNAs within the specified distance range. Genome Browser view of 5'-start, 3'-end and coverage signals (RPM, reads per million) in an extended region of two snotrons: hsa-snotron-1 (**c**) and mmu-snotron-5 (**d**). The highly stable secondary structures of snotron hsa-snotron-1 (**e**) and mmu-snotron-5 (**f**). **g** Verification of hsa-snotron-1 by irNorthern blotting in HepG2 cells. Source data are provided as a Source Data file. **h** Cumulative curves and box plots showing the distance from the host 3'-SS to the snoRNA 3'-end sites (*n* = 35 examined over 17 independent experiments) identified by NAP-seq and to the other snoRNA 3'-end sites (*n* = 363 according to the snoRNAbase). *p* value was calculated by two-sided Mann–Whitney–Wilcoxon test. Each boxplot shows the minima, maxima, center, bounds of box, whiskers, first and third percentile. **i** The proposed model of snotron biogenesis. An intron lariat is spliced from the host gene and is then debranched to generate a linear intron. Next, the RNA splicing intermediate is processed into a C/D box snoRNA or an H/ACA box snoRNA after the lariat intron is debranched (left path). Because the 3'-ends of shorter linear introns are formed into complex structures quickly and bind to RBPs, making the exonuclease unable to cut them, these intronic RNAs are stable as snotrons (right path). In addition, cellular stress might suppress snotron formation.

expression of wild-type DINAP (Supplementary Fig. 20f, g). Ectopic expression of DKC1 also restored the proliferation ability of DINAP-KO cells (Fig. 6i, j and Supplementary Fig. 20h), further supporting the hypothesis that DINAP promotes cell proliferation by stabilizing the DKC1 protein in HepG2 cells (Supplementary Fig. 20i, model).

## Discussion

In this study, we initially developed the NAP-seq method to identify full-length napRNAs, making it possible to uncover an unprecedented noncapped RNA landscape. Systematic comparison with traditional sRNA-seq and RNA-seq methods demonstrated that NAP-seq extensively and accurately detected previously unidentified structured napRNAs and multiple classes of stably expressed napRNAs in diverse genomic locations that are dynamically expressed in various cellular environments. In addition, we demonstrated the potential biogenesis mechanisms of these previously uncharacterized classes of napRNAs. Importantly, we discovered that a long and structured napRNA may affect myoblast differentiation and that the napRNA DINAP associates with the DKC1 protein to promote cell proliferation by maintaining DKC1 protein stability.

NAP-seq was developed to identify full-length napRNAs of various lengths and diverse terminal modifications by improving adapter ligation and strand-specific and quantitative sequencing and avoiding mispriming artefacts, rRNA contamination and the difficulties imposed by RNA modifications and stable RNA secondary structures during sequencing library construction. It shows major advantages, as follows: (1) In this study, we utilized specially designed RNA adapters to capture both ends of napRNAs. This strategy allowed us to identify the full-length sequences of napRNAs and facilitated the discovery of napRNAs with characteristic motifs and structural elements at RNA ends. (2) The use of T4 PNK and SuperScript IV RT enzymes enabled the identification of previously undetected napRNAs with various RNA modifications and highly stable structures (Fig. 1). (3) By using the RNase H method to remove high-abundance rRNAs, snRNAs and other ncRNAs, our NAP-seq method can discover and enrich low-expression napRNAs. (4) Because it uses a nested RT primer, our NAP-seq method can amplify the full-length cDNA sequence of long napRNAs without amplifying misprimed RNAs, which has previously resulted in uncorrected results and conclusions. (5) The pre-size selection procedure (>100 nt) enabled the discovery of thousands of long napRNAs that have been missed by traditional sRNA-seq. For example, NAP-seq identified a previously undescribed subgroup of known napRNA families (C/D box RNAs, H/ACA box RNAs, Pol III-transcribed RNAs) with the longest sequence length found to date. (6) Nanopore sequencing is a unique, scalable technology that enables direct, real-time analysis of long DNA or RNA molecules[50,51], including DNA/RNA modifications[52,53] and RNA structures[54]. Next-generation sequencing (NGS, e.g., Illumina) shows high sequencing accuracy. NAP-seq-TGS and NAP-seq-NGS could be independently used to napRNAs research, except for the modifications on napRNAs. It is precisely due to the innovative approach of NAP-seq mentioned above that a large number of previously undiscovered long napRNAs have been identified. In contrast, methods like TGIRT-seq utilize NGS, which has the limitation of short read lengths (such as paired-end 150 nucleotides) and cannot accurately determine the full-length sequences of long RNA molecules with complex structures[16–18].

Known functional RNA molecules demonstrate diverse half-lives. For example, current research on enhancer RNA (eRNA), which activates or enhances transcription, has shown that the half-life of eRNA is approximately ≤7.5 min[55]. Furthermore, many regulatory long non-coding RNAs (lncRNAs) have a half-life of less than 30 min. For instance, the well-characterized paraspeckle RNA NEAT1 has a half-life of less than 30 min based on array data and ~15 min according to qPCR[56]. The half-lives of many napRNAs are longer than those of eRNAs and lncRNAs, implying the potential functional roles of napRNAs (Supplementary Fig. 5). On the other hand, some of the identified napRNA species, characterized by low abundance, poor conservation, and overlapping interspersed repeated elements, could simply represent degradation products or RNA intermediates during pre-mRNA splicing or pri-miRNA processing. It is important and exciting to demonstrate the physiological or pathological significance of these napRNA species in the future.

Although repetitive elements may constitute over 50% of the mammalian genome[57,58], it was previously thought that most repeat sequences in the genome are kept in the transcriptionally repressed state through epigenetic regulation[59]. Surprisingly, we uncovered thousands of repetitive elements that were transcribed and processed into previously undescribed subgroups of structured napRNAs, including C/D box, H/ACA box and Pol III-transcribed RNAs (Fig. 2). Moreover, these napRNAs were derived from various repetitive elements, such as SINE (Alu), LINE1 (L1), LINE2 (L2), ERV and MIR elements, suggesting that almost all repetitive elements might contribute to the de novo origin of napRNAs. Interestingly, the majority of human-specific H/ACA napRNAs and mouse-specific Pol3-transcribed napRNAs were derived from specific Alu and B2 repetitive elements, suggesting that duplication and mutation events of a single type of repetitive element promoted the expansion of these napRNA genes, which may perform biological functions. One of the most intriguing aspects of napRNAs is that B2 RNA regulates target gene expression to correspond to stimulus, such as heat shock[60] and the increased amyloid-beta load in amyloid pathology[61]. Furthermore, B2 RNA is found to be self-cleaving ribozymes whose activity is enhanced by EZH2[62]. Given canonical snoRNAs that guide proteins to introduce site-specific 2'-O-Me and pseudouridine modifications in rRNAs and snRNAs, which are essential for the functional fidelity of ribosomes and gene expression[63], we further explored the possibility of potential interactions between napRNAs and rRNA/U-snRNAs using snoSeeker software. However, we did not find strong evidence of extensive base-pairing interactions between snoRNA-like napRNAs and known rRNAs/snRNAs with RNA modification sites. Significantly, we did not discover any genes exhibiting significant proximity interactions with mmu-novel-CD-6, which implies that mmu-novel-CD-6 might not serve as a snoRNA guiding 2'-O-methylation during myoblast differentiation. Consequently, we speculate that these orphan napRNAs may target other types of RNAs, serving distinct biological functions. Moreover,

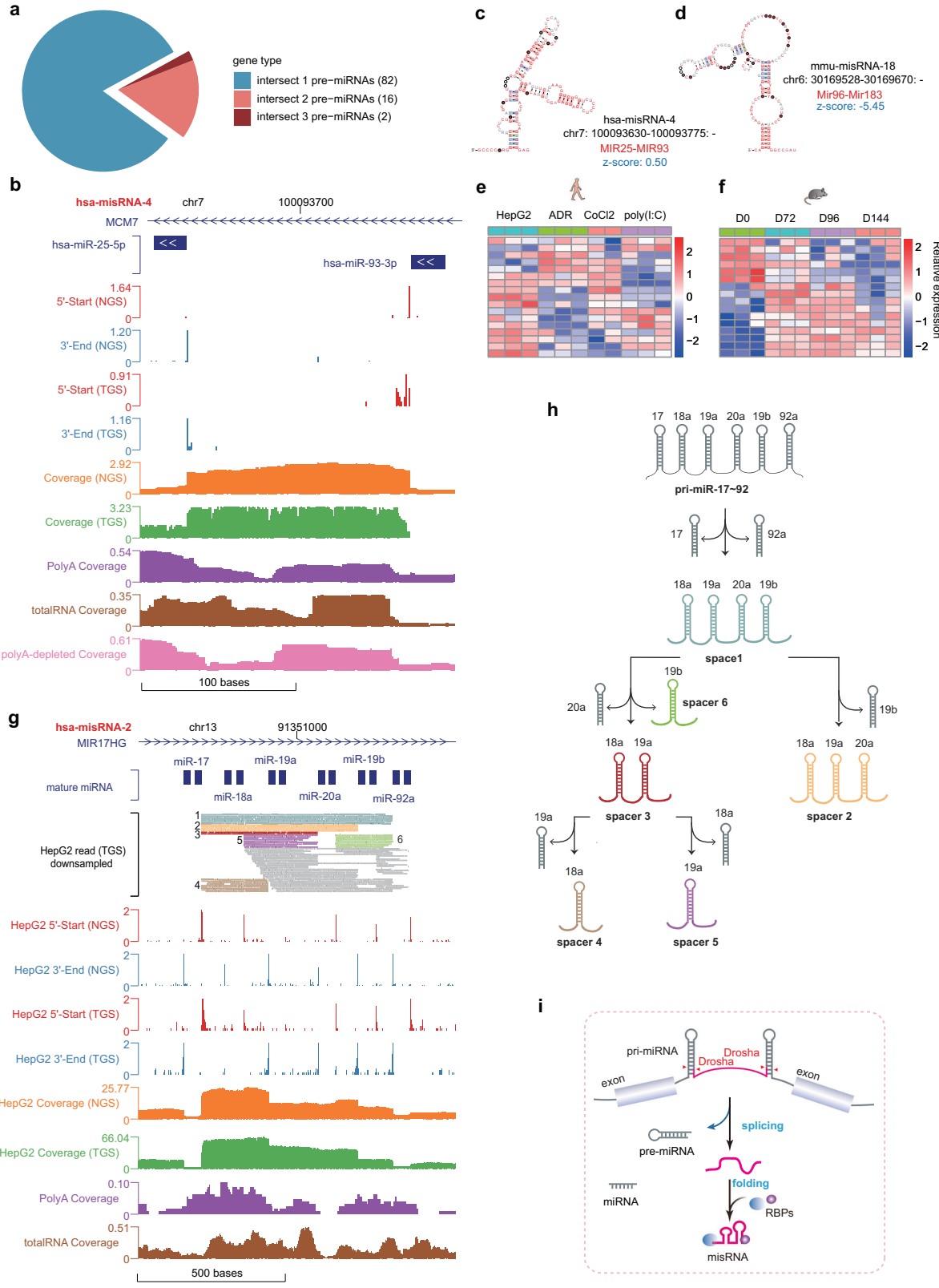

we discovered that the expression of many napRNAs derived from repetitive elements is dynamically changed under different cellular conditions, and we showed, as one example, that these napRNAs affect myoblast differentiation, strongly demonstrating that these napRNAs may regulate gene expression by yet-unknown mechanisms (Supplementary Fig. 11). Therefore, the discovery of a large number of repetitive element-derived napRNAs by the NAP-seq method not only

expands the landscape of these important functional RNA molecules but also strongly supports the important point that ignoring this landscape of RNAs derived from repetitive elements is like walking blindfolded into a beautiful wilderness[15].

Spliceosomal introns are an ancient and general feature found across all eukaryotic life. However, almost all introns are destined for rapid debranching and degradation, although some introns are

**Fig. 5 | A class of napRNAs embedded in miRNA spacer regions (misRNAs).**
**a** Statistics of the number of pre-miRNAs that intersected with the napRNAs.
**b** Genome Browser view of the full-length misRNAs (RPM, reads per million) in an extended region of misRNA hsa-misRNA-4. The highly stable secondary structures of misRNAs hsa-misRNA-4 (**c**) and mmu-misRNA-18 (**d**). **e, f** The dynamic changes in misRNAs under different conditions or in different stages. Heatmaps showing the differentially expressed misRNAs in HepG2 cells under four stress-response conditions (**e**) and in C2C12 cells at four developmental stages (**f**). Each row represents a differentially expressed misRNA, and each column represents a treatment type. The color, ranging from blue to red, represents the relative expression value from low to high, respectively. **g** Genome Browser view of the processing of misRNAs in the miRNA-17-92 cluster. Genome Browser view of 5′-start, 3′-end and coverage signals (RPM, reads per million) in an extended region of the miRNA-17-92 cluster. **h** Diagram of miRNA processing from the miRNA cluster. The miRNA cluster pri-miR-17-92 simultaneously yielded five miRNA spacers. These miRNA spacers are shown in various colors based on the spacer type, which are the same as those for the TGS reads in (**g**). **i** The proposed model of misRNA biogenesis. During the miRNA maturation process in humans and mice, the spacer region between the two pre-miRNAs is separated and is then folded into a structural misRNA and bound by a microprocessor complex to prevent further degradation.

processed into functional ncRNAs, such as miRNAs and snoRNAs[64]. Although intron lariats and circular introns have previously been reported in many species[65–68], excised linear introns have been identified only in yeast[69,70]. In this study, we discovered hundreds of stable linear and full-length introns (sliRNAs) in humans and mice by the NAP-seq method. Moreover, the expression of these sliRNAs is dynamically changed during mouse myoblast differentiation and in response to environmental changes, implying that they may act as regulatory napRNAs, as described in yeast[69,70]. However, unlike the excised linear introns in yeast[69], these sliRNAs do not feature a short distance between their lariat branch point and 3′ splice site (Supplementary Fig. 13i). Intriguingly, these sliRNAs are associated with components of the spliceosome and differ from other introns in that they have a short length and significantly lower minimum free energy (Fig. 3d, e), which may be necessary and sufficient for their stabilization.

A previous study reported a type of snoRNA-containing napRNA, sno-lncRNA, which is widely expressed in cells and tissues and closely associated with diseases[71,72]. In this study, we discovered a previously undescribed class of snoRNA-related ncRNAs (snotrons), as well as many known and previously undiscovered sno-lncRNAs, by our NAP-seq method (Fig. 4). Although snotron bearing a snoRNA sequence at the 5′ or 3′ end is derived from intron, snotron is different from slb-snoRNA that are circular lariat (sisRNA) bearing snoRNA[73]. Unlike the 3′ snoRNA end of sno-lncRNAs[71] and hmsnoRNA[74], the 3′-end site of snotrons is located at the 3′-SS, which is probably the abnormality of the 3′/5′-end of the intron for the further processing. Interestingly, although it has been reported that the efficient processing of C/D box-type snoRNP often requires that the snoRNA gene be positioned near the 3′-SS[75], we found that the distance between the snoRNA in snotrons and the 3′-SS is shorter than that between other snoRNAs and the 3′-SS (Fig. 4h). However, it would be interesting to confirm whether these characteristics are mechanistically connected to the processing efficiency or stability of snotrons. Moreover, although the majority of snotrons are derived from C/D box snoRNAs, H/ACA box snoRNAs can also generate snotrons, further highlighting the generality of this previously unreported class of napRNAs, snotrons.

More than one-fourth of miRNAs are organized as miRNA clusters in the same transcript[76]. These miRNA clusters are processed into mature miRNAs and intermediates. However, the fate and function of miRNA processing intermediates remain unknown. In this study, we discovered by the NAP-seq method that many miRNA clusters produce miRNA spacer sequences (misRNAs) in various cells. Moreover, each end of these misRNAs coincided with one end of the two corresponding mature miRNAs, suggesting that they are processed by the Drosha protein (Fig. 5b and Supplementary Fig. 16a, b). Our analysis also showed that microprocessor complex components (e.g., DGCR8 and Drosha) are significantly primarily located at each end of misRNAs, which might enhance the stability of these misRNAs by inhibiting their exonucleolytic degradation (Supplementary Fig. 18e). Interestingly, by analyzing these misRNAs, we can discover the precise processing procedures of various miRNA clusters, such as the miR-17-92 oncogenic cluster (Fig. 5g, h)[77]. Importantly, the dynamic changes in the expression of misRNAs in response to different cellular conditions (Fig. 5e and Supplementary Fig. 18a) and the dysregulated accumulation of miRNA intermediates during tumorigenesis[77] showed that these stably expressed misRNAs identified in this study may play important regulatory roles.

In summary, our method achieves full-length sequencing of napRNAs via several experimental strategies and by combining NGS and TGS platforms, and opens avenues to discover previously uncharacterized classes of napRNAs. Although we confirmed the functions of two napRNAs, the large number of sliRNAs, snotrons, misRNAs, previously undescribed structured napRNAs and repRNAs represent a pivotal continent in the modern "RNA world".

## Methods
### Cell culture and RNA extraction
All cell lines used for NAP-seq library preparation were purchased from the Shanghai Institute of Cell Biology, Chinese Academy of Sciences, and cultured according to the instructions. Cells obtained from the Cell Bank of the Chinese Academy of Sciences relied on the company's certificates of analysis. HEK293T (SCSP-502) cells were cultured in Dulbecco's modified Eagle's medium (DMEM, Gibco, C11995500BT) supplemented with 10% fetal bovine serum (FBS), 100 units/ml penicillin and 100 µg/ml streptomycin, while both HepG2 (SCSP-510) and U87 (TCHu138) cells were cultured in MEM (Gibco) supplemented with 10% FBS, 100 units/ml penicillin and 100 µg/ml streptomycin. To assay protein stability, HepG2 cells were treated with 0.05 mg/ml cycloheximide (MdBio, 66-81-9) for 15 min.

For stress stimulation, poly (I:C) (InvivoGen, tlrl-pic) was transfected into HepG2 cells with Lipofectamine 2000 (Thermo Fisher, 11668019) at a final concentration of 2 µg/ml. Cells were harvested for RNA extraction after induction of responses by poly (I:C) for 12 h. ADR (Sigma-Aldrich, D1515) and $CoCl_2$ (Sigma-Aldrich, C8661-100MG) were added directly to the medium (500 ng/ml and 500 µM, respectively), and HepG2 cells were harvested to analyze $ADR/CoCl_2$-induced responses 24 h after exposure.

When C2C12 (SCSP-505) cells were ~80% confluent, the medium was changed to differentiation medium, which consisted of DMEM supplemented with 2% horse serum (GE HyClone, SH30074). Total protein and RNA were harvested 0, 72, 96 and 144 h after the cells were induced to differentiate.

The cells were free of mycoplasma contamination based on the MycoBlue Mycoplasma Detector (Vazyme, D101).

Total RNA was isolated using RNAzol[78]. For RNA integrity, the RNA quality number (RQN) was determined by Qsep1 (BiOptic) before library construction.

### NAP-seq library construction
NAP-seq libraries were constructed with the following steps: end repair by T4 PNK, 3′/5′ adapter ligation, rRNA and specific gene removal, nested RT-PCR, and library construction for TGS or NGS.

**End repair and 3′/5′ adapter ligation.** RNA was mixed with 50 µl of 10 × T4 PNK buffer (5 µl), 10 mM ATP (5 µl), 10 U/µl T4 PNK (2 µl, NEB, M0201), 40 U/µl Ribolock RNase inhibitor (RRI; 1 µl, Thermo Fisher,

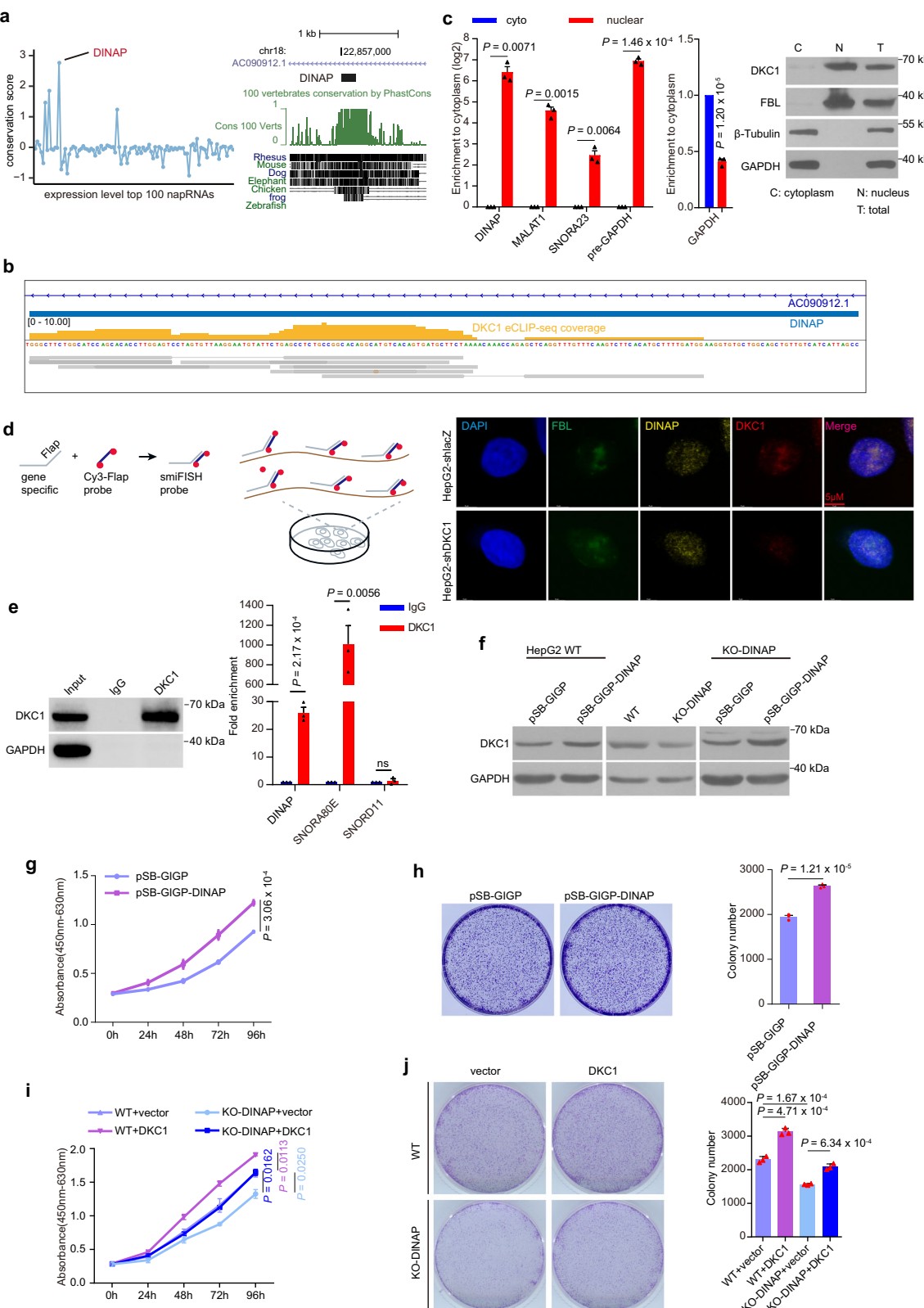

EO0384) and nuclease-free water (37 μl, Thermo Fisher, AM9938) and incubated at 37 °C for 1 h. T4 PNK-treated RNA (2 μg) was ligated to 3' adapters directly in the ligation mixture (2 μl of 10 × T4 RNA ligase reaction buffer, 5 μl of 50% PEG 8000 MW, 1 μl of 200 U/μl T4 RNA Ligase 2, truncated KQand 1 μl of RRI) at 16 °C for 18 h. To remove the unligated 3' adapters, we added (1) 2 μl of 50 U/μl 5' Deadenylase (NEB, M0331), with incubation for 1 h at 30 °C; (2) 0.44 μl of 30 U/μl *E. coli* single-stranded binding protein (Promega, M3011), with incubation on ice for 30 min; and (3) 2 μl of 30 U/μl RecJf exonuclease (NEB, M0246), with incubation at 37 °C for 1 h. Then, 5' adapter ligation buffer (2 μl of 10 × T4 RNA ligase reaction buffer, 4 μl of 10 mM adenosine-5'-tripho-sphate, 1 μl of 10 U/μl T4 RNA ligase 1 (NEB, M0204) and 1 μl of RRI) was added to the mixture above and reacted at 16 °C for 18 h to obtain the best ligation efficiency.

**Fig. 6 | The napRNA DINAP interacts with DKC1 to promote cell proliferation by maintaining protein stability. a** Evolutionary conservation analysis of the top 100 napRNAs (left) and the evolutionary conservation of DINAP in 100 vertebrates (right, conservation score calculated by PhyloP). **b** IGV plot showing the signal density of DKC1 binding in the region surrounding DINAP in HepG2 eCLIP data (ENCFF252TQP), Yellow columns represent the coverage of eCLIP enriched reads. **c** qPCR quantification (left) and western blotting analysis (right) of the nuclear and cytoplasmic distribution of DINAP and DKC1 in HepG2 cells. The subcellular distribution of MALAT1, SNORA23, pre-GAPDH and GAPDH RNA was assessed by qPCR. FBL, β-Tubulin and GAPDH in western blotting served as the nuclear and cytoplasmic markers, respectively. These genes were used as the positive controls. Protein loading in the western blotting was quantified by Coomassie staining. **d** In HepG2 WT cells and DKC1 knock-down cells respectively, subcellular localization of DINAP (red) is assessed by smiFISH (the principle of smiFISH is shown on the top), and of DKC1, as assessed by immunofluorescence staining. Nuclear DNA was stained with DAPI. All images are representative of two biological replicates. **e** The interaction between DINAP and the DKC1 protein was confirmed by RIP-qPCR in lysates of wild-type HepG2 cells. **f** WB analysis of DKC1 protein levels in DINAP-overexpressing cells, KO-DINAP cells and KO-DINAP cells with restoration of DINAP expression treated with 0.05 mg/ml CHX. **g** The promotion of cell proliferation in DINAP-overexpressing cells and control cells was examined by a CCK-8 assay. **h** Effect of DINAP overexpression on the colony formation ability. Colonies were counted by ImageJ. **i** CCK-8 assays showing the effect of DKC1 on restoring cell proliferation in KO-DINAP cells. **j** Colony formation assay in wild-type HepG2 cells and KO-DINAP HepG2 cells with ectopic expression of DKC1. Representative images of crystal violet staining of cells in (**h**) and (**j**) are shown beside the histograms of colony numbers. Colonies were counted in three replicates. The values in (**c**, **e**, **g**–**j**) are mean ± SEM of three independent experiments. Two-sided t-test. Source data are provided as a Source Data file.

**Removal of high-abundance rRNAs and ncRNAs by the RNase H method.** To increase the diversity of napRNA species, we removed rRNAs and some known ncRNAs of high abundance with specific DNA probes according to previous study[11]. We designed DNA probes targeting 45S (hg38), 18S (hg38), 5.8S (hg38), SNORA73A/B (hg38), RMRP (hg38) and RPPH1 (hg38) in humans and mice. The length of each oligo is about 45–59 nt length. After annealing, the DNA/RNA hybrids were cleaved with 5 U/µl RNase H (4 µl, NEB, M0523). Then, the DNA probes in the reaction were digested via the combination of 30 U/µl RecJf (4 µl) with 2 µl of 30 U/µl *E. coli* single-stranded binding protein to avoid affecting the subsequent RT-PCR.

**Nested RT-PCR strategy.** In the modified RT-PCR strategy, 20 µM nested primers were first mixed with adapter-ligated RNA and 1 µl of 10 mM dNTPs (NEB, N0446) for denaturation at 65 °C for 5 min followed immediately by incubation in an ice bath. Then, 4 µl of 5 × Superscript IV (SSIV) reaction buffer, 1 µl of 100 mM DTT and 1 µl of 200 U/µl SSIV reverse transcriptase (Thermo Fisher, 18090200) were added, and a thermal cycling program of 60 °C for 1 h and 80 °C for 10 min was performed for SSIV inactivation. The nested primer was designed as described previously[21]. cDNA was purified by oligo Clean & Concentrator (ZYMO research, D4060) and eluted in 20 µl nuclease-free water for library construction.

**MinION direct DNA sequencing (TGS).** One half of the NAP-cDNA (10 µl cDNA obtained in the nested RT-PCR step) was amplified by KAPA HiFi HotStart ReadyMix (Roche, KR0370) for MinION nanopore sequencing. PCR products were quantitated with a Qubit fluorometer (Thermo Qubit 2.0) and an Equalbit® 1 × dsDNA HS Assay Kit (Vazyme, EQ121-01). Five hundred nanograms of DNA was prepared for direct DNA sequencing by the nanopore approach in general accordance with the standard protocol of a Flow Cell Priming Kit (Nanopore, EXP-FLP002). Sequencing was started on the MinION MK1C platform using FLO-MINSP6 flow cells and the MinKNOW (version v3.2.6) script.

NCAP_72h_sequencing_FLO-MIN106_SQK-LSK109, which is recommended by ONT, with the exception that we restarted the sequencing runs an additional time to increase the active pore count during the first 2 h.

**NGS library construction.** The other half of the total cDNA (10 µl cDNA obtained in the nested RT-PCR step) was used to construct the NGS libraries. Preliminary PCR was carried out with KAPA HiFi HotStart ReadyMix. PCR amplicons were incubated with Exo I (NEB, M0568) at 37 °C for 15 min to eliminate ssDNA in the reaction volume and were then purified by VAHTS DNA Clean Beads (Vazyme, N411-01) at a 1:1 ratio.

A total of 140 ng of amplified products was fragmented by dsDNA Fragmentase (NEB, M0348), and Illumina sequencing libraries were constructed using the NEBNext Ultra II end repair-dA tailing module (NEB, E7546) and ligation module (NEB, E7595). Seven to nine cycles were performed for the final PCR amplification, depending on the DNA fragments used as input. Finally, the region from 200 bp to 700 bp was excised and recovered from 4% Nusieve 3:1 agarose (LONZA, 50090) by a Zymoclean Gel DNA Recovery Kit (Zymo, D4008). During the process of library construction, DNA purification was carried out by VAHTS DNA Clean Beads, and RNA or cDNA fragments longer than ~100 bp were purified by RNA Clean & Concentrator™-5 (Zymo, R1015) or Oligo Clean & Concentrator (Zymo, D4061) kits according to the specifications.

**NAP-SHAPE-MaP library construction**
For in vivo napRNA structure probing, HepG2 cells were treated with NAI-N3 (MCE, HY-103006) as previously described[32,33] with some modifications. Briefly, $5 \times 10^5$ HepG2 cells were seeded into 6-well plate and grown for 48 h up to 80% confluence. The cells were scraped and washed with PBS by centrifugation at $500 \times g$ for 3 min. Then the pellets were resuspended with 100 µl 100 mM NAI-N3 or DMSO mix and incubated at 37 °C on a Thermomixer at 1000 rpm for 5 min. The reaction was stopped by centrifugation at 4 °C $1000 \times g$ for 2 min. Subsequently, the supernatant was removed, and the cells were lysed in 1 ml RNAzol for RNA exaction. NAI-N3 modified RNA and DMSO-treated RNA were used for NAP-seq-NGS library construction.

**CAP-seq library construction**
PolyA-RNA was purified and enriched from 25 µg HepG2 total RNA by using VAHTS mRNA Capture Beads (Vazyme, N401-01). Enriched polyA-RNA were fragmented with NEBNext First Strand Synthesis Reaction Buffer (5 ×) in 15 µl volume at 94 °C for 9 min. Then the fragments were purified with 2 volumes of VAHTS RNA Clean beads. To enrich the 5'capped fragments, the fragmented RNA was incubated with 2 U Terminator exonuclease (Lucigen, TER51020), 1x Terminator reaction buffer A, and 20 U RRI in 40 µl volumes for 30 °C, 1 h. The RNA was purified by using RNA Clean & Concentrator Kits (ZYMO research, R1013). The terminator-treated RNA was treated with 5 µl Quick CIP (NEB, M0525S) and 1x CutSmart buffer at 37 °C for 30 min, followed by purification by RNA Clean & Concentrator Kits. Then, the eluted RNA was mixed with Cap-Clip Acid Pyrophosphatase (CELLSCRIPT, C-CC15011H) according to the induction to hydrolyze the pyrophosphate bonds of the 5'cap structures. The RNA was purified by RNA Clean & Concentrator Kits and then ligated with the 3'/5' adapters. Adapters-ligated RNA was reversed-transcribed by using SSIV and nested RT-PCR strategy for library construction.

**Plasmid construction and transfection**
An intron expression vector was generated by inserting a 136 nt β-globin/immunoglobulin intron at nucleotide position 275 in an Emerald GFP plasmid. The IRDR-L/R (inverted repeat-direct repeat left/right regions) elements and an SV40 promoter-driven puromycin

expression cassette were inserted to construct a Sleeping Beauty (SB) transposon-based intron expression backbone vector, namely, pSB-GIGP. Recombinant vectors expressing human DINAP, mouse mmu-novel-CD-6 and mmu-novel-CD-6-mD were constructed by PCR-based amplification of genomic DNA and were then subcloned into the β-globin intron expression vector at the Bsa I restriction site in the pSB-GIGP backbone. The cDNA sequences encoding the human DKC1 protein was amplified by PCR and was then subcloned into the vector pcDNA3.1-FLAG[79] at the Kpn I and BamH I sites, and the resulting constructs were designated DKC1. The fragments of the human DKC1 protein gene and FBL protein gene were respectively subcloned into pSBV2[80] at the EcoR I and EcoR V sites, and the resulting construct was named pSBV2-FLAG-DKC1 or pSBV2-FLAG-FBL. The primers used for cloning are listed in Supplementary Data 9.

Viafect reagent (Promega, E4981) was used to transfect plasmids into HepG2 cells in accordance with the specifications. Lipofectamine LTX with Plus Reagent (Thermo Fisher, 15338100) was used to transfect plasmids into C2C12 cells. To construct cell lines with stable DKC1 protein or FBL protein expression for the RIP experiment, the pSBV2-FLAG-DKC1 or pSBV2-FLAG-FBL construct was respectively co-transfected with pSB100× into HepG2 cells undergoing puromycin (Beyotime, ST551-250 mg) selection.

### CRISPR/Cas9 KO
A modified pSpCas9(BB)-2A-GFP plasmid (Addgene, 48138) containing dual guide RNAs recognizing the sequence of DINAP (chr18:22,857,010-22,857,186) with an optimal scaffold were transfected into HepG2 cells using Viafect according to the manufacturer's instructions. Forty-eight hours after transfection. single cells with a high level of green fluorescent protein (GFP) expression were isolated by sorting with a MoFlo XDP instrument (Beckman Coulter) into 96-well plates. Independent clones were allowed to grow for 2–3 weeks and were then tested for positive identification. The sgRNA sequences for DINAP were as follows: sgRNA1, 5′-GCTGGATGCCAGAAGCCCAG-3′; sgRNA2, 5′-TGTTGTCATCATTAGCCTTG-3′.

### RNA immunoprecipitation experiments
RNA immunoprecipitation (RIP) was carried out as previously described[81] with some modifications. Briefly, cells were harvested at 80% confluence by scraping and were then lysed with lysis buffer (10 mM HEPES, 100 mM KCl, 5 mM MgCl$_2$ and 0.5% NP-40 (V/V)) supplemented with 1 mM DTT, 100 U/ml RNase inhibitor, 1× protease inhibitor cocktail and 400 μM ribonucleoside-vanadyl complex (RVC) for 15 min on ice. After centrifugation at 4 °C and 15,000 × g for 15 min, the supernatant was isolated and precleared with 10 μl of protein G beads (Thermo Fisher) at 4 °C for 1 h. Then, 5% of the lysate was aliquoted as input, and 3 μg of an anti-FLAG antibody (mouse, Proteintech, 66008-3-Ig), anti-DKC1 antibody (mouse, Santa Cruz, sc-373956), or the corresponding mouse IgG (mouse, Proteintech, B900620) was added to the cell lysate for reincubation at 4 °C overnight. On the second day, 60 μl of protein G beads was conjugated to the antibody–protein complexes by incubation at 4 °C for 4 h. After washing with RIP washing buffer (50 mM Tris-HCl (pH 7.5), 1 mM MgCl$_2$, 150 mM NaCl and 0.05% NP-40 (V/V)) four times (3 min per wash), the beads were resuspended in 90 μl of RIP washing buffer. Simultaneously, the input lysate was also brought to 90 μl with RIP washing buffer. A total of 1/3 of the input or coimmunoprecipitation product was added to 5 × LB for denaturation at 99 °C for 10 min and was then analyzed by western blotting; the remaining 2/3 was used for RNA extraction by RNAzol and qPCR analysis.

### Isolation of cytoplasmic and nuclear fractions
In brief, 1 million HepG2 cells were seeded in 6 cm dishes. At 80% confluence, after being washed twice with cold PBS, the cells were harvested with 200 μl of precooled hypotonic lysis buffer (HLB; 10 mM Tris-HCl (pH 7.5), 3 mM MgCl$_2$, 10 mM NaCl, 0.3% NP-40 (V/V) and 10% glycerin (V/V)) supplemented with 1× protease inhibitor cocktail, 1 mM DTT, 10 U/ml RNase inhibitor, and 40 μM RVC for 10 min on ice. Then, the cells were collected by cell scraping and centrifuged at 10,000 × g for 3 min at 4 °C. Approximately 120 μl of the cell lysate supernatant was recovered, which contained the cytoplasmic components. After discarding the remaining supernatant, 1 ml of HLB was added to gently resuspend the precipitate containing the nuclear components, which was then centrifuged at 200 × g for 2 min at 4 °C. The supernatant was discarded, and the precipitate was washed once. The precipitate was divided into two parts (3:1) for RNA extraction with RNAzol and protein extraction with RIPA lysis buffer (25 mM Tris (pH 7.5), 150 mM NaCl, 0.1% SDS, 1% sodium deoxycholate and 1% NP-40) supplemented with 1× protease inhibitor cocktail (Roche). The cytoplasmic component fraction was also divided (3:1) for RNA and protein extraction. The cytoplasmic and nuclear fractions were then analyzed by western blotting and qPCR. For western blotting analysis, FBL (mouse, Proteintech, 66985-1-Ig, 1:2000) was used as the nuclear marker; β-Tubulin (rabbit, Cell Signaling Technology, 2146, 1:5000) and GAPDH (mouse, Proteintech, 10494-1-AP, 1:5000) were used as cytoplasmic markers. For the qPCR assay, MALAT1, SNORA23, and the GAPDH precursor mRNA (pre-GAPDH) were used as the nuclear markers, and GAPDH mRNA was used as the cytoplasmic marker.

### Immunofluorescence (IF) and smiFISH
smiFISH probes were designed online (https://bitbucket.org/muellerflorian/fish_quant) with Oligostan software. The secondary probes (FLAP), with two Cy3 moieties conjugated to both the 3′ and 5′ termini, and the primary probes, were synthesized by Sangon, Shanghai. For hybridization, the primary probes were hybridized with the FLAPs to form compound probes for performance. Both smiFISH and colocalization assays were performed in parallel according to previous methods with some modifications[82]. Cells at ~60% confluence were fixed with 4% Paraformaldehyde Fix Solution (Beyotime, P0099) and permeabilized for 5 min at 4 °C. After that, the slides were blocked with 5% FBS in 1× PBS (Thermo Fisher, AM9624) at room temperature for 30 min. Immunofluorescence staining was carried out with a primary antibody in vivo at a 1:250 dilution overnight at 4 °C. The next day, the cells were stained with Alexa Fluor™ 488/594-conjugated donkey anti-rabbit/mouse IgG (H + L) (Thermo Fisher, A21206; A21203) as the secondary antibody for 90 min at room temperature. Next, the slides were washed with PBS 3 times, and the cells were then dehydrated by sequential incubation with 70% alcohol, 95% alcohol and absolute ethyl alcohol for 5 min each for smiFISH. Then, rehydration was performed using hydration buffer consisting of 2 × SSC (Thermo Fisher, AM9763) and 50% formamide (Sangon, A100606) at the final concentration. Next, the cells were hybridized with the compound probes overnight at 37 °C. The next day, nonspecific hybridization products were eluted with hybridization buffers I, II, and III, which were made of 4 ×, 2 ×, and 1 × SSC from high to low concentrations. Finally, the cells were stained with 1× DAPI for 10 min at room temperature. Images were acquired with an LSM880 laser scanning confocal microscope with an Airyscan Fast module (Zeiss). The Gamma value of all FISH/IF images are assigned as default 1. A Fluorescent In Situ Hybridization Kit (RiboBio, C10910) was used for smiFISH. Primary antibodies used were as follows: anti-DKC1 antibody (mouse, Santa cruz, sc-373956); anti-FBL antibody (rabbit, Bethyl, A303-891A); anti-SC35 antibody (mouse, Abcam, ab11826). All probes used are listed in Supplementary Data 9.

### irNorthern blotting
irNorthern blotting (irNB) was carried out as described previously[83] with some modifications. DNA probes for has-snotron1, U6 and a marker were synthesized by Sangon and labeled with IRDye 800CW DBCO (LI-COR, 929-50000) or IRDye 680RD DBCO (LI-COR,

929-50005) respectively in 1× PBS at 25 °C overnight. The IR dye-labeled oligonucleotides were purified with Oligo Clean & Concentrator Kits (Zymo, D4061). The marker used for irNB consists of four RNA transcripts, including 55, 100, 150, and 200 bases, are synthesized in vitro from GFP using the T7 promoter. The DNA probe targeting the RNA marker was designed within the common region of the four RNA transcripts. For irNB analysis, 30 µg total RNA and the RNA markers were loaded on 8% urea-PAGE and then were transferred onto Hybond-N+ membranes (GE Healthcare, RPN303B). By fixing the membrane with 254 nm UV cross-linking, the membrane was hybridized with 40 pmol probes of hsa-snotron-1, 5pmol probes of Markers and 0.2 pmol probes of U6, and then washed at 42 °C. The signal was detected by a Li-Cor Odyssey Infrared Imager (LI-COR). The probes used for irNB are listed in Supplementary Data 9.

## qRT-PCR and western blotting analyses

Total RNA was reverse transcribed using HiScript Q RT SuperMix for qPCR (+gDNA Wiper) (Vazyme, R123-01), which contains genomic DNA removal agent. qPCR was performed with TB Green Premix Ex Taq II (Tli RNase H Plus) (Takara, RR820A). U6 and GAPDH were used as endogenous controls. The primers applied in qRT–PCR are listed in Supplementary Data 9.

Cells were harvested for extraction of total protein with RIPA lysis buffer. Equal amounts of total protein were loaded and separated by 4–10% SDS–PAGE (Thermo Fisher), transferred to nitrocellulose membranes (General Electric), and detected by immunoblotting with 20× LumiGLO Reagent and 20 Peroxide (Cell Signaling Technology, #7003) and Immobilon Western Chemiluminescent HRP Substrate (Millipore, 42029053). The primary antibodies used for western blotting were as follows: anti-DKC1 (rabbit, Abcam, ab156877, 1:2000), anti-MHC (mouse, R&D Systems, MAB4470, 1:4000), anti-Mef2c (rabbit, Proteintech, 10056-1-AP, 1:1000), and anti-GAPDH (mouse, Proteintech, 10494-1-AP, 1:5000).

## RNA stability assay

In total, $5 \times 10^5$ HepG2 cells were seeded into 3.5 cm dish to get 50% confluence overnight. Cells were treated with 5 µg/ml Act D (Selleck, America) and harvested at 0 h, 20 min, 40 min, 1 h, 2 h, 4 h and 8 h with RNAzol. The relative abundance of RNA was analyzed by qRT-PCR. Primers used in qRT-PCR assay were provided in Supplementary Data 9.

## Poly(T) RT-PCR

Poly(T) RT-PCR was applied to analyze RNA abundances. After gDNA removal by DNase I (Promega, M6101), a poly(A) tail was added to the 3′-OH end of RNA transcripts within the total RNA using *E. coli* Poly(A) Polymerase (NEB, M0276). RNA transcripts with poly(A) tails were reverse transcribed into cDNA with a poly(T) adapter consisting of a poly(T)$_{12}$-mer and a specific sequence by HiScript II Reverse Transcriptase (Vazyme, R201-01). With modification, the poly(A) tailing reaction and reverse transcription could be performed simultaneously in one reaction volume. Then, PCR amplification was conducted with Phusion High-Fidelity DNA polymerase (Thermo Fisher, F530) using a gene-specific forwards primer and a universal specific sequence as the reverse primer. The PCR amplicons were identified by agarose gel electrophoresis to verify the length and by Sanger sequencing. The primers are listed in Supplementary Data 9.

## Cell proliferation and colony formation assays

All cell lines used for the cell proliferation assay were seeded in 96-well plates at 3000 cells/well. Cell proliferation was evaluated using a Cell Counting Kit-8 (Dojindo, CK04) for 96 h in total. For the colony formation assay, 2000 cells were seeded in 3.5 cm dishes and grown for up to 10 days. On the 10th day, 0.11% crystal violet was utilized to stain cell colonies for 20 min after methanol fixation for 10 min. ImageJ was used to determine the number of colonies.

## NAP-seq data processing

The Illumina sequencing data contained 2 kinds of adapters: one was the sequencing adapters, and the other was the NAP-seq specific adapters. In contrast, the nanopore sequencing data contained only the latter adapters. Therefore, the sequencing data were processed differently.

For Illumina sequencing data, Cutadapt (v2.8)[84] was first used with the following parameters to remove the sequencing adapters from the paired-end reads: cutadapt -a AGATCGGAAGAGCACACGTCTG -A AGATCGGAAGAGCGTCGT -m 15 -e 0.15. The NAP-seq specific 5′ adapter (AAGCAGTGGTATCAACGCAGAGT) and 3′ adapter (AGTCGTAGTAAG TCTGTGCTCG), which marked the boundaries of the napRNAs, were subsequently moved by our in-house-developed program cutNapAdapter with the following parameters: -l 15 -e 0.1 -c 6 −C 6. Finally, the reads were mapped to the reference genome (hg38 or mm10) with STAR[85] software with the following parameters: --genomeLoad NoSharedMemory  --limitBAMsortRAM  60000000000  --alignEndsType EndToEnd --outFilterType BySJout --outFilterMultimapScoreRange 0 --outFilterMultimapNmax 20 --outFilterMismatchNmax 10 --outFilter MismatchNoverLmax 0.05 --outFilterScoreMin 0 --outFilterScoreMin OverLread 0 --outFilterMatchNmin 20 --outFilterMatchNminOverLread 0.8  --seedSearchStartLmax 15  --seedSearchStartLmaxOverLread 1 --alignIntronMin 20 --alignIntronMax 1000000 --alignMatesGapMax 1000000 --alignSJoverhangMin 20 --alignSJDBoverhangMin 10 --outSAMtype BAM Unsorted --outSAMmode Full --outSAMattributes All --outSAMunmapped None --outSAMorder Paired --outSAMprimary Flag AllBestScore --outSAMreadID Standard --outReadsUnmapped Fastx --limitOutSJcollapsed 5000000 --alignEndsProtrude 150 ConcordantPair --readFilesCommand zcat.

For nanopore sequencing data, Cutadapt (v2.8) was used with the following parameters to remove the NAP-seq specific adapters: -j 16 -g AAGCAGTGGTATCAACGCAGAGT -a AGTCGTAGTAAGTCTGTGCTCG -m 20 -e 0.3 -O 10. Cutadapt was then used again with the following parameters to move any possible adapters: -j 16 -g CGAGCACAGACTTACTACGACT -a ACTCTGCGTTGATACCACTGCTT -m 20 -e 0.3 -O 10. Then the randomized barcodes N6 were removed by Perl script. We retained only reads that contained NAP-seq specific adapters. Finally, the reads were mapped to the reference genome (hg38) with minimap2[86] with the following parameters: --junc-bed hg38.gencode.v30.geneAnno.bed -t 16 -k15 -w5 --splice -g2000 -G200k -A2 -B4 -O4,96 -E2,0 -C18 -z400,200 -ub --end-bonus=18 --junc-bonus=18 --splice-flank=yes -ub --sam-hit-only --secondary=no -a.

## CAP-seq data processing

For CAP-seq data, Cutadapt (v2.8)[84] was first used with the following parameters to remove the sequencing adapters from the paired-end reads: cutadapt -a TGGAATTCTCGGGTGCCAAGG -A GATCGTCGGACTGTAGAACTCTGAAC -m 3 -e 0.15. Then the clean reads were mapped to the reference genome (hg38) with STAR (Dobin et al.[85]) software. Finally, we calculated the 5′-end sites coverage of the sequencing read (endCov), the one nucleotide upstream (upCov) and the one nucleotide downstream (downCov). A positive 5′-Cap sites had to meet the following criteria: (1) upFC = endCov/upCov≥2; (2) downFC = endCov/downCov≥2; (3) read counts ≥ 10; (4) $p$ value < 0.05; (5) existing in two replicates; (6) located within mRNA 5′UTR region.

## Transcriptome-wide identification of napRNAs

To identify definite high-confidence napRNAs, we first assembled the continuous reads into contigs and then calculated the numbers of start reads containing specific 5′ adapters (startReadNum) and end reads containing specific 3′ adapters (endReadNum). We reasoned that the startReadNum and endReadNum of a candidate napRNA should be significantly higher over the upstream and downstream sequences, while the coverage of the contigs should also be significantly higher over the surrounding regions. Thus, we developed the computational

software napSeeker to calculate the numbers of start reads containing specific 5′ adapters (startReadNum) and end reads containing specific 3′ adapters (endReadNum). We calculated the fold change between startReadNum and the number of reads containing specific 5′ adapters located within 100 nt upstream and downstream (startFC); similarly, we calculated the fold change between endReadNum and the number of reads containing specific 3′ adapters located within 100 nt upstream and downstream (endFC). Next, we calculated the fold change between the coverage of the contigs and the regions within 20 nt upstream/downstream (up20ntFC/down20ntFC).

A high-confidence napRNA had to meet the following criteria: (1) startReadNum and endReadNum ≥7; (2) startFold and endFold ≥2; (3) up20ntFold and down20ntFold ≥ 2; (4) length ≥ 100. Moreover, the candidate napRNAs had to be expressed in at least 2 samples among all the human (or mouse) samples. Finally, we retained only napRNAs with a summary count ≥20. These stringent parameters allowed us to identify the highest-confidence candidate napRNAs.

### Identification of sliRNAs, snotrons, and misRNAs

The annotation files of known introns and exons were downloaded from GENCODE (human release 30 and mouse release 23)[25], the annotation files of snoRNAs were downloaded from snoRNABase (v3)[87], and the annotation files of miRNAs were downloaded from miRBase (release 22)[88]. The date of access to these datasets was Dec 1, 2021.

For sliRNA identification, we mapped our candidate napRNAs to the known introns and retained the napRNAs that shared both the 5′ start and 3′ end sites with a known intron.

For snotron identification, we mapped our candidate napRNAs to the known snoRNAs and retained the napRNAs in which one end was a known snoRNA and one end was located within 10 nt upstream or downstream of the known splice site.

For misRNA identification, we first clustered the known pre-miRNAs if they were located at a distance of <10 kb from each other. Then, these miRNA clusters were intersected with the napRNAs using BEDtools[89] with the parameter: intersect -s. Finally, we filtered the napRNAs that were located entirely within the pre-miRNAs, which were likely to be known pre-miRNAs or mature miRNAs.

### Integrated analysis of CLIP-seq and NAP-seq data

We downloaded CLIP-seq data from starBase (v2)[90] and intersected the peak regions with previously undiscovered napRNAs using BEDtools with the parameter: intersect -s. For sliRNAs and misRNAs, we extended the sequences 20 nt upstream and downstream of both ends of the corresponding napRNA and then intersected these 41-nt sequences with the peak regions identified by CLIP-seq. The method for snotrons was similar, but we extended these sequences 20 nt upstream and downstream of only the 3′-ends.

### Differential expression analysis

The R package limma[91] was used for differential expression analysis between each pair of different human cell lines (HEK293T, HepG2 and U87), between different HepG2 cell conditions (poly (I:C), CoCl2 and ADR treatment), and between different stages of mouse cell (C2C12) differentiation. The heatmap was plotted by the R package pheatmap 1.0.10 (https://CRAN.R-project.org/package=pheatmap), and the log2RPM (reads per million) values of the napRNAs are shown.

### Analysis of the histone modification distribution in napRNAs

The ChIP-seq data for different histone modifications in HepG2 and C2C12 cells were downloaded from the GEO database (GSE29611 and GSE36023, respectively). The date of access to these datasets was Dec 1, 2021. We used 4660 HepG2 napRNAs and 12,100 C2C12 napRNAs to analyze the histone modifications located in napRNA bodies. First, the ChIP-seq reads were mapped to the reference genome (hg38 or mm10) with STAR software with the parameters described in ENCODE.

Second, we calculated the value of each kind of histone modification for each napRNA, i.e., log2RPM (modification/control) within the napRNA regions, and we clustered the napRNAs by these histone modification values using Manhattan clustering. Finally, we used deepTools[92] to generate the heatmap of the different clusters.

### NAP-SHAPE-MaP score calculation

Preprocessing of NAP-SHAPE-MaP sequencing data was in consistent with NAP-seq Illumina sequencing data, including remove the same sequencing adapters and NAP-seq specific adapters. Then clean paired-end reads of three biological replicates were merge and mapped to each identified napRNAs using STAR software with the following parameters: --genomeLoad NoSharedMemory --limitBAMsortRAM 60000000000 --alignEndsType Local --outFilterType BySJout --outFilterMultimapScoreRange 0 --outFilterMultimapNmax 20 --outFilterScoreMin 0 --outFilterScoreMinOverLread 0 --outFilterMatchNmin 20 --seedSearchStartLmax 15 --seedSearchStartLmaxOverLread 1 --alignIntronMin 20 --alignIntronMax 1000000 --alignMatesGapMax 1000000 --alignSJoverhangMin 20 --alignSJDBoverhangMin 10 --outSAMmode Full --outSAMunmapped None --outSAMorder Paired --outSAMprimaryFlag AllBestScore --outSAMreadID Standard --outReadsUnmapped Fastx --limitOutSJcollapsed 5000000 --alignEndsProtrude 150 ConcordantPair --readFilesCommand zcat --scoreGap -1000000 --scoreDelBase -1 --scoreInsBase -1 --outFilterMismatchNmax 30 --outFilterMismatchNoverLmax 0.3 --outMultimapperOrder Random --outSAMmultNmax 1 --outSAMattributes MD --outFilterMatchNminOverLread 0.66. The alignments for individual napRNAs were extracted and processed with shapemapper_mutation_parser, shapemapper_mutation_counter, make_reactivity_profiles.py and normalize_profiles.py from ShapeMapper2 to produce the NAP-SHAPE-MaP score.

### Computational prediction of RNA secondary structures with constraints

The RNAalifold program in the ViennaRNA package (v2.4.18)[36] was used to predict the secondary structures of RNAs. MFE, based on the principles of thermodynamics, is utilized in RNA structure prediction by using a loop-based energy model and the dynamic programming algorithm introduced by Zuker et al.[93] to predict the most stable secondary structure of an RNA molecule. This lowest free energy structure is typically considered the most likely structure to exist within a biological system. The NAP-SHAPE-Map scores were used as constraints with parameters: --shape.

### RNA secondary structure visualization

Coevolutionary visualization of napRNA structures was performed by R2R (version 1.0.6)[94]. The z scores were calculated by RNAz (version 2.1)[95].

### Conservation analysis of napRNAs in different species

Sequences were aligned by Clustal W (version 2.1)[96]. UCSC chain files and liftOver tool were used for napRNA sequence conversion among different species.

### Stress-response and temporal profile analysis of napRNA expression

napRNAs in three HepG2 cellular stress-response states and four C2C12 cell differentiation stages were grouped into different clusters (patterns) using the fuzzy c-means algorithm in Mfuzz[97].

### Motif and GO analysis of sliRNAs

For motif analysis of sliRNAs, we extended the sequences 5 nt upstream and downstream of the splice sites and branchpoints[98] and then plotted the motifs of these 10-nt sequences using the R package ggseqlogo[99]. GO analysis was performed by clusterProfiler[100], and only the GO terms with similarity less than 0.7 were retained.

## Normalization and visualization of sequencing data in Genomics Viewer

The public HepG2 and C2C12 data were downloaded from the GEO or ENCODE database (The date of access was Dec 1, 2021). Total RNA-seq data were from the GSE88089 dataset, poly(A)-depleted RNA-seq data were from the GSE90229 dataset, poly(A) RNA-seq data were from the GSE90173 dataset, C2C12 cell microRNA-seq data were from the GSE143099 dataset, and HepG2 cell microRNA-seq data were from the ENCSR730NEO dataset. NAP-seq data and all downloaded data were normalized by the reads per million mapped reads (RPM) values and were visualized using the UCSC Genome Browser[101] or the Integrative Genomics Viewer (IGV) visualization tool (version 2.8.2)[102].

## Statistics and reproducibility

Data are presented as the mean values ± SEMs. Student's $t$ tests were employed for comparisons between the two experimental groups. All statistical analyses were conducted using GraphPad Prism 8 or R (4.2.0 (2022-04-22)). The number of biological replicates for each experiment is specified in the figure legends. For NAP-seq-NGS, three independent experiments were conducted for HepG2, HEK293T, U87, and four different stages of differentiation of C2C12 cells, respectively. Additionally, NAP-seq-TGS was independently performed three times for HepG2 cells and C2C12 myoblasts. Three independent experiments of NAP-SHAPE-MaP were carried out for HepG2 cells. CAP-seq analysis and polyA-selected RNA-seq analysis in HepG2 cells were each performed at least two times. PolyA-selected RNA-seq analysis during four differentiated stages of C2C12 cells were conducted for three times. All images of polyacrylamide gels, agarose gels and northern blots are representative of at least two biological replicates. No statistical methods were used to predetermine sample size.

## Reporting summary

Further information on research design is available in the Nature Portfolio Reporting Summary linked to this article.

## Data availability

The NAP-seq data generated in this study have been deposited in the NCBI Gene Expression Omnibus (GEO) under accession code GSE192632. The RNA-seq, CAP-seq, NAP-SHAPE-MaP, nanopore NAP-seq-TGS data generated in this study have been deposited in the NCBI GEO under accession code GSE228168. GSE88089 for total RNA-seq, ENCSR000CRX for small RNA-seq and GSE160887 for PEN-seq in HepG2 cells are public data. There are no restrictions on data availability. The candidate napRNAs data generated in this study are provided in the Supplementary Data. Source data are provided with this paper.

## Code availability

Software are available on the Github: napSeeker (https://github.com/junhong-huang/napSeeker), https://doi.org/10.5281/zenodo.10657490 and cutNapAdapter (github, https://github.com/junhong-huang/cutNapAdapter), https://doi.org/10.5281/zenodo.10670898. The Perl scripts are available in our Github: (https://github.com/junhong-huang/NAP-seq-Perl-scripts), https://doi.org/10.5281/zenodo.10670900.

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

## Acknowledgements

This research was supported by the National Key R&D Program of China (2019YFA0802202 (to J.Y.), 2022YFA1303300 (to J.Y.)); the National Natural Science Foundation of China (32225011 (to J.Y.), 91940304 (to J.Y.), 31971228 (to J.Y.), 31770879 (to J.Y.), 32100467 (to S.L.), 32370588 (to B.L.), and 31970604 (to L.Q.)); Youth Science and Technology Innovation Talent of Guangdong TeZhi Plan (2019TQ05Y181 (to J.Y.)).

## Author contributions

J.Y., S.R.L., L.Q. and B.L. conceived and designed the entire project. J.Y. and L.Q. designed and supervised the research. S.R.L., J.H., J.Z., S.C., W.Z., C.L., Q.L., D.W., P.Z., S.H., S.L., K.Z. and J.Y.Y. performed the experiments and/or data analyses. J.Y. and J.H. performed the genome-wide or transcriptome-wide data analyses. J.Y., S.R.L., B.L. and L.Q. contributed reagents/analytic tools and/or grant support. J.Y., L.Q., S.R.L., J.H. and B.L. wrote and revised the paper. All authors discussed the results and commented on the manuscript.

## Competing interests

The authors declare no competing interests.
