## [Peer Review File · Nature Communications]

REVIEWER COMMENTS

Reviewer #1 (Remarks to the Author):

In this manuscript, Liu et al describe an experimental method (NAP-seq) designed to identify low abundance, uncapped transcripts (> 100 nt in size). This approach has allowed the identification of new RNA species, some of which can be classified as structurally related families such as snoRNAs (box C/D and H/ACA snoRNA), linear intron RNAs (sliRNAs), snoRNA-intron (snotrons) and RNAs embedded in miRNA spacer regions (misRNAs). It is an interesting work and the reading of the manuscript is relatively pleasant. Although I consider this study as a preliminary descriptive work, expression data and loss-of-function experiments (CRISPR-mediated knockouts) suggest that some of these newly-identified RNA species, collectively named noncapped RNAs (napRNAs), may exert regulatory functions, possibly during myoblast differentiation or cell proliferation. However, I find that authors tend to over-interpret some of their observations. Indeed, they should discuss the trivial possibility that some (most) of the newly identified RNA species (low abundance, poorly conserved and overlapping interspersed repeated elements) could simply represent degradation products and/or RNA intermediates during pre-mRNA splicing or pri-miRNA processing. In this framework, that expression of some these ncRNAs vary according to biological conditions does not necessarily mean that they are functionally important. Finally, I recommend that the authors avoid the excessive use of words like "exceptional/exceptionally" which in my opinion does not bring anything scientifically (i.e., What does it mean? exceptional compared to what?). Before considering publication of this manuscript, I believe it is essential that the authors address the specific points listed below.

>>>Lanes 59-60: "Using NAP-seq, we compiled the first human and mouse napRNA transcriptomes and demonstrated their evolutionary conservation as well as their response to different cellular conditions".

I find this sentence ambiguous. napRNAs can be detected in both humans and mice, but the authors do not show any functional and/or structural conservation between a given human and mouse napRNA. See also my other comment about hsa-napRNA-1642.

>>>Lanes 89-90: "To increase the diversity of napRNA species and discover low-expression napRNAs, we used the RNase H method to remove high-abundance rRNAs, snRNAs and snoRNAs (Fig. 1a)".

The list of antisense DNA oligos should be given. On the other hand, it would be useful to give more information on the efficiency of this depletion method, perhaps by giving the number of sequencing reads corresponding to the rRNA species. In other words, what is the percentage of remaining reads corresponding to ncRNAs that were initially destined to be degraded by RNase H treatment?

>>>Lanes 129-130: “We further profiled the transcriptomes of 3 different stress responses (poly (I:C), adriamycin (ADR), and CoCl₂) in human HepG2 cells ...”.

I recommend that the authors give some additional information about the rationale of this experiment, for example what are the modes of action of the chemical compounds employed. This will make the reading easier for people not familiar with these molecules/treatments.

>>> Lanes 134-136: “We further filtered the candidate napRNAs identified by napSeeker through requiring that the napRNA must exist in at least two sequencing libraries, have more than 20 sequencing reads and not overlap with known annotations, such as rRNAs, tRNAs, snoRNAs and CDSs (Supplementary Fig. 2a, see Methods).

Just out of curiosity, I looked at the list of newly identified snoRNAs, it turns out that mmu-novel-CD-1 is actually scaRNA28 (= ZL1). I did not check the other snoRNA candidates. It is important that the authors verify that the other snoRNA candidates do not correspond to already annotated ncRNAs.

>>>Lanes 147-149: “One particularly interesting example is a new long napRNA (hsa-napRNA-1642) that has a significantly conserved RNA secondary structure identified by evaluating pairwise covariations using R-scape software 20 (Supplementary Fig. 2f)”.

If the authors wish to emphasize that some napRNAs are conserved through evolution, and to exploit the presence of compensatory base change as evidence for the conservation of their 2D structures, then the authors should compare RNA structures between phylogenetically distant species. It would indeed be more informative, and especially more convincing, to examine napRNAs that originated more than 300 Myr ago, as mentioned but not shown in Supplementary Fig. 2g. Indeed, as I understand, hsa-napRNA-1642 is only found in primates and it is not indicated which species were used to identify covariations in the Supplementary Fig. 2f. In addition, I do not see any “covarying mutations” as indicated by the green code. In my view, it is important that the authors show sequence alignments and compensatory base changes supporting the presence of stems.

>>>Lanes 153-155: “We discovered that most (14284) napRNA families were primate-specific, but 3313 families likely originated more than 90 million years (Myr) ago, and 186 families likely originated more than 300 Myr ago (Supplementary Fig. 2g)”.

Do the most conserved sequences correspond to intronic sequences and/or intergenic sequences? I think it would be useful to give more information about the most phylogenetically conserved napRNAs, and therefore more likely to perform functions retained by natural selection.

>>>Lanes 164-168: “Interestingly, we discovered that 1620 differentially expressed napRNAs displayed significant changes in expression for up to 72 hours (72 h, D72) after C2C12 cells were induced to differentiate and maintained relatively stable expression until 144 h (D144) after induction of differentiation (Supplementary Fig. 3d, 167 e and Table 3), suggesting that napRNAs may play crucial roles at the early stage of skeletal muscle differentiation”.

I am not familiar with the differentiation pathways in C2C12 cells. I imagine that there are already gene expression data available. Do the differentially expressed napRNAs correspond to portions (intronic?) of transcripts from genes already known to vary in expression after induction of skeletal muscle differentiation? Or are these differentially expressed napRNAs generated from independent (intergenic?) transcription units? Finally, and more importantly, that expression levels of some napRNAs vary does not imply functionality and, without functional validations, I find premature to propose a “crucial role at the early stage of skeletal muscle differentiation”

>>>Lanes 170_184

I am not sure I understood the point of this "epigenetic section". What is the message that the authors wanted to deliver? Are we to understand that some napRNAs are generated independently from their own promoters? I think it is necessary to conclude this paragraph more precisely.

>>> Lanes 206 – “We next developed new bioinformatic pipelines to identify novel small nucleolar RNAs (snoRNAs), including C/D box and H/ACA box RNAs, from NAP-seq profiles (Supplementary Fig. 5a)”.

As indicated by their names snoRNAs accumulate in the nucleoli. As I understand it, the sub-cellular localization of these putative snoRNAs has not been experimentally tested. I advise the authors to remain more factual and write RNA species containing sequence motifs found in C/D and H/ACA snoRNAs. In this context, are the new snoRNA-like conserved between human and mouse? I also find it unfortunate that the authors do not show the interaction of these putative “snoRNAs” with fibrillarin and/or Dyskerin. This could be done easily by immunoprecipitation experiments.

>>>Lanes 228-230 - Surprisingly, compared with the known AluACA RNAs that cover only approximately 25% of the whole length of the host Alu element, full-length AluACA RNAs could be identified by our NAP-seq method (Supplementary Fig. 7h), and most of these were found to cover more than 50% of the Alu element length (Fig. 2g)”.

I am not sure I understand. Does this imply that the previously published AluACA RNAs result from the processing of longer RNA precursors? Or did the authors detect a new class of AluACA RNAs?

>>>Lanes 239-241 – “We also detected the subcellular localization of mmu-novel-CD-6 by fluorescence in situ hybridization (FISH) and found that it was localized mainly in the nucleus (Supplementary Fig. 8e)”.

It would be interesting to provide more details on the nucleoplasmic localization of CD-6. It seems that RNA FISH signals do not correspond to nucleoli. Do these signals correspond to the SC-35 domains (nuclear speckles)? Given that CD6 is embedded within and processed from an intron and because its sequence overlaps a repeated (L2) element, how do the authors know that FISH signals originate from the fully processed CD36, and not from the unspliced (host) pre-mRNA, or even from other L2 elements generated elsewhere in the genome?

>>>Lane 245-248 ... “found that its overexpression had no influence on the mRNA level of its host gene (Exosc2) (Supplementary Fig. 8g) and significantly decreased the protein expression levels of the established myogenic markers MHC and Mef2c (Supplementary Fig. 8h), suggesting that mmu-novel-CD-6 inhibits myoblast differentiation”.

I am worried about the biological relevance of this type of experiment where the authors introduce supraphysiological doses of a given RNA ? In other words, what is the relative abundance of the transfected (ectopic) form of CD6 vs the endogenous one? The authors should introduce a negative control such as an inactive form of CD6 bearing mutations in the C- and/or D-box. Indeed, this mutated version of CD6 is not expect to accumulate. Finally, how do the authors envision CD6 inhibiting myoblast differentiation? Although I am aware that answering this question is beyond the scope of this study, it would still be interesting to mention some hypotheses in the discussion section.

>>>Lanes 258 – “We analysed our NAP-seq reads located within human introns and discovered that thousands of 5'-start and 3'-end sequencing reads were precisely located within the 5' splice site (5'-SS) of 6,448 introns or the 3'-SS of 17,397 introns in HepG2 cells (Fig. 3a, b), implying that these introns may be stable in linear form rather than rapidly degraded”.

From my point of view, that intronic RNA species can be detected does not mean necessarily that these latter correspond to stable RNA species. Did the authors experimentally demonstrate their intrinsic stability by treating cells with high doses of Actinomycin D?

>>>Lanes 280-283 – “In addition, we found that the expression of siRNAs was dynamically changed during mouse myoblast differentiation; siRNAs generally exhibited high expression at the late stage of myoblast differentiation (Supplementary Fig. 10e)”.

Could it be that this simply reflects changes in post-transcriptional regulation of their host-gene transcripts?

>>>Lanes 320 – 322 - “Moreover, through evolutionary conservation analysis, we observed that the intron sequences of hsa-snotron-1 which contained SNORD2 were highly conserved in mammals (Supplementary Fig. 11c), implying that hsa-snotron-1 may play a crucial biological role as a whole snotron and not only as a snoRNA”.

I am very surprised by the Northern blot shown in Figure 4G in which the expression level of snotron-1 appears to be of the same order of magnitude as that of fully processed SNORD2, or even higher than that of spliceosomal snRNA U6. Is this also the case for the other snotrons? I also suggest the authors to quote, and discuss accordingly the following publications: PMID: 34725166 and 35878033

>>>Lanes 336-340– “We further found that a number of RBPs potentially interact with the 3'-ends of the snotrons (Supplementary Fig. 11j). Many RBPs showed different binding preferences between snotrons and other introns containing snoRNAs. Notably, AQR showed the greatest difference among all RBPs (Supplementary Fig. 11j). Given that AQR interacts with many splicing factors (Supplementary Fig. 11k) and may facilitate the folding of snoRNA sequences and snoRNP biogenesis, we speculated that the AQR protein may contribute to the biogenesis and stability of snotrons”.

As I understand it, these are only in silico predictions. I find this section very speculative and unconvincing. From my point of view, this information can be removed without altering the message the authors wish to deliver. Alternatively, the authors could perform immunoprecipitation experiments. Given the expression level of snotron-1, it should not be difficult to test the interaction of AQR. In the same vein, it would also be very interesting to test if the core SNORD binding proteins (Snu13, Fibrillarin, Nop56 and Nop58) associate with snotron-1.

>>>Lanes 361-...Regarding napRNAs processed from miRNA clusters (misRNAs) -

Although I am not really convinced by the notion that misRNAs could have a regulatory function, I find interesting the possibility to propose an order in the maturation pathway for clustered miRNAs. In this respect figures 15e-f would deserve to be highlighted and discussed in the full manuscript rather than in the supplementary data.

>>>Regarding DINAP (DKC1-interacting napRNA)

I noted that DINAP could perhaps be considered a C/D box-containing RNA since it has a correctly positioned C- and D-box, with the possibility of forming a canonical 5'-3' terminal stem like those found in canonical snoRNAs involved in rRNA methylation (only the D box differs slightly from consensus; CAGA instead of CUGA). I am surprised that the authors do not mention this possibility.

>>> Lanes 407-409 “In addition, we further showed that DINAP was colocalized with DKC1 in the nucleoplasm by using fluorescence in situ hybridization (FISH) (Fig. 6d)”.

The authors should recall that DCK1 is an abundant nucleolar protein present in small nucleolar ribonucleoprotein particles (H/ACA RNPs) that convey specific uridines from ribosomal RNA into pseudouridines. Thus, I am very surprised by the IF signals that apparently do not visualize nucleoli. Can the authors comment on this rather unexpected observation? Is this the case with different antibodies? In any case, the authors should include a nucleolar marker as a control (e.g., fibrillarin). Can the authors demonstrate an interaction of DINAP with endogenously expressed DCK1? As I understand it, the interaction was only validated after overexpression of a tagged version of DCK1.

>>>Lane 428 - “These results revealed that DINAP regulates the stability of the DKC1 protein” -

How do the authors imagine that DINAP regulates the stability of DCK1? I find the model proposed in figure sup17i ambiguous because it suggests that DINAP is much more abundant than DKC1, which in my opinion may not be the case. Can the authors argue, at least estimate the relative stoichiometry of the two partners likely to interact.

>>> Lane 485 – “these novel snoRNAs derived from repetitive elements are not associated with rRNAs and snRNAs, suggesting that they may perform novel regulatory functions”.

I find this sentence ambiguous. Did the authors look for potential base pairing interactions between napRNAs and rRNA/U-snoRNAs? What could be the mode(s) of action of these C/D and H/ACA-like small RNAs?

>>> Lane 524-525 - “In this study, we discovered by the NAP-seq method that many miRNA clusters produce stable miRNA spacer sequences (i.e., misRNAs) in various cells”.

As I mentioned earlier for other napRNAs, there are no experimental data indicating that misRNAs are stable.

Reviewer #2 (Remarks to the Author):

In this manuscript Liu and colleagues apply a new RNA-seq method to be able to characterize cellular transcripts that lack 5'cap and/or 3'PolyA and therefore are not detected by standard sequencing methodologies. They use both short-read (Illumina) and long-read (Nanopore) RNA sequencing in order to obtain more information about the transcripts, and apply it to several human cell lines with different treatments and different differentiation stages of the mouse skeletal muscle cell line C2C12. With this approach, they identify a number of previously non-annotated transcripts that they name napRNAs, and that in many cases contain repetitive elements. The authors identify several significant types of transcripts, like PolIII-transcribed RNAs, RNAs containing snoRNA sequences, linear introns, snoRNA-intron or containing miRNA clusters. Many have in common being highly structured and show changing levels upon stress or differentiation. Finally, the authors show some experimental data on one of these napRNAs, named DINAP, which stabilizes DKC1 protein in HepG2 cells.

This is interesting work, presenting a novel methodology that allows to have a better view of the different RNA species that are present in the cells. However, it remains to be shown whether the identified RNAs are not intermediaries of processing without a specific function, therefore not biologically relevant.

More specifically:

1. Although central to the paper, the way data are analyzed is not sufficiently explained in the main text. For instance, how are the long-read and short-read data integrated? The use of LR-seq is a very positive aspect of the study, however it only has been applied to one cell line. How many of the findings are supported by LR-seq? This is not sufficiently clear.

2. The finding of many repeated sequences is very intriguing. Is it possible that this is due to some technical artefact? The authors should provide additional analyses and evidences to exclude this possibility.

3. Most of the RNA species found seem to be some type of processing intermediate that is more stable due to its structure. It would be a significant finding if some of these RNAs are proved to have some

important biological role. Changes in expression or association with histone modifications however are not functional evidence. The authors should avoid overstatements.

4. Regarding the changes in expression, it should be shown whether they correlate with changes in expression of the host genes, and therefore could be explained by them.

5. The authors should analyze the half-life of the napRNAs.

6. What are the steady-state levels of the napRNAs? How do they compare to other functional RNA species? (i.e. mRNAs and ncRNAs)

7. The experimental data supporting the functionality of DINAP are not convincing. It is unclear what type of RNA is DINAP. Does it contain snoRNA sequences? DKC1 is known to interact with snoRNAs. How was the DINAP-DKC1 interaction identified in the first place? Just based on eCLIP data? Is it the only protein that interacts with it by eCLIP? The way the eCLIP data are presented should be improved, for instance showing a larger genomic window. Also, how many DKC1 peaks are in the genome? What is the intensity of DINAP peak compared to others?

8. The FISH experiment in Fig 6c lacks controls (knockdown or knockout of DKC1 or DINAP) to show the specificity of the signal.

9. How does DINAP overexpression levels compare to its endogenous levels?

10. It should be shown in the same experiment in parallel the effect of DKC1 and DINAP knockdown and overexpression alone and in combination.

Reviewer #3 (Remarks to the Author):

In the manuscript titled "NAP-seq reveals novel classes of exceptionally structured noncoding RNAs with regulatory functions," authors Liu et al, describe a novel method to capture long (>50) non-capped RNAs. Using a series of end digestion, RNA isolation, rRNA depletion and nested RT-PCR, the authors demonstrate they can indeed enrich for non-capped RNAs for which the majority arise from repetitive elements such as Alu SINE RNAs in humans and B2 SINE RNAs in mouse as well as LINE and MIR derived RNAs and snoRNAs. The authors also describe library representation of novel transcripts with

“exceptional structure” and attempt to functionally validate some of the biological functions of the “novel” non-capped RNAs. Their method is certainly an improvement for the study of non-capped RNAs, however a number of validating experiments should be conducted and some points need to be addressed before the manuscript is published.

In detail:

The authors develop two library approaches following RNA preparation and reverse transcription (Nanopore and Illumina). To demonstrate they are indeed enriching non-capped RNAs, they should for both library methods, prepare a standard sequencing library for comparison to their developed method.

Furthermore, because their RNase H based digestion of rRNA and other high abundance ncRNAs used custom probes, they should at the very least quantify those targets before and after depletion to assess the efficiency of depletion.

Moreover, to validate the predicted structures as “exceptionally structured” the authors should conduct at least some in vitro validation such as SHAPE-seq or ssRNA digestion assays. Secondary structure prediction software is generally used to fit models rather than be used as a ground-truth structure. Lastly, the authors should consider the use of spiked in RNA such as a non-capped RNA and capped RNA to assess the efficiency of enrichment and representation in the library.

Beyond experimental validation, the methodology section requires soem more clarification for the protocol to be replicable. The authors should at least include: the amount of cDNA used for both libraries, “1/2” is not very quantitative. What number of PCR cycles are used? What Nanopore sequencing kit did they use? Was library barcoding use? Also, while I understand why cDNA fragmentation was used since RNA fragmentation would introduce erroneous non-capped RNA fragments, a descriptive methodology of fragmentation must be included. Additionally, detailed information of the software used is missing in many cases (for example version number), especially in the case of the supplementary schematic for “Napseeker.”

Inclusion of the following would also improve the manuscript:

RIN values for each sample. Low RIN could erroneously increase the number of Non-capped RNAs.

Discussion on the possibility of RNA processing to increase novel non-capped RNAs such as the processing of B2 and Alu RNAs described in Zovoilis et al, 2016 and Cheng et al, 2020-2021.

Also, a discussion on where Nanopore vs Illumina is more useful. Could there be extra long Non-capped RNAs that Nanopore would be more suitable to resolve. This may require modification of the

amplification protocol to include a long-amplification step instead. In suppl 3a, what distribution of differentially expressed napRNAs change in each condition?

The introduction should discuss further why up to this point, sequencing methods have not captured these RNAs. What about structure prevents their library preparation? Other techniques such as TGIRT-seq can reverse-transcribe complicated structures such as snoRNAs.

Some additional minor points that in my opinion should be addressed are:

Line 101

- Could mention the cell line names.

Sentences from line 111-116

- At this point, have they checked if their analysis missed any known napRNA (false negative)?
- Are there any unexpected peaks that appear in non-terminal region? If so, how would that be explained?

Line 130

- Better to briefly describe what each condition is.

Line 135 / line 822-824

- How values like "20" were decided? Based on preliminary tests, experience, etc.?

Line 145

- Briefly explain what lower MFE indicates to those who're not familiar with this?

Line 148 / line 870

General question about using Rscape and the secondary structure predictions

- how confident they are about predicted pairings, how conserved their input sequences are and how many sequences used for each prediction?

- Use of Rscape is okay when input sequence are mostly conserved. But with a less conserved input alignment, predicted pairings (even with a high score) can be more suspicious, and many mismatches, bulges, larger loops would be observed in most individual sequences.

- Did they do more analysis for the secondary structures? E.g. whether they share common flanking area or are from similar elements, are there other elements that may potentially interact with them?

- Are they planning to provide sample alignments that correspond to structures they show in main and suppl figures?

Line 203

- Suppl. Fig 5c,d aren't showing a secondary structure at all.

Line 292

- A bit confusing. Text says "no difference" but I read the bars are different as in suppl fig 10j.

Line 319

- "possibly because of the complex" needs more explanation.

Line 355/fig 4i

- how did they come to those within a short/long distance? -- make them clearly shown in fig 4i?

- The last step is confusing and hard to understand solely based on what's drawn. Consider better annotation text/colour-coding.

Line 802

- is the perl script included in one of their github repos or available upon request?

Line 831

- Need to note the date of access, also applies to other text where "downloaded from database XX" is mentioned.

Fig1e

- The y-axis better to use $1e+05$ to be consistent with others.

Line 1126

- What defines "same terminal sites"? Exact sequence match or other criteria?

Fig 4a

- the top structures are a bit hard to understand immediately. Could colour-code and highlight the region that's considered to be the "distance".

- Also for panel b: the last x-axis label should be ">50" since the one before it includes 50?

Fig 4f

- As a main figure, this structure is visually too messy.

Suppl fig 1c/d

- Would be easier to compare if they're in one bar plot since they have the same categories.

Suppl fig 3b

- Y-axis labels: What cell line is each treated sample?

SUMMARY:

We are very grateful to the reviewers for their positive appraisals of our manuscript and very much appreciate their thoughtful and constructive suggestions and comments. Following their suggestions, we have performed a large set of additional experiments and data analyses. With all those new results, we have carefully revised our manuscript and addressed all the concerns accordingly. The following is a brief summary of the major new experiments and data analyses that we have conducted during the revision.

- (1) To validate the secondary structure of napRNAs *in vivo*, we developed NAP-SHAPE-MaP, an approach combining SHAPE-MaP and our NAP-seq to probe intact RNA structures of napRNAs. As expected, novel napRNAs with a significantly conserved RNA secondary structure identified by RNAalifold software were verified by NAP-SHAPE-MaP. Moreover, to enhance the accessibility of structural insights for individual napRNAs, we have established a dedicated web page known as the "napRNA structure" (accessible at <https://rnasysu.com/napSeq/structure.php>). This webpage offers a comprehensive array of details, encompassing sequence alignments, structural representations, conservation scores, and species-specific co-variation identifications. This resource is designed to facilitate effortless access for readers seeking a profound understanding of the intricate structural attributes of napRNAs.
- (2) To demonstrate that NAP-seq are indeed enriching non-capped RNAs, we conducted additional experiments and analyses as follows: Firstly, to confirm the removal of capped RNAs in NAP-seq, we performed CAP-seq experiments, identifying 66665 high-confidence 5'-cap sites within HepG2 cell lines. When comparing these 66665 5'-cap sites to the 5'-start sites of napRNAs identified through NAP-seq, we found that only three sites (0.2% = 3/1220) exhibited overlap. This result implies that the NAP-seq method markedly enriches non-capped RNAs. Furthermore, we carried out polyA-selected RNA-seq experiments to investigate the extent of non-capped RNA enrichment across various experimental methods. As a result, we found that NAP-seq exhibited a significant enrichment of known non-capped ncRNAs (such as snoRNAs) in comparison to the polyA-selected RNA-seq approach conducted on the same cell lines. These results provide additional substantiation for the capacity of NAP-seq to effectively enrich non-capped RNAs.
- (3) To investigate whether changes in napRNAs during development correspond to changes in host genes, we performed RNA-seq experiments to quantify the expression of napRNA host genes. We found that most napRNA expression showed weak correlation with their host genes. Furthermore, we found that only a subset of differentially expressed napRNAs (250 out of 6808, i.e., 3.7%) corresponded to differentially expressed host genes. These results suggest that the alterations in the majority of napRNAs do not synchronize with changes in host genes.
- (4) To investigate the concurrence of napRNAs identified by NAP-seq-NGS and NAP-seq-

TGS methods, we performed NAP-seq-TGS experiments in C2C12 mouse myoblast cells. Our comprehensive analysis unveiled a considerable overlap between the napRNAs detected using NAP-seq-NGS and those identified through NAP-seq-TGS experiments in same human and mouse cell lines. This cross-validation across different sequencing platforms substantially strengthens the reliability of the NAP-seq method and the associated napRNA datasets.

- (5) To evaluate the effectiveness of the RNase H-based RNA depletion technique, we performed qRT-PCR assays, comparing gene expression (e.g. rRNAs, snRNAs and snoRNAs etc.) in samples subjected to RNase H treatment and those without. We observed that this technique achieved a remarkable depletion efficiency of up to 99% for the targeted RNAs. Furthermore, we performed an assessment of the RNA composition in the RNase H-treated HepG2 sequencing library by analyzing the fraction of reads aligning to genomic categories and specific RNA regions. The fractions of reads aligning to rRNA exhibited a substantial reduction and selected RNAs (SNORA73A, SNORA73B, RMRP, and RPPH1) were effectively reduced to undetectable levels, underscoring the successful accomplishment of depletion.
- (6) To explore the interaction between sonRNA-like napRNAs and FBL/DKC1, we conducted FBL/DKC1 RIP-qPCR experiments. This endeavor culminated in the successful detection of specific snoRNA-like napRNAs that exhibit discernible interactions with FBL/DKC1. Notably, our observations revealed a substantial enrichment of DINAP in the DKC1-RIP-qPCR analysis, providing strong evidence for a robust interaction between endogenous DKC1 and DINAP.
- (7) To demonstrate the stable expression of napRNAs *in vivo*, we performed the steady-state level analysis and half-life assays in HepG2 cells. We calculate the steady-state levels of the napRNAs and other functional RNA species using total RNA-seq dataset, and we found that the expression level of napRNAs was similar to that of known non-coding RNAs (ncRNAs). To examine the half-life of napRNAs, we treated HepG2 cells with Actinomycin D and measure the abundance of napRNAs by qRT-PCR. We observed that the half-life of snoRNA-like napRNAs is approximately 2-4 hours. Snotrons had a half-life of 20 minutes to 2 hours, which were similar to or longer than that of corresponding snoRNAs. Comparatively, the half-life of sliRNAs were around 20 minutes to 1 hour, which were longer than that of the control ACTB intron which is rapid degraded. Additionally, the half-life of misRNAs was found to be approximately half that of corresponding miRNAs. These results contribute to our understanding of the diverse stability profiles exhibited by different classes of napRNAs.
- (8) According to the constructive suggestions from the reviewers, we incorporated additional markers to determine the subcellular localization of DINAP and mmu-novel-CD-6. Specifically, we included FBL as a nucleolar marker and the SC-35 domain as a nuclear speckle marker. This refinement of our methodology enabled us to precisely decipher the subcellular distribution of DINAP and mmu-novel-CD-6. As a result, we gained deeper

insights into their precise cellular localization and potential functional roles.

- (9) To illustrate the functional role of mmu-novel-CD-6 during the myoblast differentiation, we conducted both gain-of-function and loss-of-function experiments. These experiments aimed to assess the impact of mmu-novel-CD-6 on the expression levels of key myogenic markers, specifically MHC and Mef2c. The outcomes of our study vividly demonstrated that mmu-novel-CD-6 indeed exerts a suppressive influence on the process of myoblast differentiation.

All the text in our revised manuscript that we have changed or added is marked in **red**. The deleted text has been removed entirely from the manuscript. We hope the revised manuscript would fulfill the requirements of the reviewers and will be suitable for publication in *Nature Communications*.

To avoid internal redundancy, we lay out below a detailed description of the key experiments and analyses conducted as part of this revision, allowing us to then refer to them concisely in the point-by-point response. **Our text is in blue, reviewers' text is in black.**

General point 1: Efficiency analysis of rRNA and high-abundance RNAs removal strategy based on RNase H. In the NAP-seq library preparation, we employed an RNase H-based RNA depletion technique to effectively eliminate rRNA from the input RNA. Reviewers #1 and #3 both raised questions and provided suggestions for evaluating the RNase H-based RNA depletion technique. To address this point, we performed qRT-PCR assays comparing genes with and without RNase H treatment. As depicted in **Reviewer Figure 1A (revised Supplementary Fig. 1b, c)**, we observed that this technique achieved a remarkable depletion efficiency of up to 99% for the targeted RNAs. Furthermore, we performed an assessment of the RNA composition in the RNase H-treated HepG2 rep1 library by analyzing the fraction of reads aligning to genomic categories and specific RNA regions. As depicted in **Reviewer Figure 1B**, we found a substantial reduction to approximately 13.99% in the fraction of reads aligning to rRNA which were originally 80%-90% of the total RNA in eukaryotic cells¹. This reduction serves as a clear indication of successful depletion. Moreover, the reads aligning to the selected RNAs (SNORA73A, SNORA73B, RMRP, and RPPH1) were effectively reduced to undetectable levels. These results provide compelling evidence that the RNase H-based RNA depletion technique is highly efficient in selectively depleting targeted RNAs, demonstrated by the remarkable reduction in rRNA aligning reads as well as the near complete elimination of the selected ncRNAs of interest.

Reviewer Figure 1: Removal efficiency of RNase H-based RNA depletion. **A** qRT-PCR analysis for high-abundant RNAs in HepG2 total RNA treated with RNase H or not. **B** Genomic distribution of reads in HepG2 rep1 library after RNase H treatment.

General point 2: The half-life assay of napRNAs. In the original manuscript, we identified novel napRNAs, including box C/D-napRNAs, box H/ACA-napRNAs, sliRNAs, snotrons and misRNAs which are mainly intronic RNA species. Reviewer #1 and #2 both raised the point that additional experimental evidence is required to validate the half-life of napRNAs. To examine napRNAs half-life, we treated HepG2 cells with 5 μ g/ml Actinomycin D (ActD) and then harvested RNA at specific time points to measure napRNAs abundance by qRT-PCR (**Reviewer Figure 2A-F; revised Supplementary Fig. 5b-f**). We observed that the half-life of sno-like napRNAs is approximately 2-4 hours (**Reviewer Figure 2A**), which is consistent with that of mRNA and SPA lncRNA containing both a 5' snoRNA and a 3' polyA tail (**Reviewer Figure 2F**). Comparatively, the half-life of sliRNAs were around 20 minutes to 1 hour, which were longer than that of ACTB intron which is rapid degraded (**Reviewer Figure 2B and 2C**). Snotrons had a half-life of 20 minutes to 2 hours, which were similar to or longer than that of corresponding snoRNAs (**Reviewer Figure 2D**). The half-life of misRNAs is approximately half that of many miRNAs (**Reviewer Figure 2E**).

Known functional RNA molecules demonstrate diverse half-lives. For example, current research on enhancer RNA (eRNA), which activates or enhances transcription, has shown that the half-life of eRNA is approximately ≤ 7.5 minutes². Furthermore, some regulatory long non-coding RNAs (lncRNA) have a half-life of less than 30 minutes. For instance, the well-characterized paraspeckle RNA NEAT1 has a half-life of less than 30 minutes based on array data and approximately 15 minutes according to qPCR³. The half-lives of many napRNAs are longer than those of eRNA and these lncRNAs, suggesting the potential functional roles of napRNAs.

Reviewer Figure 2 The half-life of napRNAs were detected by qRT-PCR assay in HepG2 cells with 5 μ g/ml Actinomycin D treatment. **A** The half-life of sno-like napRNAs. **B** and **C** The half-life of sliRNAs (**B**) and ACTB intron (**C**). **D** The half-life of snotrons and snoRNA. **E** The half-life of misRNAs and miRNA. **F** The half-life of mRNA (ZNF148 and UEB3A) and lncRNA (SPA).

General point 3: DINAP napRNA is a chimeric RNA composed of both a box C/D and a H/ACA structure. In our original manuscript we had proposed the biological roles of the novel napRNA DINAP, which interacted with DKC1 protein to promote cell proliferation by maintaining protein stability. Both Reviewer #1 and #2 raised concerns regarding the RNA species and the structure of DINAP, which form the foundation of the interaction between DINAP and DKC1. We conducted careful analysis on its sequence and structure and we have found that DINAP is composed of a box C/D and a box H/ACA snoRNA-like domains, suggesting that DINAP is a composite box C/D-H/ACA RNA (**Reviewer Figure 3**), like U85 and U87⁴. Considering that DINAP possesses a box H/ACA structure, it can interact with the DKC1 protein⁴.

Reviewer Figure 3 The highly stable secondary structure of DINAP composed of a box C/D and a box H/ACA snoRNA-like domains, suggesting that DINAP is a composite box C/D-H/ACA RNA.

General point 4: We developed NAP-SHAPE-MaP method to analyse the structures of napRNAs. Reviewer #3 raised concerns on the limited experimental evidence of the predicted napRNAs secondary structures. To address this concern, we developed NAP-SHAPE-MaP, an approach combining SHAPE-MaP⁵⁻⁹ and our NAP-seq experiments to probe intact RNA structures of napRNAs. NAP-SHAPE-MaP analysis showed that the increase in SHAPE reactivity was more significant in unpaired sequence than paired regions (**Reviewer Figure 4A; revised Supplementary Fig. 3f**), consistent with previous observations^{6,10}. We also obtained accurate structure reactivity for snoRNAs with known secondary or tertiary structure (**Reviewer Figure 4B; revised Supplementary Fig. 3g**). These results indicated the accuracy of NAP-SHAPE-MaP to examine the RNA structure. Novel napRNAs with a significantly conserved RNA secondary structure identified by RNAalifold software¹¹ were also verified by our NAP-SHAPE-MaP method (**Reviewer Figure 4C, D; revised Supplementary Fig. 3h, i**).

Reviewer Figure 4. NAP-SHAPE-MaP analyses reveal napRNA structure. A. NAP-SHAPE-MaP reactivity of the napRNA. X axis represent various secondary structure region of napRNA. Y axis represent the mean reactivity of NAP-SHAPE-MaP. **B.** The secondary structure model and NAP-SHAPE-MaP reactivity score for each base of SCARNA3 (left) and SNORA14B (right), with different colors representing different range of reactivity scores. **C.** Covariation structure of a specific napRNA, hsa-napRNA-1852 (chr3:193,155,217-193,155,403: -) and hsa-napRNA-2666 (chrX: 20194567-20194942: -). **D.** The secondary structure model and SHAPE-MaP reactivity score for each base of hsa-napRNA-1852 and hsa-napRNA-2666, with different colors representing different range of reactivity scores.

The following are our point-by-point responses to the comments/suggestions from the Reviewers.

Reviewer #1 (Remarks to the Author):

In this manuscript, Liu et al describe an experimental method (NAP-seq) designed to identify low abundance, uncapped transcripts (> 100 nt in size). This approach has allowed the identification of new RNA species, some of which can be classified as structurally related

families such as snoRNAs (box C/D and H/ACA snoRNA), linear intron RNAs (sliRNAs), snoRNA-intron (snotrons) and RNAs embedded in miRNA spacer regions (misRNAs). It is an interesting work and the reading of the manuscript is relatively pleasant. Although I consider this study as a preliminary descriptive work, expression data and loss-of-function experiments (CRISPR-mediated knockouts) suggest that some of these newly-identified RNA species, collectively named noncapped RNAs (napRNAs), may exert regulatory functions, possibly during myoblast differentiation or cell proliferation. However, I find that authors tend to over-interpret some of their observations. Indeed, they should discuss the trivial possibility that some (most) of the newly identified RNA species (low abundance, poorly conserved and overlapping interspersed repeated elements) could simply represent degradation products and/or RNA intermediates during pre-mRNA splicing or pri-miRNA processing. In this framework, that expression of some these ncRNAs vary according to biological conditions does not necessarily mean that they are functionally important. Finally, I recommend that the authors avoid the excessive use of words like "exceptional/exceptionally" which in my opinion does not bring anything scientifically (i.e., What does it mean? exceptional compared to what?). Before considering publication of this manuscript, I believe it is essential that the authors address the specific points listed below.

Response: We very much appreciate the Reviewer's thoughtful and constructive suggestions and comments, which are a great help for us to revise and improve the manuscript. According to reviewer's suggestion, we have discussed the trivial possibility of the newly identified RNA species in the revised manuscript (Discussion section). In addition, we have appropriately removed the words like "exceptional/exceptionally" in the revised manuscript.

1. >>>Lanes 59-60: "Using NAP-seq, we compiled the first human and mouse napRNA transcriptomes and demonstrated their evolutionary conservation as well as their response to different cellular conditions".

I find this sentence ambiguous. napRNAs can be detected in both humans and mice, but the authors do not show any functional and/or structural conservation between a given human and mouse napRNA. See also my other comment about hsa-napRNA-1642.

Response: Thank you for your constructive comments. Based on the reviewer's suggestions, we had provided the examples of structural conservation within napRNAs in the revised manuscript (**Reviewer Figure 6, revised Supplementary Fig. 3h**). This question, along with Question 5, pertains to the evolutionary conservation of napRNAs. For a detailed revision, please refer to my response to **Question 5 and Reviewer Figure 6**.

2. >>>Lanes 89-90: "To increase the diversity of napRNA species and discover low-expression napRNAs, we used the RNase H method to remove high-abundance rRNAs, snRNAs and snoRNAs (Fig. 1a)".

The list of antisense DNA oligos should be given. On the other hand, it would be useful to give more information on the efficiency of this depletion method, perhaps by giving the number of sequencing reads corresponding to the rRNA species. In other words, what is the percentage of remaining reads corresponding to ncRNAs that were initially destined to be degraded by RNase H treatment?

Response: Thank you for the constructive suggestions. We have added detailed information on

the list of antisense DNA oligos in the “**Methods**” and **Supplementary Table 10** in the revised manuscript. In addition, we have performed qRT-PCR assays to evaluate the removal efficiency of RNase H-based depletion (see “**General point 1**”). As depicted in **Reviewer Figure 1A (revised Supplementary Fig. 1b, c)**, we observed that this technique achieved a remarkable depletion efficiency of up to 99% for the targeted RNAs. Furthermore, we performed an assessment of the RNA composition in the RNase H-treated HepG2 rep1 library by analyzing the fraction of reads aligning to genomic categories and specific RNA regions. As depicted in **Reviewer Figure 1B**, we found a substantial reduction to approximately 13.99% in the fraction of reads aligning to rRNA which were originally 80%-90% of the total RNA in eukaryotic cells¹. This reduction serves as a clear indication of successful depletion. Moreover, the reads aligning to the selected RNAs (SNORA73A, SNORA73B, RMRP, and RPPH1) were effectively reduced to undetectable levels. These results provide compelling evidence that the RNase H-based RNA depletion technique is highly efficient in selectively depleting targeted RNAs, demonstrated by the remarkable reduction in rRNA aligning reads as well as the near complete elimination of the selected ncRNAs of interest.

3. >>>Lanes 129-130: “We further profiled the transcriptomes of 3 different stress responses (poly (I:C), adriamycin (ADR), and CoCl₂) in human HepG2 cells ...”.

I recommend that the authors give some additional information about the rationale of this experiment, for example what are the modes of action of the chemical compounds employed. This will make the reading easier for people not familiar with these molecules/treatments.

Response: We sincerely appreciate the valuable suggestions provided by the reviewer, as they have greatly contributed to the improvement of our paper. In response to these suggestions, we have incorporated the following additional information into the revised manuscript (**Result 2, Paragraph 1**). We have added the followings in the revised result 2, Paragraph 1:

In these stress responses, poly (I:C) mimics double-stranded RNA to induce the activation of immune response¹²; ADR causes DNA damage, triggering cells to activate their DNA damage repair machinery^{13, 14}; CoCl₂ artificially induces hypoxia¹⁵, which has allowed the characterization of the hypoxia response at the cellular, biochemical and molecular levels¹⁶.

4. >>> Lanes 134-136: “We further filtered the candidate napRNAs identified by napSeeker through requiring that the napRNA must exist in at least two sequencing libraries, have more than 20 sequencing reads and not overlap with known annotations, such as rRNAs, tRNAs, snoRNAs and CDSs (Supplementary Fig. 2a, see Methods).

Just out of curiosity, I looked at the list of newly identified snoRNAs, it turns out that mmu-novel-CD-1 is actually scaRNA28 (= ZL1). I did not check the other snoRNA candidates. It is important that the authors verify that the other snoRNA candidates do not correspond to already annotated ncRNAs.

Response: We appreciate the reviewer’s comments and have carefully examined the novel snoRNA-like napRNAs in relation to the latest GENCODE annotations (human: GRCh38.p13, Release 42; mouse: GRCm39, Release M31). Upon comparison, we found no intersection between the identified napRNAs and the GENCODE annotations. Therefore, we annotate it (mmu-novel-CD-1) as a new box C/D RNA.

We further conducted alignment analysis between mmu-novel-CD-1 and human SCARNA28. This analysis indicated that mmu-novel-CD-1 may indeed serve as the homolog of SCARNA28 in mice (**Reviewer Figure 5**). Interestingly, our NAP-seq identifies the full-length information of each napRNA with single-base precision, revealing that both the 5' and 3' ends of mmu-novel-CD-1 are distinct from the existing annotation. Based on the constructive comments from the reviewers, we have made revisions to **revised Supplementary Table 6** by adding a column indicating homology and designating 'mmu-novel-CD-1' as homologous.

```

mmu-novel-CD-1 1 CAC AATGATGAAC AATAC TG CTGGAGCC ACAGAGAGGCAC AAGTGTGTGTGCCAGATTGCC TGTG 65
SCARNA28 1 AAGCAAAGTGATGAGTAATAC TGCTGGAGCCCAAAGAGGCACGTGTGTGTGT TTTG TGTG 61

mmu-novel-CD-1 66 TGTATGTGTGTCTCTTGTCTGTGCATGCACACGTGAGTCTGGGAGGATGGATGTGTGTGACTGGTGGTAG 134
SCARNA28 62 TGTGTGTATAATGCTTGTCAAGTGCATGCACGTGTATGTC TGGGAGTACAAATGGGTGCGACTGGTGTAG 130

mmu-novel-CD-1 135 GGACCTAGCCC TGTGCC ATCTCTTGGGCTGTGACAGTCAAAC TGATAAGATCTGATT 191
SCARNA28 131 GGAAC TAGCTATGTGCC TTCTATTAGGCCATGACAGTCAAAC TGATAAGATCTGATTGCTTCTC 194

```

Reviewer Figure 5. Alignment of mmu-novel-CD-1 in mice and its homologous gene SCARNA28 in humans.

5. >>>Lanes 147-149: “One particularly interesting example is a new long napRNA (hsa-napRNA-1642) that has a significantly conserved RNA secondary structure identified by evaluating pairwise covariations using R-scape software 20 (Supplementary Fig. 2f)”.

If the authors wish to emphasize that some napRNAs are conserved through evolution, and to exploit the presence of compensatory base change as evidence for the conservation of their 2D structures, then the authors should compare RNA structures between phylogenetically distant species. It would indeed be more informative, and especially more convincing, to examine napRNAs that originated more than 300 Myr ago, as mentioned but not shown in Supplementary Fig. 2g. Indeed, as I understand, hsa-napRNA-1642 is only found in primates and it is not indicated which species were used to identify co-variations in the Supplementary Fig. 2f. In addition, I do not see any “covarying mutations” as indicated by the green code. In my view, it is important that the authors show sequence alignments and compensatory base changes supporting the presence of stems.

Response: Thank you for your valuable suggestion. We have included the presentation of napRNA structures originating from phylogenetically distant species, with co-varying mutation pairings, in our revised manuscript (**Reviewer Figure 6A and 6B**). To ensure easy accessibility to the structural information of each napRNA, we have created a dedicated webpage called “napRNA structure”. This webpage (accessible at <https://rnasysu.com/napSeq/structure.php>) offers comprehensive information, including sequence alignments, structures, conservation scores, and the species involved in identifying co-variations. By incorporating these enhancements, we aim to facilitate a clearer understanding of the structural characteristics of napRNAs among diverse species. These additions have been made to the revised manuscript, specifically in the **revised Result 2, Paragraph 2**, to further enrich the information available to readers.

Reviewer Figure 6. Some napRNAs are conserved through evolution. **A** The RNA structure of hsa-napRNA-1852 originating from phylogenetically distant species (before 300 Myr) with co-varying mutations pairings. **B** The RNA structure of hsa-napRNA-2666 originating from phylogenetically distant species (before 300 Myr) with co-varying mutations pairings. The coevolving base sequences are annotated with a green background pattern.

6. >>>Lanes 153-155: “We discovered that most (14284) napRNA families were primate-specific, but 3313 families likely originated more than 90 million years (Myr) ago, and 186 families likely originated more than 300 Myr ago (Supplementary Fig. 2g)”.

Do the most conserved sequences correspond to intronic sequences and/or intergenic sequences? I think it would be useful to give more information about the most phylogenetically conserved napRNAs, and therefore more likely to perform functions retained by natural selection.

Response: We express our gratitude for the valuable suggestions provided by the reviewer. In line with your suggestion, we have made extensive efforts to provide comprehensive details regarding the most phylogenetically conserved napRNAs. Notably, the highly conserved sequences predominantly align with intronic and intergenic regions (**Reviewer Figure 7A; revised Supplementary Fig. 4a**). For instance, hsa-napRNA-2120 derived from intron region is most phylogenetically conserved (**Reviewer Figure 7B; revised Supplementary Fig. 4b**). To gain preliminary insights into the potential functions associated with these highly conserved napRNAs, which originated more than 300 million years ago, we conducted Gene Ontology (GO) enrichment analysis on the host genes of these napRNAs. This analysis revealed that these host genes primarily participate in cell morphogenesis and skeletal system development (**Reviewer Figure 7C; revised Supplementary Fig. 4c**). By incorporating these additional details, we aimed to provide a more comprehensive understanding of the most conserved napRNAs and their potential functional implications.

Reviewer Figure 7. Phylogenetically conserved napRNAs. A Simplified phylogenetic tree of napRNAs by species. The tree tips indicate the number of napRNAs in each species. The pie plots indicate the genomic distribution of napRNAs with varying degrees of conservation. **B** Sequence alignment for conservation analysis of hsa-napRNA-2120 (https://rnasysu.com/napSeq/structure_browser.php?organism=human&assembly=hg38&protein=&factor_type=napRNA&sample_id=hsa-napRNA-2120). **C** Top enriched Gene Ontology (GO) enriched with the host genes of the novel napRNAs originated more than 300 Myr ago.

7. >>>Lanes 164-168: “Interestingly, we discovered that 1620 differentially expressed napRNAs displayed significant changes in expression for up to 72 hours (72 h, D72) after C2C12 cells were induced to differentiate and maintained relatively stable expression until 144 h (D144) after induction of differentiation (Supplementary Fig. 3d, 167 e and Table 3), suggesting that napRNAs may play crucial roles at the early stage of skeletal muscle differentiation”.

I am not familiar with the differentiation pathways in C2C12 cells. I imagine that there are already gene expression data available. Do the differentially expressed napRNAs correspond to portions (intronic?) of transcripts from genes already known to vary in expression after induction of skeletal muscle differentiation? Or are these differentially expressed napRNAs generated from independent (intergenic?) transcription units? Finally, and more importantly, that expression levels of some napRNAs vary does not imply functionality and, without functional validations, I find premature to propose a “crucial role at the early stage of skeletal muscle differentiation”.

Response: Thank you for your suggestions. Following your constructive suggestions, we have provided detailed information on the differentially expressed genes (**revised Supplementary Table 4**). Additionally, we have depicted their genomic distribution (**Reviewer Figure 8A, revised Supplementary Fig. 6g**). Among the total 6808 napRNAs, 57.6% originate from intron regions, 39.42% are generated from repeat-elements, and 1.65% are from intergenic regions (**Reviewer Figure 8A; revised Supplementary Fig. 6g**).

To investigate whether changes in napRNAs expression during development correspond to changes in host genes, we quantified the expression of napRNA host genes using RNA-seq. We observed a rapid increase in the expression of myogenic markers such as Mef2c, Mef2a, and Myog once C2C12 cells were induced to differentiate (**Reviewer Figure 8B**). Furthermore, the fold change of most napRNA expression during C2C12 differentiation showed weak correlation with their host genes (**Reviewer Figure 8C-E; revised Supplementary Fig. 6h-j**).

We agree with the reviewer’s point that the variability in napRNA expression levels does not necessarily imply functionality. Interestingly, these napRNAs with variable expression can serve as potential candidates for experimental verification and functional validation. For example, in the **revised Supplementary Fig. 11**, we demonstrate that downregulation of mmu-novel-CD-6 inhibits skeletal muscle differentiation during myoblast differentiation through gene gain-of-function or loss-of-function experiments.

According to the reviewer's suggestion, we have removed the sentence “suggesting that napRNAs may play crucial roles at the early stage of skeletal muscle differentiation” from the revised manuscript.

Reviewer Figure 8 Differentially expressed napRNAs during C2C12 differentiation. **A** Genomic distribution of differentially expressed napRNAs in mouse C2C12 cells during developmental stages. **B** Heatmap show the expression of representative genes already known to vary in expression after induction of skeletal muscle differentiation. **C**, **D** and **E** Scatterplot show the correlation of differential gene changes between NAP-seq (napRNAs) and RNA-seq (host mRNAs of napRNAs). D72 represents for C2C12 cells induced to differentiate for 72h; D96 represents for C2C12 cells induced to differentiate for 96h; D144 represents for C2C12 cells induced to differentiate for 144h.

8. >>>Lanes 170_184

I am not sure I understood the point of this "epigenetic section". What is the message that the authors wanted to deliver? Are we to understand that some napRNAs are generated independently from their own promoters? I think it is necessary to conclude this paragraph more precisely.

Response: We sincerely appreciate the insightful suggestions provided by the reviewer. It is widely acknowledged that histone modifications play a crucial role in various chromatin-dependent processes, including transcription¹⁷⁻¹⁹. Regions exhibiting histone modifications are generally associated with high accessibility, facilitating transcription and gene expression. In our study, we conducted an analysis of the chromatin state of napRNAs to gain insights into their expression patterns and provide additional evidences to demonstrate that napRNAs are transcribed. Additionally, histone modifications such as H3K27Ac and H3K4me3 are commonly associated with the transcription activation of nearby genes²⁰. Therefore, napRNAs marked with these two histone modifications are potential transcriptional enhancers (e.g. enhancer RNAs) bound by mediators, facilitating enhancer-promoter crosstalk.

We apologize for any ambiguity in the initial description of this section and have taken your feedback into consideration. In response, we have refined and provided a more precise description in the revised manuscript, specifically in **Result 2, Paragraph 5**.

9. >>> Lanes 206 – “We next developed new bioinformatic pipelines to identify novel small nucleolar RNAs (snoRNAs), including C/D box and H/ACA box RNAs, from NAP-seq profiles (Supplementary Fig. 5a)”.

As indicated by their names snoRNAs accumulate in the nucleoli. As I understand it, the sub-cellular localization of these putative snoRNAs has not been experimentally tested. I advise the authors to remain more factual and write RNA species containing sequence motifs found in C/D and H/ACA snoRNAs. In this context, are the new snoRNA-like conserved between human and mouse? I also find it unfortunate that the authors do not show the interaction of these putative “snoRNAs” with fibrillarin and/or Dyskerin. This could be done easily by immunoprecipitation experiments.

Response: We greatly appreciate the reviewer for bringing this to our attention. In our study, we have identified 1115 novel snoRNA-like napRNAs in humans and 198 novel snoRNA-like napRNAs in mice. Interestingly, only 9 snoRNA-like napRNAs are conserved between humans and mice (**Reviewer Figure 9A; revised Supplementary Fig. 10j**). This conservation pattern observed aligns with the fact that most of these snoRNA-like napRNAs originate from species-specific elements, such as Alu elements in humans and B1/B2/B3 elements in mice (**Reviewer Figure 9B-E**). Furthermore, we have taken your suggestion into account and randomly selected 5 snoRNA-like napRNAs for RIP-qPCR experiments to investigate the interaction between snoRNA-like napRNAs and the well-known snoRNA-binding proteins, FBL and DKC1. These experiments successfully identified a group of snoRNA-like napRNAs that interact with FBL or DKC1 proteins (**Reviewer Figure 9F-G; revised Supplementary Fig. 9h; revised Supplementary Fig. 10i**). We have provided a detailed description of these findings in the revised manuscript.

Reviewer Figure 9. SnoRNA-like napRNAs in humans and mice. **A** Intersection of snoRNA-like napRNAs identified from humans and mice. **B and C** Distribution of candidate human CD-napRNAs (**B**) or H/ACA-napRNAs (**C**) in annotated gene types. **D and E** Distribution of candidate mouse CD-napRNAs (**D**) or H/ACA-napRNAs (**E**) in annotated gene types. **F and G** The interaction between the FBL (**F**) or DKC1 (**G**) proteins and the novel snoRNA identified by NAP-seq were confirmed by RIP-qPCR in HepG2 cells, respectively.

10. >>>Lanes 228-230 - Surprisingly, compared with the known AluACA RNAs that cover only approximately 25% of the whole length of the host Alu element, full-length AluACA RNAs could be identified by our NAP-seq method (Supplementary Fig. 7h), and most of these were found to cover more than 50% of the Alu element length (Fig. 2g)''.

I am not sure I understand. Does this imply that the previously published AluACA RNAs result from the processing of longer RNA precursors? Or did the authors detect a new class of AluACA RNAs?

Response: Thank you for the reviewer's comments, and we apologize for any ambiguity in our previous description. In our study, we have successfully identified a novel class of Alu-ACA

RNAs on a large scale, which exhibits significant differences compared to previously discovered Alu-ACAs. In particular, the proportion of sequence length originated from Alu elements within the novel ACA RNAs is markedly higher.

To accurately reflect this distinction, we have revised the description in the revised manuscript (**Result 3, paragraph 3**) as follows: “Surprisingly, our NAP-seq method has unveiled a novel group of Alu-ACA RNAs (**revised Supplementary Fig. 10h**) that display a distinct feature. Unlike the known Alu-ACA RNAs that cover approximately 25% of the entire length of the host Alu element²¹, we have discovered that this novel class of Alu-ACA RNAs spans more than 50% of the length of the Alu element (**revised Fig. 2g**).”

11. >>>Lanes 239-241 – “We also detected the subcellular localization of mmu-novel-CD-6 by fluorescence in situ hybridization (FISH) and found that it was localized mainly in the nucleus (Supplementary Fig. 8e)”.

It would be interesting to provide more details on the nucleoplasmic localization of CD-6. It seems that RNA FISH signals do not correspond to nucleoli. Do these signals correspond to the SC-35 domains (nuclear speckles)? Given that CD6 is embedded within and processed from an intron and because its sequence overlaps a repeated (L2) element, how do the authors know that FISH signals originate from the fully processed CD36, and not from the unspliced (host) pre-mRNA, or even from other L2 elements generated elsewhere in the genome?

Response: We appreciate the reviewer’s constructive comments, and we fully acknowledge their validity. In response, we have conducted further experiments to investigate the subcellular localization of mmu-novel-CD-6. Specifically, we performed FISH (Fluorescence In Situ Hybridization) using the nucleolus marker FBL and the nuclear speckles marker SC-35 domains to examine the localization of mmu-novel-CD-6 within the cell nucleus (**Reviewer Figure 10; revised Supplementary Fig. 11e**).

The results demonstrated that mmu-novel-CD-6 exhibited co-localization with both the FBL marker, indicative of the nucleolus, and the SC-35 domains marker associated with nuclear speckles. It is important to note that, with the current FISH method employed, we were unable to distinguish whether the FISH signal emanated from the mature transcripts or the host gene (pre-mRNA) because of the overlapping sequences between the mature transcripts and its host gene. However, currently, there is no research demonstrating the interaction between the pre-mRNA of the mmu-novel-CD-6 host gene and FBL. Therefore, our results are more likely to suggest subcellular co-localization between mmu-novel-CD-6 and FBL protein.

The probes we utilized in FISH are specifically designed for mmu-novel-CD-6. Upon alignment, these probe sequences do not exhibit alignment to any other locations in the genome. Consequently, the FISH signals are not originating from other L2 elements or genes located elsewhere in the genome.

Reviewer Figure 10. Subcellular localization of mmu-novel-CD-6 (yellow), as assessed by smiFISH. Nuclear DNA was stained with DAPI (blue). Simultaneously, FBL protein (green)

and SC-35 domains (red) was stained by immunofluorescence (IF).

12. >>>Lane 245-248 ... “found that its overexpression had no influence on the mRNA level of its host gene (Exosc2) (Supplementary Fig. 8g) and significantly decreased the protein expression levels of the established myogenic markers MHC and Mef2c (Supplementary Fig. 8h), suggesting that mmu-novel-CD-6 inhibits myoblast differentiation”.

I am worried about the biological relevance of this type of experiment where the authors introduce supraphysiological doses of a given RNA ? In other words, what is the relative abundance of the transfected (ectopic) form of CD6 vs the endogenous one? The authors should introduce a negative control such as an inactive form of CD6 bearing mutations in the C- and/or D-box. Indeed, this mutated version of CD6 is not expect to accumulate. Finally, how do the authors envision CD6 inhibiting myoblast differentiation? Although I am aware that answering this question is beyond the scope of this study, it would still be interesting to mention some hypotheses in the discussion section.

Response: We appreciate the reviewer’s constructive suggestions, and we have thoroughly addressed them in our study. In comparison to the endogenous expression, the ectopic expression of mmu-novel-CD-6 was found to be approximately 300-fold (**Reviewer Figure 11A; revised Supplementary Fig. 11g**). In line with the reviewer’s recommendation, we introduced a mutation in the D box of mmu-novel-CD-6 (changing the CUGA motif to CUAG) and transfected this mutant into C2C12 cells. As a result, the mutant mmu-novel-CD-6 could not be properly processed, which did not affect the expression of Exosc2, and had no impact on myoblast differentiation (**Reviewer Figure 11A; revised Supplementary Fig. 11g**). Moreover, we conducted a knockdown of mmu-novel-CD-6 expression using antisense oligonucleotides (ASOs), which led to the upregulation of myogenic markers MHC and Mef2c (**Reviewer Figure 11B; revised Supplementary Fig. 11h**). These results suggest that mmu-novel-CD-6 plays a role in inhibiting myoblast differentiation.

Given that mmu-novel-CD-6 is a newly discovered CD-napRNA containing conserved C box (AUGAUGA) and D box (CUGA) motifs, and we have observed its co-localization with FBL in the nucleus through FISH and immunofluorescence analysis (**Reviewer Figure 11; revised Supplementary Fig. 11e**). Our initial hypothesis was that mmu-novel-CD-6 might function similarly to known snoRNAs in guiding RNA modification, such as 2’-O-methylation, to regulate target gene expression and myoblast differentiation. However, our intermolecular interaction analysis did not reveal any genes with significant proximity interaction with mmu-novel-CD-6 by using CLASH data from our ENCORI platform (<https://rnasysu.com/encori/index.php>). Therefore, we speculate that mmu-novel-CD-6 is unlikely to function as a snoRNA guiding 2’-O-methylation during myoblast differentiation. We have incorporated this section into the revised discussion (**revised Paragraph 4**).

Reviewer Figure 11. The expression of mmu-novel-CD-6 and its host gene (Exosc2) and the myogenic markers MHC and Mef2c were tested by qPCR and WB in mmu-novel-CD-6-overexpressing cells and mmu-novel-CD-6 mutant with D box mutation-overexpression cells (A), and mmu-novel-CD-6 knockdown cells (B).

13. >>>Lanes 258 – “We analysed our NAP-seq reads located within human introns and discovered that thousands of 5’-start and 3’-end sequencing reads were precisely located within the 5’ splice site (5’-SS) of 6,448 introns or the 3’-SS of 17,397 introns in HepG2 cells (Fig. 3a, b), implying that these introns may be stable in linear form rather than rapidly degraded”.

From my point of view, that intronic RNA species can be detected does not mean necessarily that these latter correspond to stable RNA species. Did the authors experimentally demonstrate their intrinsic stability by treating cells with high doses of Actinomycin D?

Response: We appreciate the reviewer to point it out and agree with the reviewer’s comments. To examine napRNAs half-life, we treated HepG2 cells with 5 µg/ml Actinomycin D (ActD) and then harvested RNA at specific time points to measure sliRNAs abundance by qRT-PCR (**Reviewer Figure 2B**; revised **Supplementary Fig. 4c**). Comparatively, the half-life of sliRNAs were around 20 minutes to 1 hour, which were longer than that of ACTB intron which is rapid degraded (**Reviewer Figure 2B-C**). Simultaneously, we also assess the half-life of other napRNAs. We observed that the half-life of sno-like napRNAs is approximately 2-4 hours (**Reviewer Figure 2A**). Snotrons had a half-life of 20 minutes to 2 hours, which were similar to or longer than that of snoRNAs (**Reviewer Figure 2D**). The half-life of misRNAs is approximately half that of many miRNAs (**Reviewer Figure 2E**).

Known functional RNA molecules demonstrate diverse half-lives. For example, current research on enhancer RNA (eRNA), which activates or enhances transcription, has shown that the half-life of eRNA is approximately ≤ 7.5 minutes². Furthermore, some regulatory long non-coding RNAs (lncRNA) have a half-life of less than 30 minutes. For instance, the well-characterized paraspeckle RNA NEAT1 has a half-life of less than 30 minutes based on array data and approximately 15 minutes according to qPCR³. The half-lives of many napRNAs are longer than those of eRNA and these lncRNAs, suggesting the potential functional roles of

napRNAs. (Discussion, Paragraph 3).

14. >>>Lanes 280-283 – “In addition, we found that the expression of sliRNAs was dynamically changed during mouse myoblast differentiation; sliRNAs generally exhibited high expression at the late stage of myoblast differentiation (Supplementary Fig. 10e)”.

Could it be that this simply reflects changes in post-transcriptional regulation of their host-gene transcripts?

Response: Thank you for the reviewer’s valuable comments. To explore the relationship between changes in napRNAs and their host genes during development, we conducted quantification of the expression levels of napRNA host genes using mRNA-seq analysis. We observed a rapid increase in the expression of myogenic markers such as Mef2c, Mef2a, and Myog once C2C12 cells were induced to differentiate (**Reviewer Figure 8B**). Furthermore, the fold change of most napRNA expression during C2C12 differentiation showed weak correlation with their host genes (**Reviewer Figure 8C-E; revised Supplementary Fig. 6h-j**).

These findings indicate that the changes observed in napRNAs during development may not always be directly linked to alterations in their host genes. It suggests the presence of distinct regulatory mechanisms for these two molecular components.

15. >>>Lanes 320 – 322 - “Moreover, through evolutionary conservation analysis, we observed that the intron sequences of hsa-snotron-1 which contained SNORD2 were highly conserved in mammals (Supplementary Fig. 11c), implying that hsa-snotron-1 may play a crucial biological role as a whole snotron and not only as a snoRNA”.

I am very surprised by the Northern blot shown in Figure 4G in which the expression level of snotron-1 appears to be of the same order of magnitude as that of fully processed SNORD2, or even higher than that of spliceosomal snRNA U6. Is this also the case for the other snotrons? I also suggest the authors to quote, and discuss accordingly the following publications: PMID: 34725166 and 35878033

Response: We appreciate the professional comments and advice provided. The preliminary examination of snotron-1 and U6 expression levels was conducted using qRT-PCR, which indicated that the expression of snotron-1 was lower than that of U6. To confirm the presence of snotron-1, we performed an inNorthern blot analysis using gene-specific probes labeled with IRDye800 for snotron-1 and SNORD2, and IRDye680 for U6. Visualization was carried out using the Odyssey system. In order to prevent strong signals from high-abundance genes, such as U6, from interfering with the signal of target genes, we used a limited number of probes for hybridization. As a result, the band signal intensity could not be used to compare the RNA levels among genes. To prevent readers from encountering the same confusion, we have revised the plotting of the revised Fig. 4g (**reviewer Figure 12**).

Reviewer Figure 12 Verification of hsa-snotron-1 by irNorthern blotting in HepG2 cells.

To investigate the expression levels of snotrons and their snoRNA counterparts, we employed PEN-seq sequencing data to calculate the expression of snoRNAs in HepG2 (GSE160887) ²². We addressed the batch effect, corrected the library depth, and normalized the gene length to compare the snoRNAs with snotrons. Our findings revealed that the expression levels of all snotrons were lower than those of their snoRNA counterparts (**Reviewer Figure 13; revised Supplementary Fig. 14e**).

Based on the reviewer's suggestion, we quoted and discussed the publications: PMID 34725166 and 35878033 in the revised discussion section. Although snotron, slb-snoRNA, and hmsnoRNA all possess a snoRNA sequence, they are distinct RNA species generated through different RNA splicing processes. Slb-snoRNA is a circular lariat (sisRNA) harboring a snoRNA ²³, which is formed due to a branching failure of a lariat intron. HmsnoRNA, generated by splicing inactivation, undergoes processing mediated by the nuclear exosome to yield mature 3' ends of the snoRNAs, which are subsequently bound by small nucleolar ribonucleoproteins ²⁴. Snotrons, identified through NAP-seq, represent a novel class of snoRNA-intron napRNAs. One end of these molecules corresponds to the 5'/3'-end position of an intronic snoRNA, while the other end corresponds to the 3'/5'-end of an intron (**revised Fig. 4a, b**). These unique RNA molecules indicate that abnormalities at various stages of RNA processing can give rise to different types of RNA products. We have incorporated discussions about the publications (PMID 34725166 and 35878033) into the revised manuscript (**Discussion, Paragraph 6**).

Reviewer Figure 13 Comparison of the snoRNA and snotron expression.

16. >>>Lanes 336-340–“We further found that a number of RBPs potentially interact with the 3'-ends of the snotrons (Supplementary Fig. 11j). Many RBPs showed different binding preferences between snotrons and other introns containing snoRNAs. Notably, AQR showed the greatest difference among all RBPs (Supplementary Fig. 11j). Given that AQR interacts with many splicing factors (Supplementary Fig. 11k) and may facilitate the folding of snoRNA sequences and snoRNP biogenesis, we speculated that the AQR protein may contribute to the biogenesis and stability of snotrons”.

As I understand it, these are only in silico predictions. I find this section very speculative and unconvincing. From my point of view, this information can be removed without altering the message the authors wish to deliver. Alternatively, the authors could perform immunoprecipitation experiments. Given the expression level of snotron-1, it should not be difficult to test the interaction of AQR. In the same vein, it would also be very interesting to test if the core SNORD binding proteins (Snul3, Fibrillarin, Nop56 and Nop58) associate with

snotron-1.

Response: We agree with the reviewer and remove the analysis in **Supplementary Fig. 11j and 11k**. The predictions and insights regarding the potential interaction of RBPs with the 3'-ends of snotrons were included only as a basis for further research. Additionally, in RIP-qPCR of Fibrillarin (FBL), we observed a significant enrichment of snotron-1 and snotron-2 with FBL (**Reviewer Figure 14, revised Supplementary Fig. 14c**). This finding suggests a potential interaction between snotron-1, snotron-2, and FBL in HepG2 cells.

Reviewer Figure 14 The interaction between the FBL and snotrons containing C/D box snoRNA sequences was confirmed by RIP-qPCR in HepG2 cells.

17. >>>Lanes 361-...Regarding napRNAs processed from miRNA clusters (misRNAs) - Although I am not really convinced by the notion that misRNAs could have a regulatory function, I find interesting the possibility to propose an order in the maturation pathway for clustered miRNAs. In this respect figures 15e-f would deserve to be highlighted and discussed in the full manuscript rather than in the supplementary data.

Response: We greatly appreciate the reviewer highlighted this intriguing result. In the revised manuscript, we have emphasized and discussed on this result within the body of the manuscript (**revised Fig. 5g, h**).

18. >>>Regarding DINAP (DKC1-interacting napRNA)

I noted that DINAP could perhaps be considered a C/D box-containing RNA since it has a correctly positioned C- and D-box, with the possibility of forming a canonical 5'-3' terminal stem like those found in canonical snoRNAs involved in rRNA methylation (only the D box differs slightly from consensus; CAGA instead of CUGA). I am surprised that the authors do not mention this possibility.

Response: We would like to express our gratitude to the reviewer for bringing up this issue and you are highly knowledgeable and proficient in the field of C/D box RNAs. After careful analysis of the sequence and structure features of DINAP, we have found that DINAP is a chimeric napRNA composed of a box C/D and a box H/ACA snoRNA-like domains, suggesting that DINAP is a composite box C/D-H/ACA RNA (**Reviewer Figure 3**), like previous chimeric ncRNAs, including U85 and U87⁴.

19. >>> Lanes 407-409 “In addition, we further showed that DINAP was colocalized with DKC1 in the nucleoplasm by using fluorescence in situ hybridization (FISH) (Fig. 6d)”.

The authors should recall that DCK1 is an abundant nucleolar protein present in small nucleolar ribonucleoprotein particles (H/ACA RNPs) that convey specific uridines from ribosomal RNA into pseudouridines. Thus, I am very surprised by the IF signals that apparently do not visualize nucleoli. Can the authors comment on this rather unexpected observation? Is this the case with different antibodies? In any case, the authors should include a nucleolar marker as a control (e.g., fibrillarin). Can the authors demonstrate an interaction of DINAP with endogenously expressed DCK1? As I understand it, the interaction was only validated after overexpression of a tagged version of DCK1.

Response: We sincerely appreciate these valuable suggestions. In the revised manuscript, we have conducted an analysis on the subcellular localization of DINAP and DKC1 using the nucleolus marker FBL in both wild-type and DKC1 knock-down HepG2 cells (**Reviewer Figure 15A-B, revised Fig. 6d**). In the DKC1 knock-down cells, we observed a decrease in the signal of DKC1 protein compared to the control cells. Furthermore, by FISH and immunofluorescence (IF) techniques, we found that DINAP was co-localized with both DKC1 and FBL proteins within the cell nucleus.

Additionally, we performed DKC1-RIP-qPCR in wild-type HepG2 cells and made an interesting observation that endogenous DKC1 significantly enriched DINAP, indicating a potential interaction between endogenous DKC1 and DINAP (**Reviewer Figure 15C; revised Fig. 6e**). This finding supports the interaction between DKC1 and DINAP. DINAP containing the H/ACA motif further supports the conclusion that DINAP interacts with DKC1 protein. We have included these results in the revised manuscript.

Reviewer Figure 15. DINAP interacted with DKC1 protein in HepG2 cells. **A** The expression level of DKC1 was examined by qPCR and western blotting in DKC1-knockdown HepG2 cells. **B** Subcellular localization of DINAP (yellow) as assessed by smiFiSH, and of DKC1 (red) as assessed by immunofluorescence staining. Nuclear DNA was stained with DAPI. FBL (green) was as the nucleolus marker. **C** The interaction between DINAP and the DKC1 protein was confirmed by RIP-qPCR in lysates of HepG2 cells.

20. >>>Lane 428 - “These results revealed that DINAP regulates the stability of the DKC1 protein” -How do the authors imagine that DINAP regulates the stability of DCK1? I find the model proposed in figure sup17i ambiguous because it suggests that DINAP is much more abundant than DKC1, which in my opinion may not be the case. Can the authors argue, at least estimate the relative stoichiometry of the two partners likely to interact.

Response: We acknowledge the reviewer’s point and have rectified the ambiguity in the model picture in revised **Supplementary Fig. 20i (Reviewer Figure 16)**. By careful analysis of the sequence and structure features of DINAP, we have found that DINAP is a chimeric napRNA composed of a box C/D and a box H/ACA snoRNA-like domains, suggesting that DINAP is a

composite box C/D-H/ACA RNA (**Reviewer Figure 3**), like previous chimeric ncRNAs, including U85 and U87 ⁴. Given that the interaction between DINAP and DKC1 protein, we hypothesize that this interaction may promote the involvement of DKC1 in specific protein complexes, such as the formation of snoRNPs, thereby playing a pivotal role in its stability and functionality.

To assess the relative stoichiometry of DKC1 and DINAP, we performed quantification of their RNA levels using qRT-PCR. Our results indicate that the ΔCT value of DKC1 to GAPDH in HepG2 cells is approximately 4.16, while the ΔCT value of DINAP to GAPDH is about 10.99 (**Reviewer Figure 17**). This suggests that the RNA level of DKC1 is higher, implying a relatively higher protein level as well.

Reviewer Figure 16: A working model of DINAP.

Reviewer Figure 17: The relative expression of DINAP and DKC1 by qRT-PCR.

21. >>> Lane 485 – “these novel snoRNAs derived from repetitive elements are not associated with rRNAs and snRNAs, suggesting that they may perform novel regulatory functions”.

I find this sentence ambiguous. Did the authors look for potential base pairing interactions between napRNAs and rRNA/U-snRNAs? What could be the mode(s) of action of these C/D and H/ACA-like small RNAs?

Response: We appreciate the reviewer’s comments, which have helped us to improve the clarity of our research, and apologize for the confusing description. We did explore the possibility of potential interactions between napRNAs and rRNAs/snRNAs using snoSeeker software;

however, we did not find strong evidence of extensive base pairing interactions between snoRNA-like napRNAs and known rRNAs/snRNAs with RNA modification sites. As a result, we speculate that these orphan napRNAs may target other types of RNAs, and serve distinct functions. For instance, orphan napRNAs can function as scaffolds facilitating interactions with proteins, which warrants further investigation. We have included a more detailed explanation in the revised manuscript to address this point (**revised Discussion, paragraph 4, Line 574-584**).

22. >>> Lane 524-525 - “In this study, we discovered by the NAP-seq method that many miRNA clusters produce stable miRNA spacer sequences (i.e., misRNAs) in various cells”.

As I mentioned earlier for other napRNAs, there are no experimental data indicating that misRNAs are stable.

Response: We appreciate the comments. To validate the steady-state of the misRNAs, we performed the RNA half-life assay for misRNAs by treating HepG2 cells with Actinomycin D (5µl/mg). As shown in **Reviewer Figure 2E (revised Supplementary Fig. 5e)**, we observed that the half-life of misRNAs is half to that of many miRNAs. Based on the reviewer’s feedback, and to improve the paper’s rigor, we have removed the word “stable” from this section in the revised manuscript.

Reviewer #2 (Remarks to the Author):

In this manuscript Liu and colleagues apply a new RNA-seq method to be able to characterize cellular transcripts that lack 5’cap and/or 3’PolyA and therefore are not detected by standard sequencing methodologies. They use both short-read (Illumina) and long-read (Nanopore) RNA sequencing in order to obtain more information about the transcripts, and apply it to several human cell lines with different treatments and different differentiation stages of the mouse skeletal muscle cell line C2C12. With this approach, they identify a number of previously non-annotated transcripts that they name napRNAs, and that in many cases contain repetitive elements. The authors identify several significant types of transcripts, like PolIII-transcribed RNAs, RNAs containing snoRNA sequences, linear introns, snoRNA-intron or containing miRNA clusters. Many have in common being highly structured and show changing levels upon stress or differentiation. Finally, the authors show some experimental data on one of these napRNAs, named DINAP, which stabilizes DKC1 protein in HepG2 cells.

This is interesting work, presenting a novel methodology that allows to have a better view of the different RNA species that are present in the cells. However, it remains to be shown whether the identified RNAs are not intermediaries of processing without a specific function, therefore not biologically relevant.

Response: We highly appreciate the positive feedback given by the reviewer on our manuscript. In response to the valuable suggestions and comments put forward by the reviewer, we have conducted additional data analysis and experiments to enhance the quality of our work.

More specifically:

1. Although central to the paper, the way data are analyzed is not sufficiently explained in the main text. For instance, how are the long-read and short-read data integrated? The use of LR-seq is a very positive aspect of the study, however it only has been applied to one cell line. How many of the findings are supported by LR-seq? This is not sufficiently clear.

Response: We appreciate the reviewer's comments and apologize for any unclear descriptions in the previous version of the manuscript. We have made the following improvements based on the reviewer's comments:

- (1) We have conducted an analysis to assess the correlation between the NGS and TGS libraries (**Reviewer Figure 18A; revised supplementary Fig. 1d**) as well as the genomic distribution of reads (**Reviewer Figure 18B; revised supplementary Fig. 1e**). These analyses demonstrate that the two libraries exhibit similar patterns.
- (2) The majority of our identified results display the read start, end, and coverage tracks of TGS, which are consistent with the corresponding results obtained from NGS. These findings are illustrated in the genome browser (e.g., **Reviewer Figure 18C; revised Fig. 1i**).
- (3) To further support our findings, we have performed NAP-seq-TGS experiments in mouse myoblast cells C2C12, which exhibited similarities to the results obtained in the human cell line-HepG2 (**Reviewer Figure 18D-F; revised Supplementary Fig. 1f, g and revised Supplementary Table 1**).
- (4) Additionally, as suggested by the reviewer, we have conducted a global analysis and found that a large proportion of the results obtained from NAP-seq-NGS are supported by NAP-seq-TGS experiments conducted in both humans and mice (**Reviewer Figure 18E, F, revised Supplementary Fig. 1g**).

Overall, we performed NAP-seq-NGS and NAP-seq-TGS analyses in human cell line HepG2 and mouse cell line C2C12. Furthermore, the data shows a robust correlation. In both HepG2 and C2C12 cells, approximately 90% or more of the discoveries in NGS libraries are substantiated by TGS data, underscoring the robustness of the NAP-seq methodology and the findings. We thank you very much for reviewer's constructive suggestions that strengthened our manuscript.

Reviewer Figure 18. NapRNAs identified by NAP-seq-NGS were supported by NAP-seq-TGS. **A** Scatterplot show the correlation between reads obtained by the NAP-seq-NGS and NAP-seq-TGS (rep1, replicate1). Read numbers, RPM, reads per million. **B** The distribution and the percentage of NAP-seq-NGS and NAP-seq-TGS reads in annotated gene types. **C** Genome Browser visualization of NAP-seq 5'-start, 3'-end and coverage signals (RPM, reads per million), which were generated from NAP-seq-NGS and NAP-seq-TGS data, in a 5000-bp region of a human snoRNA cluster (chr17:7,573,750-7,578,750). **D** Workflow of napRNA identification in NAP-seq-NGS and NAP-seq-TGS. **E and F** Pie plot show that the napRNAs identified by NAP-seq-NGS were supported by NAP-seq-TGS in humans (**E**) and mice (**F**).

2. The finding of many repeated sequences is very intriguing. Is it possible that this is due to some technical artefact? The authors should provide additional analyses and evidences to exclude this possibility.

Response: We appreciate the reviewer's comments. We would like to address the concerns raised:

- (1) We identified a large number of sno-like RNAs originated from repetitive elements in humans and mice (**revised Result 2**). To demonstrate their presence *in vivo*, we firstly conducted a half-life analysis of napRNAs in HepG2 cells treated with Actinomycin D. Interestingly, the half-lives of CD-napRNA, ACA-napRNAs are similar to that of many mRNAs but longer than that of snoRNAs (**reviewer Figure 2; revised Supplementary Fig. 5b**). Secondly, we performed FBL/DKC1-RIP experiments to investigate their interactions with FBL/DKC1 proteins. As a result, we confirmed that certain novel C/D box napRNAs do indeed interact with the FBL protein (**Supplementary Fig. 9h**) and some novel H/ACA box napRNAs bind to the DKC1 protein (**Supplementary Fig. 10i**), which further demonstrated that these sno-like napRNAs are potential functional molecules, but not technical artefacts. Additionally, we provided evidence to support the functional significance of napRNA mmu-novel-CD-6, which contains repetitive sequences. We have demonstrated that mmu-novel-CD-6 is involved in regulating skeletal muscle differentiation (**revised Supplementary Fig. 11**).
- (2) These napRNAs feature motifs found consistently at specific locations, including the A/B box in Pol3 napRNAs, the C/D box in CD-napRNAs, and the H/ACA box in ACA-napRNAs (**revised Result 2**). These patterns strongly suggest that they are likely generated through specific transcription and processing patterns, indicating their potential as functional molecules.
- (3) We employed the starBase platform to analyze the interactions between repetitive napRNAs and RBPs. Intriguingly, a vast majority (94.8%) of repetitive napRNAs interact with at least one RBP. Nearly half of these repetitive napRNAs (45.8%) interact with more than 20 RBPs (**reviewer Figure 19A**). For example, hsa-napRNA-1 (**reviewer Figure 19B**) binds with multiple RBPs, the functions of which are primarily associated with RNA processing and localization (**reviewer Figure 19C**).

Reviewer Figure 19 napRNAs originated from repetitive sequences exhibit substantial overlap with RBP CLIP peaks. **A.** The number of RBPs binding to napRNAs originating from repetitive elements. **B.** The structure of hsa-napRNA-1 (chr8:123,434,451-123,434,610: +) which were derived from SINE element. **C.** Gene Ontology analysis of RBPs that interact with hsa-napRNA-1.

3. Most of the RNA species found seem to be some type of processing intermediate that is more stable due to its structure. It would be a significant finding if some of these RNAs are proved to have some important biological role. Changes in expression or association with histone modifications however are not functional evidence. The authors should avoid overstatements.

Response: We sincerely appreciate and acknowledge the valuable advice provided. We agree with the constructive comments and have made the following improvements:

To avoid overstatements, we have removed some sentences mentioning functionality, for instance, “suggesting that napRNAs may play crucial roles at the early stage of skeletal muscle differentiation.”, “We further found that a number of RBPs potentially interact with the 3’-ends of the snotrons (**Supplementary Fig. 11j**). Many RBPs showed different binding preferences between snotrons and other introns containing snoRNAs. Notably, AQR showed the greatest difference among all RBPs (**Supplementary Fig. 11j**). Given that AQR interacts with many splicing factors (**Supplementary Fig. 11k**) and may facilitate the folding of snoRNA sequences and snoRNP biogenesis²⁵, we speculated that the AQR protein may contribute to the biogenesis and stability of snotrons.” and “suggesting that napACA68 may also play a regulatory role in skeletal muscle differentiation”. The above-mentioned sentences have been removed from the revised manuscript.

Additionally, while we have identified two functional RNAs, DINAP (**revised Result 7**) and mmu-novel-CD-6 (**Supplementary Fig. 11**), the status of other napRNAs as degradation products or functional RNAs requires further experimental validation. These results have been added in the

discussion section.

4. Regarding the changes in expression, it should be shown whether they correlate with changes in expression of the host genes, and therefore could be explained by them.

Response: Thank you for the constructive suggestions. To assess the potential correlation between napRNA expression and their host genes, we quantified the expression levels of host genes using RNA-seq data. Our analysis revealed that the expression patterns of most napRNAs exhibit weak correlation with their respective host genes (**Reviewer Figure 20; revised Supplementary Fig. 6d, h-j**).

Reviewer Figure 20. Scatterplot show the correlation between reads (\log_2 (RPM), reads per million) obtained by the NAP-seq (napRNA regions) and RNA-seq (host gene).

5. The authors should analyze the half-life of the napRNAs.

Response: We appreciated this constructive advice. In the revised manuscript, we conducted a half-life analysis of napRNAs in HepG2 cells treated with Actinomycin D. Generally, we have assessed that half-lives of a total of five classes of napRNAs, including snoRNA-like RNAs, sliRNAs, snotrons and misRNAs (**Reviewer Figure 2A-F; revised Supplementary Fig. 5b-f**). We observed that the half-life of sno-like napRNAs is approximately 2-4 hours (**Reviewer Figure 2A**), which is consistent with that of mRNA and SPA lncRNA containing both a 5' snoRNA and a 3' polyA tail (**Reviewer Figure 2F**). Comparatively, the half-life of sliRNAs were around 20 minutes to 1 hour, which were longer than that of ACTB intron which is rapid degraded (**Reviewer Figure 2B and 2C**). Snotrons had a half-life of 20 minutes to 2 hours, which were similar to or longer than that of corresponding snoRNAs (**Reviewer Figure 2D**). The half-life of misRNAs is approximately half that of many miRNAs (**Reviewer Figure 2E**).

6. What are the steady-state levels of the napRNAs? How do they compare to other functional RNA species? (i.e. mRNAs and ncRNAs)

Response: We appreciate this constructive question regarding the expression levels of napRNAs compared to other functional RNA species. To address this, we calculated the steady-state expression levels of napRNAs and other RNA species using total RNA-seq data from the GSE88089 dataset and small RNA-seq data from the ENCSR000CRX dataset.

Our analysis revealed that the expression level of napRNAs is higher than that of long non-coding RNAs (lncRNAs) (**Reviewer Figure 21; revised Supplementary Fig. 5a**). However, the expression level of napRNAs is lower than that of mRNAs and snoRNAs (**Reviewer Figure 21; revised Supplementary Fig. 5a**). These comparisons provide an understanding of the relative expression levels between napRNAs and other functional RNA species.

Reviewer Figure 21. Box plots showing the expression level of mRNA, ncRNA and novel-napRNAs. Each box shows the first quartile, median, and third quartile.

7. The experimental data supporting the functionality of DINAP are not convincing. It is unclear what type of RNA is DINAP. Does it contain snoRNA sequences? DKC1 is known to interact with snoRNAs. How was the DINAP-DKC1 interaction identified in the first place? Just based on eCLIP data? Is it the only protein that interacts with it by eCLIP? The way the eCLIP data are presented should be improved, for instance showing a larger genomic window. Also, how many DKC1 peaks are in the genome? What is the intensity of DINAP peak compared to others?

Response: We would like to express our gratitude to the reviewer for their insightful comments, which have helped enhance the quality of our research. By carefully analyzing the sequence and structure of DINAP, we surprisingly have determined that DINAP is a chimeric ncRNA consists of a box C/D and a box H/ACA snoRNA-like domains. This finding suggests that DINAP is similar to other chimeric box C/D-H/ACA RNAs, such as U85 and U87⁴ (**Reviewer Figure 22A**).

DKC1 can interact with the H/ACA box of DINAP. Additionally, we have identified the interaction through eCLIP data and RIP-qPCR experiments (**Reviewer Figure 22B-C; revised Fig. 6e**). We also observed a relatively high intensity of the DINAP peak (ENCFF841ZPX, rank 8064, 13.4%) compared to other DKC1 peaks in the genome (**Reviewer Figure 22C**).

In response to the reviewer's suggestion, we have improved the presentation of the eCLIP

data by displaying a larger genomic window (Reviewer Figure 22C; revised Fig. 6b).

Reviewer Figure 22. DKC1 potentially interacted with DINAP. **A** The highly stable secondary structure of DINAP. **B** Genome Browser view showing the signal density of DKC1 binding in the region surrounding DINAP in DKC1 eCLIP data. **C** IGV plot showing the signal density of DKC1 binding in the region surrounding DINAP in HepG2 eCLIP-seq data (ENCFF252TQP). Yellow columns represent the coverage of eCLIP enriched read.

8. The FISH experiment in Fig 6c lacks controls (knockdown or knockout of DKC1 or DINA) to show the specificity of the signal.

Response: We sincerely appreciate the valuable advice provided. According to your constructive suggestions, in the revised manuscript, we have conducted an analysis of the subcellular localization of DINAP and DKC1 using the nucleolus marker FBL in both wild type and DKC1 knock-down HepG2 cells.

In the DKC1 knock-down HepG2 cells, we observed a decrease in the subcellular signal of DKC1. Furthermore, we found that DINAP co-localized with DKC1 in the cell nucleus (Reviewer Figure 23; revised Fig. 6d).

Reviewer Figure 23. DINAP interacted with DKC1 protein in HepG2 cells. **A** The expression level of DKC1 was examined by qPCR and western blotting in DKC1-knockdown HepG2 cells. **B** Subcellular localization of DINAP (yellow) as assessed by smiFiSH, and of DKC1 (red) as assessed by immunofluorescence staining. Nuclear DNA was stained with DAPI. FBL (green) was as the nucleolus marker.

9. How does DINAP overexpression levels compare to its endogenous levels?

Response: Thank you for the review's comments. Compared to the endogenous level, DINAP overexpression levels is $\sim 10^5$ fold. We conducted gain-of-function and loss-of-function experiments to confirm the functions of DINAP. We overexpressed DINAP in HepG2 cells and then examined the turnover of endogenous DKC1 protein using a cycloheximide (CHX) chase assay (**revised Fig. 6f; Supplementary Fig. 19g**). Ectopic expression of DINAP increased the protein level of DKC1 but did not affect its mRNA level (**revised Fig. 6f; Supplementary Fig. 19g**). Moreover, we generated DINAP-knockout (KO) cells by deleting the DINAP gene using CRISPR/Cas9 genome editing (**revised Supplementary Fig. 19h**). The protein level of DKC1 was decreased in DINAP-KO cells compared to WT cells (**revised Fig. 6f**). These gain-of-function and loss-of-function results revealed that DINAP regulates the stability of the DKC1 protein.

10. It should be shown in the same experiment in parallel the effect of DKC1 and DINAP knockdown and overexpression alone and in combination.

Response: Thank you for the reviewer's suggestions. We have corrected it in the revised manuscript (**Reviewer Figure 24; revised Fig. 6f**).

Reviewer Figure 24 WB analysis of DKC1 protein levels in DINAP-overexpressing cells, KO-DINAP cells and KO-DINAP cells with restoration of DINAP expression treated with 0.05 mg/ml CHX.

Reviewer #3 (Remarks to the Author):

In the manuscript titled “NAP-seq reveals novel classes of exceptionally structured noncoding RNAs with regulatory functions,” authors Liu et al, describe a novel method to capture long (>50) non-capped RNAs. Using a series of end digestion, RNA isolation, rRNA depletion and nested RT-PCR, the authors demonstrate they can indeed enrich for non-capped RNAs for which the majority arise from repetitive elements such as Alu SINE RNAs in humans and B2 SINE RNAs in mouse as well as LINE and MIR derived RNAs and snoRNAs. The authors also describe library representation of novel transcripts with “exceptional structure” and attempt to functionally validate some of the biological functions of the “novel” non-capped RNAs. Their method is certainly an improvement for the study of non-capped RNAs, however a number of validating experiments should be conducted and some points need to be addressed before the manuscript is published.

Response: We very much appreciate the Reviewer’s positive appraisal of our manuscript. We very much appreciate the Reviewer’s thoughtful and constructive suggestions and comments, which help us to revise and improve the manuscript. According to the Reviewer’s suggestions and comments, we have performed additional experiments and data analyses. With all those new results, we have carefully revised our manuscript and addressed all the concerns accordingly.

In detail:

1. The authors develop two library approaches following RNA preparation and reverse transcription (Nanopore and Illumina). To demonstrate they are indeed enriching non-capped RNAs, they should for both library methods, prepare a standard sequencing library for comparison to their developed method.

Response: We sincerely appreciate the constructive suggestions. In order to demonstrate that we are indeed enriching non-capped RNAs, we conducted additional analyses as follows:

To demonstrate that NAP-seq are indeed enriching non-capped RNAs, we conducted additional experiments and analyses as follows: Firstly, to confirm the removal of capped RNAs in NAP-seq, we performed CAP-seq experiments, identifying 66665 high-confidence 5’-cap sites within HepG2 cell lines. When comparing these 66665 5’-cap sites to the 5’-start sites of

napRNAs identified through NAP-seq, we found that only three sites (0.2% = 3/1220) exhibited overlap (**Reviewer Figure 25A; revised supplementary Fig. 2b**). Upon careful analysis of the terminal characteristics of these three identified RNAs in NAP-seq library, we found that their expression levels were extremely low and lacked significant 5' start site signals (**Reviewer Figure 25B; revised supplementary Fig. 2c**). These results indicate that the NAP-seq method markedly enriches non-capped RNAs. Furthermore, we carried out polyA-selected RNA-seq experiments to investigate the extent of non-capped RNA enrichment across various experimental methods. As a result, we found that NAP-seq exhibited a significant enrichment of known non-capped ncRNAs (such as snoRNAs) in comparison to the polyA-selected RNA-seq approach conducted on the same cell lines (**Reviewer Figure 25C; revised supplementary Fig. 2a**). These results provide additional substantiation for the capacity of NAP-seq to effectively enrich non-capped RNAs.

Reviewer Figure 25. NAP-seq enriched noncapped RNA. **A** Intersection of 5'-start-sites of napRNAs detected by NAP-seq and 5'-cap-sites detected by CAP-seq. **B** Genome Browser view of 5'-start signals from NAP-seq and CAP-seq (RPM, reads per million) in an extended region of three overlapped sites. **C** Heatmaps showing the expression of known snoRNAs from NAP-

seq and RNA-seq.

2. Furthermore, because their RNase H based digestion of rRNA and other high abundance ncRNAs used custom probes, they should at the very least quantify those targets before and after depletion to assess the efficiency of depletion.

Response: We greatly appreciate your valuable suggestion. In the revised manuscript, we have emphasized the remarkable efficiency of the RNase H method, which has been demonstrated to achieve an impressive removal rate of up to 99% (**Reviewer Figure 1A, revised supplementary Fig. 1b**).

To evaluate the efficiency of RNase H-based RNA depletion technique, we performed qRT-PCR assays comparing genes with and without RNase H treatment. As depicted in **Reviewer Figure 1A (revised Supplementary Fig. 1b)**, we observed that this technique achieved a remarkable depletion efficiency of up to 99% for the targeted RNAs. Furthermore, we performed an assessment of the RNA composition in the RNase H-treated HepG2 rep1 library by analyzing the fraction of reads aligning to genomic categories and specific RNA regions. As depicted in **Reviewer Figure 1B**, we found a substantial reduction to approximately 13.99% in the fraction of reads aligning to rRNA which were originally 80%-90% of the total RNA in eukaryotic cells¹, indicative of successful depletion. Moreover, the reads aligning to the selected RNAs (SNORA73A, SNORA73B, RMRP, and RPPH1) were effectively reduced to undetectable levels. These results provide compelling evidence that the RNase H-based RNA depletion technique is highly efficient in selectively depleting targeted RNAs, demonstrated by the remarkable reduction in rRNA aligning reads as well as the near complete elimination of the selected RNAs of interest.

3. Moreover, to validate the predicted structures as “exceptionally structured” the authors should conduct at least some in vitro validation such as SHAPE-seq or ssRNA digestion assays. Secondary structure prediction software is generally used to fit models rather than be used as a ground-truth structure. Lastly, the authors should consider the use of spiked in RNA such as a non-capped RNA and capped RNA to assess the efficiency of enrichment and representation in the library.

Response: We thank you very much for the professional comments and suggestions from the reviewer. Following the reviewer’s constructive advice, we developed NAP-SHAPE-MaP, an approach combining SHAPE-MaP⁵⁻⁹ and our NAP-seq experiments to probe intact RNA structures of napRNAs. NAP-SHAPE-MaP analysis showed that the increase in SHAPE reactivity was more significant in single sequence than paired regions (**Reviewer Figure 26A; revised Supplementary Fig. 3f**), consistent with previous observations^{6,10}. We also obtained accurate structure reactivity for snoRNAs with known secondary structure (**Reviewer Figure 26B; revised Supplementary Fig. 3g**). These results indicated the accuracy of NAP-SHAPE-Map to examine the RNA structure. Novel napRNAs with a significantly conserved RNA secondary structure identified by RNAalifold software¹¹ were also verified by NAP-SHAPE-MaP (**Reviewer Figure 26C, D; revised Supplementary Fig. 3h, i**). Moreover, to enhance the accessibility of structural insights for individual napRNAs, we have established a dedicated web page known as the “napRNA structure” (accessible at <https://rnasyu.com/napSeq/structure.php>). This webpage offers a comprehensive array of

details, encompassing sequence alignments, structural representations, conservation scores, and species-specific co-variation identifications. This resource is designed to facilitate effortless access for readers seeking a profound understanding of the intricate structural attributes of napRNAs.

To demonstrate that NAP-seq are indeed enriching non-capped RNAs, we conducted additional experiments and analyses as follows: Firstly, to confirm the removal of capped RNAs in NAP-seq, we performed CAP-seq experiments, identifying 66665 high-confidence 5'-cap sites within HepG2 cell lines. When comparing these 66665 5'-cap sites to the 5'-start sites of napRNAs identified through NAP-seq, we found that only three sites (0.2% = 3/1220) exhibited overlap (**Reviewer Figure 25A; revised supplementary Fig. 2b**). Upon careful analysis of the terminal characteristics of these three identified RNAs in NAP-seq library, we found that their expression levels were extremely low and lacked significant 5' start site signals (**Reviewer Figure 25B; revised supplementary Fig. 2c**). This result implies that the NAP-seq method markedly enriches non-capped RNAs. Furthermore, we carried out polyA-selected RNA-seq experiments to investigate the extent of non-capped RNA enrichment across various experimental methods. As a result, we found that NAP-seq exhibited a significant enrichment of known non-capped ncRNAs (such as snoRNAs) in comparison to the polyA-selected RNA-seq approach conducted on the same cell lines (**Reviewer Figure 25C; revised supplementary Fig. 2a**). These results provide additional substantiation for the capacity of NAP-seq to effectively enrich non-capped RNAs.

Reviewer Figure 26 NapRNAs structure analysis. **A** NAP-SHAPE-MaP reactivity of the napRNA. X axis represent various secondary structure region of napRNA. Y axis represent the mean reactivity of SHAPE-MaP. **B** The secondary structure model and SHAPE-MaP reactivity score for each base of SCARNA3 (left) and SNORA14B (right), with different colors representing different range of reactivity scores. **C** Covariation structure of a specific napRNA, hsa-napRNA-1852 (chr3:193,155,217-193,155,403: -) and hsa-napRNA-2666 (chrX:20194567-20194942: -). **D** The secondary structure model and SHAPE-MaP reactivity score for each base of hsa-napRNA-1852 and hsa-napRNA-2666, with different colors representing different range of reactivity scores.

4. Beyond experimental validation, the methodology section requires soem more clarification for the protocol to be replicable. The authors should at least include: the amount of cDNA used for both libraries, “1/2” is not very quantitative. What number of PCR cycles are used? What Nanopore sequencing kit did they use? Was library barcoding use? Also, while I understand why cDNA fragmentation was used since RNA fragmentation would introduce erroneous non-capped RNA fragments, a descriptive methodology of fragmentation must be included. Additionally, detailed information of the software used is missing in many cases (for example version number), especially in the case of the supplementary schematic for “Napseeker.”

Response: Thank you for your comments, and we apologize for the missing information in the original manuscript. In the revised version, we have included additional details in the method section. The cDNA was obtained through reverse transcription and column purification, using 20 μ L of nuclease-free water. For the construction of the NGS library, 10 μ L of the cDNA was utilized. The preliminary PCR consisted of 18-21 cycles to generate amplicons. Another 10 μ L of the cDNA was used for the preparation of the TGS library. The TGS library was constructed for Nanopore sequencing using SQK-LSKSP9. In NAP-seq, adapters with 6N barcoding were utilized, where N represents A, T/U, C, or G. To prevent mis-priming or introduction of artifacts, we directly ligated adapters with barcoding to the ends of napRNAs and performed fragmentation using the cDNA product.

We have made every effort to provide comprehensive details by sharing the source code and tutorial of the napSeeker on GitHub. Additionally, in the revised manuscript (**Method section**), we have included information regarding the software versions used, which can be found in **Reviewer Table 1**.

The following section provides a comprehensive description of the software tools employed in this study:

1. Cutadapt (v2.8): Used to remove sequencing adapters from paired-end reads in Illumina sequencing data.
2. CutNapAdapter (self-developed): Utilized for the removal of NAP-seq specific adapters from sequencing reads.
3. STAR (v2.7.0): Employed for mapping clean reads to the reference genome in Illumina sequencing data.
4. minimap2 (v2.23): Used to map clean reads to the reference genome in nanopore sequencing data.
5. NapSeeker (self-developed): Applied to identify napRNAs from NAP-seq data.
6. Bedtools (v2.17.0): Utilized for intersecting candidate napRNAs with known annotations

from GENCODE (human release 30 and mouse release 23), snoRNABase (v3), and snoRNABase (v3).

7. Limma (v3.5.0): Employed for identifying differentially expressed napRNAs.
8. Pheatmap (v1.0.10): Utilized to generate expression heatmaps for differentially expressed napRNAs.
9. DeepTools (v3.5.1): Used for generating distribution profiles of various histone modifications around candidate napRNAs.
10. R2R (v1.0.6): Applied for visualizing napRNA structures.
11. RNAz (v1.0.6): Used to calculate z-scores for the napRNAs.
12. Clustal W (v2.1): Employed for aligning sequences across multiple species.
13. Mfuzz (v2.54.0): Utilized to cluster napRNAs based on their expression values.
14. Ggseqlogo (v1): Employed to create sequence motifs plots for napRNAs.
15. ClusterProfiler (v1): Utilized for performing Gene Ontology (GO) analysis.
16. IGV (v2.8.2): Used to visualize napRNA expression values within specific regions.
17. RNAalifold (2.4.14): Applied for predicting the secondary structures of RNAs.

Reviewer Table 1. Software version information.

Software and algorithms	source	links
Cutadapt v2.8	Cutadapt Software	https://cutadapt.readthedocs.io/en/stable/
cutNapAdapter v1	This paper	https://github.com/junhong-huang/cutNapAdapter
STAR 2.7.0	STAR Software	https://github.com/alexdobin/STAR
minimap2 v2.23	minimap2 Software	https://github.com/lh3/minimap2
napSeeker v1	This paper	https://github.com/junhong-huang/napSeeker
bedtools v2.17.0	bedtools Software	https://bedtools.readthedocs.io/en/latest/
limma v3.5.0	R package	https://bioconductor.org/packages/release/bioc/html/limma.html
pheatmap v1.0.10	R package	https://CRAN.R-project.org/package=pheatmap
deepTools v3.5.1	deepTools Software	https://github.com/deeptools/deepTools
R2R v1.0.6	R2R Software	https://sourceforge.net/projects/weinberg-r2r/
RNAz v2.1	RNAz Software	https://www.tbi.univie.ac.at/software/RNAz/
Clustal W v2.1	Clustal W Software	http://www.clustal.org/clustal2/
Mfuzz v2.54.0	R package	https://www.bioconductor.org/packages/release/bioc/html/Mfuzz.html
ggseqlogo v1	R package	https://github.com/omarwagih/ggseqlogo

clusterProfiler v1	R package	https://github.com/YuLab-SMU/clusterProfiler
IGV v2.8.2	IGV Software	http://software.broadinstitute.org/software/igv/
RNAalifold 2.4.14	RNAalifold Software	http://rna.tbi.univie.ac.at/cgi-bin/RNAWebSuite/RNAalifold.cgi

5. Inclusion of the following would also improve the manuscript:

RIN values for each sample. Low RIN could erroneously increase the number of Non-capped RNAs.

Response: Thank you for your valuable suggestion. Prior to library construction, we evaluated the integrity of the RNA samples using formaldehyde denaturing gel and Qsep1 to determine the RNA quality number (RQN) values. RQN and RIN are both important metrics for assessing RNA integrity in research studies. RQN is assessed on a scale ranging from 1 to 10, where a score of 1 indicates total RNA that is completely degraded, and 10 represents intact total RNA. A higher RQN value signifies higher-quality RNA with minimal degradation in the sample. In the revised manuscript, we have included the RQN values for each sample used in library preparation. You can find these values in **Reviewer Table 2 (Methods and revised Supplementary Table 1)**.

Typically, researchers use a threshold of 7 or higher to classify RNA as “high quality”, making it suitable for library preparation methods. In our research, the RQN values of the samples for library preparation range from 8.69 to 9.74, indicating that these RNAs are of exceptionally high quality.

Reviewer Table 2. RQN of RNA used to construct NAP-seq library

RNA samples	RQN
HEK293T_Replicate_1	8.75
HEK293T_Replicate_2	8.74
HEK293T_Replicate_3	9.73
U87_Replicate_1	9.75
U87_Replicate_2	8.74
U87_Replicate_3	8.72
HepG2_Replicate_1	9.73
HepG2_Replicate_2	9.71
HepG2_Replicate_3	9.71
HepG2_ADR_Replicate_1	9.71
HepG2_ADR_Replicate_2	9.70
HepG2_ADR_Replicate_3	9.73
HepG2_CoCl2_Replicate_1	8.75
HepG2_CoCl2_Replicate_2	8.77
HepG2_CoCl2_Replicate_3	8.73
HepG2_polyIC_Replicate_1	9.74
HepG2_polyIC_Replicate_2	9.72
HepG2_polyIC_Replicate_3	9.72

C2C12_D0_Replicate_1	8.84
C2C12_D0_Replicate_2	8.84
C2C12_D0_Replicate_3	8.81
C2C12_D72_Replicate_1	8.79
C2C12_D72_Replicate_2	8.77
C2C12_D72_Replicate_3	8.76
C2C12_D96_Replicate_1	8.79
C2C12_D96_Replicate_2	8.73
C2C12_D96_Replicate_3	8.73
C2C12_D144_Replicate_1	8.72
C2C12_D144_Replicate_2	8.70
C2C12_D144_Replicate_3	8.69

6. Discussion on the possibility of RNA processing to increase novel non-capped RNAs such as the processing of B2 and Alu RNAs described in Zovoilis et al, 2016 and Cheng et al, 2020-2021.

Response: We do appreciate this great suggestion raised by the reviewer, and we have quoted and added the discussion about the function of RNA derived from various repetitive elements, including the B2 RNAs and napRNAs which were identified by NAP-seq in the revised manuscript (**Discussion, Paragraph 4**).

7. Also, a discussion on where Nanopore vs Illumina is more useful. Could there be extra long Non-capped RNAs that Nanopore would be more suitable to resolve. This may require modification of the amplification protocol to include a long-amplification step instead. In supple 3a, what distribution of differentially expressed napRNAs change in each condition?

Response: We sincerely appreciate your valuable input, as it has greatly contributed to the clarity and completeness of our manuscript. Nanopore sequencing is a unique and scalable technology that enables direct, real-time analysis of long DNA or RNA molecules^{26, 27}. It provides advantages for studying various aspects of nucleic acids, including DNA/RNA modifications^{28, 29} and RNA structures³⁰. Due to the high error rate of Nanopore sequencing, especially at the ends, we add sequencing adapters to RNA molecules for more accurate identification of the 5' and 3' ends of RNA during the Nanopore sequencing. On the other hand, next-generation sequencing (NGS), such as Illumina sequencing, offers extremely high sequencing accuracy. Therefore, combining both methods in the NAP-seq would result in greater accuracy of sequencing. Both NAP-seq-TGS and NAP-seq-NGS can be employed independently for investigating napRNAs, except when studying modifications on these RNAs (**revised Discussion, Paragraph 2**).

In NGS, RNA molecules are typically fragmented during library preparation. If there are multiple 5'-start and 3'-end signals within a single long RNA molecule, it can be challenging to determine which signals correspond to the same RNA molecule. In such cases, we need to refer to TGS data to gain more complete information about the intact full-length RNA. For example, in the miRNA processing diagram (**revised Fig. 5g**), we incorporated TGS reads to provide a more intuitive representation.

To obtain the full-length sequences of napRNAs, we employed KAPA HiFi HotStart

ReadyMix during napRNA amplification in the construction of NAP-seq-TGS or NAP-seq-NGS libraries (see "Methods" section). KAPA HiFi HotStart polymerase demonstrates industry-leading fidelity and performance in amplifying long targets, as observed in smart-seq experiments³¹, surpassing other commonly used polymerases. Additionally, in NAP-seq method, the use of SuperScript IV RT enzymes enabled the identification of previously undetected napRNAs with highly stable structures (revised Fig.1). By using a nested RT primer, NAP-seq method can amplify the full-length cDNA sequence of long napRNAs while avoiding widespread mispriming artefacts.

Regarding the distribution of differentially expressed napRNAs across each condition, we have visualized these changes in Reviewer Figure 27. In the humans, differentially expressed napRNAs are primarily located in intronic regions and repetitive elements, such as Alu and MIR, as well as intergenic regions. Notably, napRNAs distributed in Alu elements are significantly increased during the immune response (poly(I:C) treatment) in HepG2 cells. However, when cells are under DNA repair (ADR treatment) or hypoxia stress (CoCl₂ treatment), the expression distribution of napRNA remains relatively unchanged (Reviewer Figure 27A). These results indicate that napRNA originated from Alu elements may participate in the immune response. In mice, differentially expressed napRNAs are found in repetitive elements (B2) and intronic regions. Throughout myoblast differentiation, the expression of napRNAs in B2 elements decreases, while napRNAs in intronic regions exhibit a notable increase (Reviewer Figure 27B).

Reviewer Figure 27 Genomic distribution of differentially expressed napRNAs in humans (A)

and mice (**B**).

8. The introduction should discuss further why up to this point, sequencing methods have not captured these RNAs. What about structure prevents their library preparation? Other techniques such as TGIRT-seq can reverse-transcribe complicated structures such as snoRNAs.

Response: We sincerely appreciate your excellent suggestion, and we have incorporated the relevant description in the revised manuscript (**Introduction, Paragraph 2 and Discussion, Paragraph 2**). Current sequencing methods are incapable of capturing these RNAs primarily due to the limitations in next-generation sequencing (NGS) read lengths (e.g., paired-end 150 nucleotides) and the complex higher-order structures of RNA. Approaches like TGIRT-seq utilized TGIRT enzyme with higher processivity and fidelity than conventional retroviral reverse transcriptase to reverse-transcribe complicated structures of RNAs³²⁻³⁴. However, due to the limitations of short NGS read lengths, TGIRT-seq cannot detect complete sequences of long and structurally complex RNA molecules.

Some additional minor points that in my opinion should be addressed are:

1. Line 101

- Could mention the cell line names.

Response: Thank you for your suggestion and we apologize for the lack of detailed information. We have thoroughly reviewed the manuscript and addressed this issue by providing comprehensive details about the cell lines used in our study in the revised version. We have revised the sentence in Line 101 as follows: We next performed NAP-seq on samples from human cell lines, including HepG2, HEK293T and U87, and a mouse cell line-C2C12 and obtained approximately 46 million reads and 10 million reads per library, on average, through deep sequencing (**revised Supplementary Table 1**) by NAP-seq-NGS and NAP-seq-TGS, respectively (**revised Line 105-107**).

2. Sentences from line 111-116

- At this point, have they checked if their analysis missed any known napRNA (false negative)?

Response: Thank you for the reviewer's comments. We have examined the expression levels of the known napRNAs and observed that snoRNA HBI-36 and U70C were not detected (**Reviewer Figure 28**). Considering the absence of clear signals for these two snoRNAs in the RNA-seq data, we postulate that they may be expressed exclusively in specific cell types, for instance, HBI-36 was found to be specifically expressed in the choroid plexus³⁵.

Reviewer Figure 28. The heatmap showing the expression level of known snoRNAs.

3. - Are there any unexpected peaks that appear in non-terminal region? If so, how would that be explained?

Response: Thanks for the reviewer's comments. We have observed peaks in the non-terminal regions of certain genes. These non-terminal peaks are often lower or significantly lower than the main terminal peak, indicating that RNA molecules are undergoing degradation processes.

4. Line 130

- Better to briefly describe what each condition is.

Response: We sincerely appreciate the valuable suggestions provided by the reviewer, as they have greatly contributed to the improvement of our paper. In response to these suggestions, we have incorporated the following additional information into the revised manuscript (**Result 2, Paragraph 1**).

We have added the followings in the revised result 2, Paragraph 1:

In these stress responses, poly (I:C) mimics double-stranded RNA to induce the activation of immune response ¹²; ADR causes DNA damage, triggering cells to activate their DNA damage repair machinery ^{13, 14}; CoCl₂ artificially induces hypoxia ¹⁵, which has allowed the characterization of the hypoxia response at the cellular, biochemical and molecular levels ¹⁶.

5. Line 135 / line 822-824

- How values like "20" were decided? Based on preliminary tests, experience, etc.?

Response: Thank you for the reviewer's comments. Our decision was made based on our experience and a preliminary test. In this study, we choose to retain napRNAs if they are detected with sequencing reads above 20 and expressed in at least two samples. Our decision was made in accordance with the general rule, given the absence of a gold standard. In the case of detections in more than two libraries, using 20 as the filter threshold is enough to balance sensitivity and specificity for identification of putative napRNAs. Subsequently, we examined the quantity of known napRNAs identified by NAP-seq, such as snoRNA. The results revealed the identification of all previously annotated snoRNAs, except for HBI-36 and U70C (**Reviewer Figure 29**). Considering the absence of clear signals for these two snoRNAs in the RNA-seq data, we postulate that they may be expressed exclusively in specific cell types, for instance, HBI-36 was found to be specifically expressed in the choroid plexus³⁵. These results indicated the appropriateness of our screening criteria.

6. Line 145

- Briefly explain what lower MFE indicates to those who're not familiar with this?

Response: Thank you for the reviewer's comments, and we apologize for the lack of detailed description. We have included an explanation of the minimum free energy (MFE) value in the revised manuscript (**detail see Methods**). It is important to note that a lower MFE value indicates a more stable RNA structure.

MFE, based on the principles of thermodynamics, is utilized in RNA structure prediction by using a loop-based energy model and the dynamic programming algorithm introduced by Zuker et al³⁶. to predict the most stable secondary structure of an RNA molecule. This lowest free energy structure is typically considered the most likely structure to exist within a biological system.

7. Line 148 / line 870

General question about using Rscope and the secondary structure predictions

- how confident they are about predicted pairings, how conserved their input sequences are and how many sequences used for each prediction?

Response: Thank you for the reviewer's comments. Given that secondary structure prediction software is typically used to generate models rather than provide ground-truth structures, we combined experimental methods (SHAPE-Map) and computational methods (RNAalifold) instead of using the Rscope software. This approach was chosen to bolster the confidence in our predicted structures.

(1) Confidence in Predictions: To enhance our confidence in the predictions, we leveraged SHAPE-Map scores to guide RNA secondary structure prediction using RNAalifold with the "--shape" parameter. Higher SHAPE-Map scores indicate a greater likelihood of unpairing in specific regions.

(2) Sequence Conservation: We ensured that the input sequences we used had a sequence identity of at least 0.8 across different species. This stringent criterion helped us focus on conserved sequences.

(3) Diverse Sequence Set: It's important to note that different sequences were derived from various numbers of species. You can explore the specific sequences in our "napRNA structure" webpage, accessible at <https://rnasysu.com/napSeq/structure.php> (**Reviewer Figures 6-7**). The

diversity across species contributes richness and robustness to our analysis.

8. - Use of Rscope is okay when input sequence are mostly conserved. But with a less conserved input alignment, predicted pairings (even with a high score) can be more suspicious, and many mismatches, bulges, larger loops would be observed in most individual sequences.

Response: We appreciate the reviewer for raising this concern. In order to address this issue, we implemented stringent criteria for sequence retention. Specifically, we only kept sequences with a minimum of 80% similarity and 30% identity, while ensuring that the percentage of sequence gaps was kept below 10%. These criteria were applied to ensure the overall quality and reliability of the retained sequences.

9. - Did they do more analysis for the secondary structures? E.g. whether they share common flanking area or are from similar elements, are there other elements that may potentially interact with them?

Response: We sincerely appreciate the reviewer for providing us with constructive comments. We conducted further analysis and identified additional positions of the Alu element within the secondary structure. One particular position of interest is its involvement in forming the poly-A pocket in ACA RNAs (**revised Fig. 2h**). However, we did not observe any other notable findings beyond this observation.

We sincerely appreciate the reviewer for providing us with constructive comments. To analysis the napRNA secondary structures, we analyzed the minimum free energies (MFEs) of the napRNAs and found that they exhibited lower MFEs compared to random sequences, indicating their thermodynamic stability. This evaluation was performed using RNAfold. Furthermore, we conducted NAP-SHAPE-MaP, a combined approach of SHAPE-MaP and our NAP-seq, to explore the intact RNA structures of napRNAs. Our results revealed that these RNAs could fold into diverse stem-loop structures and possessed relatively low z-scores, suggesting the presence of stable secondary structures (**General point 4**).

We examined whether these napRNAs contained known sequence motifs and discovered that some napRNAs harbored specific motifs, including UUUU motifs located at the RNA 3' end, H/ACA boxes, and C/D boxes (**revised Result 3**). Additionally, we identified some napRNAs with GU-AG motifs at their respective beginnings and ends, which we named sliRNAs (**revised Result 4**). Furthermore, we attempted to de novo identify motifs but did not uncover additional patterns.

1. We attempted to identify the three-dimensional structures of these napRNAs, such as k-turn and backward k-turn. Besides the C/D-box napRNAs mentioned earlier, which could potentially form k-turn structures, we also identified DINAP, a napRNA that simultaneously exhibited H/ACA hairpin-hinge-hairpin structures and k-turn structures (**General point 3**).
2. We conducted a genomic distribution analysis of napRNAs with distinctive secondary structures. We observed that C/D-box snoRNAs primarily originated from intronic sequences, whereas snoRNAs with H/ACA boxes mainly originated from Alu repetitive sequences. This finding differs from previous reports, as these Alu-ACA elements accounted for over 50% of Alu sequences. Importantly, we noted that a substantial portion of Alu-ACA elements featured an open poly-A pocket (**revised Result 3**).

10. - Are they planning to provide sample alignments that correspond to structures they show in main and suppl figures?

Response: Thank you for the reviewer's constructive comments. To provide information of sample alignments that correspond to structures, we have constructed the webpage "napRNA structure" available at (<https://rnasysu.com/napSeq/structure.php>).

11. Line 203

- Suppl. Fig 5c,d aren't showing a secondary structure at all.

Response: Thanks for the reviewer's comments. We have provided the secondary structures. The structure mentioned in Supplementary Fig. 5c can be found in **revised Supplementary Fig. 8e**, while the structure mentioned in Supplementary Fig. 5d is now represented in **revised Fig. 2b**.

12. Line 292

- A bit confusing. Text says "no difference" but I read the bars are different as in suppl fig 10j.

Response: We sincerely thank the reviewer for bringing this to our attention, and we apologize for the ambiguous description in the original manuscript. In our computational analysis, our aim was to identify RNA binding proteins (RBPs) that exhibit specific binding to the 5'-SS and 3'-SS of sliRNAs compared to other introns. We have rectified the statement to "no difference of binding specificity" in the revised manuscript (**Result 4, Paragraph 3**) to provide clear and accurate information.

13. Line 319

- "possibly because of the complex" needs more explanation.

Response: We are sorry for the ambiguous description. We have corrected it in the revised manuscript (**result 5, Paragraph 1**) as follows: traditional methods, such as poly(A)+ RNA-seq, total RNA-seq and poly(A)- RNA-seq, could not identify these snotrons (**Fig. 4c, d; Supplementary Fig. 14a**), possibly because of the complex structure of snotrons which could be not read-through in reverse transcription.

14. Line 355/fig 4i

- how did they come to those within a short/long distance? -- make them clearly shown in fig 4i?

- The last step is confusing and hard to understand solely based on what's drawn. Consider better annotation text/colour-coding.

Response: We sincerely appreciate the reviewer for bringing this to our attention. In the revised manuscript, we have made the necessary correction to the model picture, which can now be found in **revised Fig. 4i**.

15. Line 802

- is the perl script included in one of their github repos or available upon request?

Response: Thanks for the reviewer's comments. The Perl scripts are available in our github <https://github.com/junhong-huang/NAP-seq-Perl-scripts> . We have added the description in the

revised manuscript (**Code availability section**).

16. Line 831

- Need to note the date of access, also applies to other text where "downloaded from database XX" is mentioned.

Response: Thank you for the reviewer's comments. We apologize for any oversight on our part. In response to this, we have taken the necessary steps to address the issue by adding a detailed description in the revised manuscript.

17. Fig1e

- The y-axis better to use $1e+05$ to be consistent with others.

Response: Thanks for the reviewer's comments and we sorry for our carelessness. We have corrected the y-axis to use " $0e+00, 5e+05$ ".

18. Line 1126

- What defines "same terminal sites"? Exact sequence match or other criteria?

Response: We would like to express our gratitude for the reviewer's comments and apologize for the ambiguous description in our manuscript. In the context of "same terminal sites," we specifically refer to an exact sequence match. To clarify this, we have included a detailed statement in the revised manuscript to provide a clear explanation in Fig.1 legend, as follows: The bottom panel shows a heatmap, in which each row represents a gene that matches the terminal sites with the NAP-seq-NGS read exactly, and each column represents the expression values of genes at a specific distance. expr, expression value.

19. Fig 4a

- the top structures are a bit hard to understand immediately. Could colour-code and highlight the region that's considered to be the "distance".

- Also for panel b: the last x-axis label should be ">50" since the one before it includes 50?

Response: We sincerely appreciate the constructive suggestions provided by the reviewer and apologize for any ambiguity in our description. Taking the reviewer's advice into consideration, we have made the necessary changes in the revised manuscript. Specifically, we have highlighted the term "distance" in the red line and corrected the last x-axis label to ">50" in both Figure 4a and 4b.

20. Fig 4f

- As a main figure, this structure is visually too messy.

Response: We appreciate the reviewer pointing this out. We have adjusted this figure in the revised manuscript (**Reviewer Figure 29**).

Reviewer Figure 29. The secondary structures of snotron mmu-snotron-5.

21. Suppl fig 1c/d

- Would be easier to compare if they're in one bar plot since they have the same categories.

Response: We appreciate the reviewer pointing this out. We have adjusted the figure in the revised manuscript (**Reviewer Figure 30; revised Supplementary Fig. 1e**).

Reviewer Figure 30 The distribution and the percentage of NAP-seq-NGS and NAP-seq-TGS reads in annotated gene types.

22. Suppl fig 3b

- Y-axis labels: What cell line is each treated sample?

Response: We apologize for any confusion caused by the initial omission and appreciate your attention to detail. The treated cell line is HepG2 which was included in the revised manuscript (**Reviewer Figure 31, revised Supplementary Fig. 6b and the method section**).

Reviewer Figure 31 Box plot showing the differential expression (3 biological replicates) of hsa-napRNA-48 (chr8:11,853,940-11,854,141: -), which was highly expressed in U87 cells compared to the other cell lines. RPM, reads per million.

References:

1. Derrien, T. et al. The GENCODE v7 catalog of human long noncoding RNAs: analysis of their gene structure, evolution, and expression. *Genome Res* **22**, 1775-1789 (2012).
2. Schaukowitch, K. et al. Enhancer RNA facilitates NELF release from immediate early genes. *Mol Cell* **56**, 29-42 (2014).
3. Clark, M.B. et al. Genome-wide analysis of long noncoding RNA stability. *Genome Res* **22**, 885-898 (2012).
4. Darzacq, X. et al. Cajal body-specific small nuclear RNAs: a novel class of 2'-O-methylation and pseudouridylation guide RNAs. *EMBO J* **21**, 2746-2756 (2002).
5. Siegfried, N.A., Busan, S., Rice, G.M., Nelson, J.A.E. & Weeks, K.M. RNA motif discovery by SHAPE and mutational profiling (SHAPE-MaP). *Nature Methods* **11**, 959-965 (2014).
6. Spitale, R.C. et al. Structural imprints in vivo decode RNA regulatory mechanisms. *Nature* **519**, 486-490 (2015).
7. Luo, Q.-J. et al. RNA structure probing reveals the structural basis of Dicer binding and cleavage. *Nature Communications* **12**, 3397 (2021).
8. Lucks, J.B. et al. Multiplexed RNA structure characterization with selective 2'-hydroxyl acylation analyzed by primer extension sequencing (SHAPE-Seq). *Proc Natl Acad Sci U S A* **108**, 11063-11068 (2011).
9. Smola, M.J., Rice, G.M., Busan, S., Siegfried, N.A. & Weeks, K.M. Selective 2'-hydroxyl acylation analyzed by primer extension and mutational profiling (SHAPE-MaP) for direct, versatile and accurate RNA structure analysis. *Nature Protocols* **10**, 1643-1669 (2015).
10. Luo, Q.J. et al. RNA structure probing reveals the structural basis of Dicer binding and cleavage. *Nat Commun* **12**, 3397 (2021).
11. Hofacker, I.L. Vienna RNA secondary structure server. *Nucleic Acids Res* **31**, 3429-3431 (2003).
12. Schlee, M. & Hartmann, G. Discriminating self from non-self in nucleic acid sensing. *Nat Rev Immunol* **16**, 566-580 (2016).
13. Tewey, K.M., Rowe, T.C., Yang, L., Halligan, B.D. & Liu, L.F. Adriamycin-induced DNA damage mediated by mammalian DNA topoisomerase II. *Science* **226**, 466-468 (1984).
14. Xu, W. et al. TP53-inducible putative long noncoding RNAs encode functional polypeptides that suppress cell proliferation. *Genome Res* **32**, 1026-1041 (2022).
15. Majmundar, A.J., Wong, W.J. & Simon, M.C. Hypoxia-inducible factors and the response to hypoxic stress. *Mol Cell* **40**, 294-309 (2010).
16. Munoz-Sanchez, J. & Chanez-Cardenas, M.E. The use of cobalt chloride as a chemical hypoxia model. *J Appl Toxicol* **39**, 556-570 (2019).
17. Bannister, A.J. & Kouzarides, T. Regulation of chromatin by histone modifications. *Cell Res* **21**, 381-395 (2011).
18. Lawrence, M., Daujat, S. & Schneider, R. Lateral Thinking: How Histone Modifications Regulate Gene Expression. *Trends Genet* **32**, 42-56 (2016).
19. Huang, J. et al. ChIPBase v3.0: the encyclopedia of transcriptional regulations of non-coding RNAs and protein-coding genes. *Nucleic Acids Res.* **51**, D46-D56 (2022).
20. Calo, E. & Wysocka, J. Modification of enhancer chromatin: what, how, and why? *Mol Cell* **49**, 825-837 (2013).
21. Jady, B.E., Ketele, A. & Kiss, T. Human intron-encoded Alu RNAs are processed and

- packaged into Wdr79-associated nucleoplasmic box H/ACA RNPs. *Genes Dev* **26**, 1897-1910 (2012).
22. Li, B. et al. RIP-PEN-seq identifies a class of kink-turn RNAs as splicing regulators. *Nat Biotechnol* (2023).
 23. Talross, G.J.S., Deryusheva, S. & Gall, J.G. Stable lariats bearing a snoRNA (slb-snoRNA) in eukaryotic cells: A level of regulation for guide RNAs. *Proc Natl Acad Sci U S A* **118** (2021).
 24. Liu, Y. et al. Splicing inactivation generates hybrid mRNA-snoRNA transcripts targeted by cytoplasmic RNA decay. *Proc Natl Acad Sci U S A* **119**, e2202473119 (2022).
 25. Hirose, T. et al. A spliceosomal intron binding protein, IBP160, links position-dependent assembly of intron-encoded box C/D snoRNP to pre-mRNA splicing. *Mol Cell* **23**, 673-684 (2006).
 26. Jain, M. et al. Nanopore sequencing and assembly of a human genome with ultra-long reads. *Nat Biotechnol* **36**, 338-345 (2018).
 27. Garalde, D.R. et al. Highly parallel direct RNA sequencing on an array of nanopores. *Nat Methods* **15**, 201-206 (2018).
 28. Simpson, J.T. et al. Detecting DNA cytosine methylation using nanopore sequencing. *Nat Methods* **14**, 407-410 (2017).
 29. Workman, R.E. et al. Nanopore native RNA sequencing of a human poly(A) transcriptome. *Nat Methods* **16**, 1297-1305 (2019).
 30. Aw, J.G.A. et al. Determination of isoform-specific RNA structure with nanopore long reads. *Nat Biotechnol* (2020).
 31. Picelli, S. et al. Smart-seq2 for sensitive full-length transcriptome profiling in single cells. *Nat Methods* **10**, 1096-1098 (2013).
 32. Wang, Z., Gerstein, M. & Snyder, M. RNA-Seq: a revolutionary tool for transcriptomics. *Nat Rev Genet* **10**, 57-63 (2009).
 33. Nottingham, R.M. et al. RNA-seq of human reference RNA samples using a thermostable group II intron reverse transcriptase. *RNA* **22**, 597-613 (2016).
 34. Qin, Y. et al. High-throughput sequencing of human plasma RNA by using thermostable group II intron reverse transcriptases. *RNA* **22**, 111-128 (2016).
 35. Cavaille, J. et al. Identification of brain-specific and imprinted small nucleolar RNA genes exhibiting an unusual genomic organization. *Proc Natl Acad Sci U S A* **97**, 14311-14316 (2000).
 36. Zuker, M. & Stiegler, P. Optimal computer folding of large RNA sequences using thermodynamics and auxiliary information. *Nucleic Acids Res* **9**, 133-148 (1981).

REVIEWERS' COMMENTS

Reviewer #1 (Remarks to the Author):

I would like to sincerely thank the authors for their work during the review period. I am pleased to see that some of my criticisms and comments were also shared by the other 2 reviewers. I feel that the authors have done their utmost to improve the readability and quality of their work, both by modifying the text and by providing new experimental data. I feel that the revised version is of better quality. If the editors decide to publish this work, I would like the authors to make the following minor changes in the final version of the manuscript:

#1

JC comment (1st reading) - The authors should discuss the trivial possibility that some (most) of the newly identified RNA species (low abundance, poorly conserved and overlapping interspersed repeated elements) could simply represent degradation products and/or RNA intermediates during pre-mRNA splicing or pri-miRNA processing.

Response to Authors - According to reviewer's suggestion, we have discussed the trivial possibility of the newly identified RNA species in the revised manuscript (Discussion section).

Unless I am mistaken, my comment was not highlighted in the discussion. I think it is important that the authors explicitly mention the possibility that many novel RNA species they described may not have regulatory functions but simply represent non-functional products of degradation which have been captured by increasingly sensible methods of deletion. From my point of view, it is important not to over-interpret the observations, and only more detailed future analyses will be able to demonstrate the evolutionary or physiological significance of these new RNA species.

#2

Introduction section – “Furthermore, we demonstrated that a novel structured napRNA regulates myoblast differentiation in mice and a new napRNA, DINAP, can interact with DKC1 to maintain its protein stability, which further promotes the proliferation of HepG2 cells”

I suggest that the authors mention the fact that DINAP is a C/D and H/ACA hybrid .

#3

Page 7 – “For instance, the evolutionary conservation of hsa-napRNA-188 2120 is very highly conserved from human to zebrafish (Supplementary Fig. 4b)”.

Replace by “For instance, the hsa-napRNA-188 2120 is highly conserved from human to zebrafish (Supplementary Fig. 4b)”.

#4

Page 10 – “We next developed new bioinformatic pipelines to identify novel small nucleolar RNAs 268 (snoRNAs), including C/D box and H/ACA box RNAs, from NAP-seq profiles”.

Computer pipelines do not allow direct access to snoRNAs, ie RNAs that accumulate in the nucleoli. This is all the more true as the sub-nucleolar compartmentalisation of some of the candidates tested does not reveal the strong accumulation in nucleoli that is commonly observed with true snoRNAs.

#5

Response to Authors – “In comparison to the endogenous expression, the ectopic expression of mmu-novel-CD-6 was found to be approximately 300-fold”

In order for the reader to be able to judge the artificial and non-physiological nature of the over-expression experiments, the authors should indicate the level of over-expression (300-fold) in the text.

Reviewer #2 (Remarks to the Author):

The authors have done a good job addressing my concerns, and the revised manuscript is significantly improved. I don't have further comments.

Reviewer #3 (Remarks to the Author):

The authors have addressed the concerns that I raised in my first review and have made a good job in responding to the concerns of the other reviewers in my opinion. I believe that the community would benefit from this being published.

Reviewer #3 (Remarks on code availability):

The file for installation in github is not yet available.

Under section:"Download napSeeker-1.0.tar.gz from <https://github.com/junhong-huang/napSeeker/releases> ; unpack it, and make:

`tar -xzvf napSeeker-1.0.tar.gz`" the link leads to a not available file. Authors need to fix this before publication.

We are very grateful to the reviewers for their positive appraisals of our manuscript. We have carefully revised our manuscript and addressed all the concerns accordingly. The following are our point-by-point responses to the comments/suggestions from the Reviewers.

REVIEWERS' COMMENTS

Reviewer #1 (Remarks to the Author):

I would like to sincerely thank the authors for their work during the review period. I am pleased to see that some of my criticisms and comments were also shared by the other 2 reviewers. I feel that the authors have done their utmost to improve the readability and quality of their work, both by modifying the text and by providing new experimental data. I feel that the revised version is of better quality. If the editors decide to publish this work, I would like the authors to make the following minor changes in the final version of the manuscript:

Response: We highly appreciate the positive appraisals by the reviewer on our manuscript.

#1

JC comment (1st reading) - The authors should discuss the trivial possibility that some (most) of the newly identified RNA species (low abundance, poorly conserved and overlapping interspersed repeated elements) could simply represent degradation products and/or RNA intermediates during pre-mRNA splicing or pri-miRNA processing.

Response to Authors - According to reviewer's suggestion, we have discussed the trivial possibility of the newly identified RNA species in the revised manuscript (Discussion section).

Unless I am mistaken, my comment was not highlighted in the discussion. I think it is important that the authors explicitly mention the possibility that many novel RNA species they described may not have regulatory functions but simply represent non-functional products of degradation which have been captured by increasingly sensible methods of deletion. From my point of view, it is important not to over-interpret the observations, and only more detailed future analyses will be able to demonstrate the evolutionary or physiological significance of these new RNA species.

Response: We greatly appreciate the reviewer for bringing this to our attention. We are sorry for our carelessness in the past. In this revised manuscript, we added the discussion about the trivial possibility that some (most) of the newly identified RNA species, which need more analyses or experimental demonstration in the future (**Line 566-570**). The details are as follows:

On the other hand, some of the newly identified RNA species (low abundance, poorly conserved and overlapping interspersed repeated elements) could simply represent degradation products and/or RNA intermediates during pre-mRNA splicing or pri-miRNA processing. It is important to demonstrate the evolutionary or physiological significance of these new RNA species in the future.

#2

Introduction section – “Furthermore, we demonstrated that a novel structured napRNA

regulates myoblast differentiation in mice and a new napRNA, DINAP, can interact with DKC1 to maintain its protein stability, which further promotes the proliferation of HepG2 cells”

I suggest that the authors mention the fact that DINAP is a C/D and H/ACA hybrid .

Response: Thank you for your constructive comment. We have already modified this sentence, adding a description of DINAP (**revised Introduction, Line 67-68**).

#3

Page 7 – “For instance, the evolutionary conservation of hsa-napRNA-188 2120 is very highly conserved from human to zebrafish (Supplementary Fig. 4b)”.

Replace by “For instance, the hsa-napRNA-188 2120 is highly conserved from human to zebrafish (Supplementary Fig. 4b)”.

Response: Thank you for your constructive suggestion. We had revised the sentence as the reviewer’s advice (**Line 188**).

#4

Page 10 – “We next developed new bioinformatic pipelines to identify novel small nucleolar RNAs 268 (snoRNAs), including C/D box and H/ACA box RNAs, from NAP-seq profiles”. Computer pipelines do not allow direct access to snoRNAs, ie RNAs that accumulate in the nucleoli. This is all the more true as the sub-nucleolar compartmentalisation of some of the candidates tested does not reveal the strong accumulation in nucleoli that is commonly observed with true snoRNAs.

Response: We very much appreciate the Reviewer’s thoughtful and constructive comments. As the reviewer’s comment, snoRNA identified by our bioinformatic pipelines need further experimental verification. To be more precise, we have modified the sentence (**Line 267**) as the following: We next developed bioinformatic pipelines to identify previously undescribed candidates of small nucleolar RNAs (snoRNAs), including C/D box and H/ACA box RNAs, from NAP-seq profiles.

#5

Response to Authors – “In comparison to the endogenous expression, the ectopic expression of mmu-novel-CD-6 was found to be approximately 300-fold”

In order for the reader to be able to judge the artificial and non-physiological nature of the over-expression experiments, the authors should indicate the level of over-expression (300-fo) in the text.

Response: Thank you for your valuable suggestion. We have added fold-change description for the ectopic expression of mmu-novel-CD-6 in the revised manuscript (**Line 315**).

Reviewer #2 (Remarks to the Author):

The authors have done a good job addressing my concerns, and the revised manuscript is significantly improved. I don’t have further comments.

Response: We highly appreciate the positive appraisals by the reviewer on our manuscript.

Reviewer #3 (Remarks to the Author):

The authors have addressed the concerns that I raised in my first review and have made a good job in responding to the concerns of the other reviewers in my opinion. I believe that the community would benefit from this being published.

Response: We are very grateful to the reviewer for the positive appraisals of our manuscript.

Reviewer #3 (Remarks on code availability):

The file for installation in github is not yet available.

Under section: "Download napSeeker-1.0.tar.gz from <https://github.com/junhong-huang/napSeeker/releases> ; unpack it, and make:

tar -xzvf napSeeker-1.0.tar.gz" the link leads to a not available file. Authors need to fix this before publication.

Response: Thanks for the reviewer's comment and we are sorry for our carelessness. In the revised manuscript, we have made the necessary adjustments to the files on GitHub to ensure accessibility.